# ON THE UNIVERSALITY OF SELF-SUPERVISED LEARNING

## ABSTRACT

In this paper, we investigate what constitutes a good representation or model in self-supervised learning (SSL). We argue that a good representation should exhibit universality, characterized by three essential properties: discriminability, generalizability, and transferability. While these capabilities are implicitly desired in most SSL frameworks, existing methods lack an explicit modeling of universality, and its theoretical foundations remain underexplored. To address these gaps, we propose General SSL (GeSSL), a novel framework that explicitly models universality from three complementary dimensions: the optimization objective, the parameter update mechanism, and the learning paradigm. GeSSL integrates a bi-level optimization structure that jointly models task-specific adaptation and cross-task consistency, thereby capturing all three aspects of universality within a unified SSL objective. Furthermore, we derive a theoretical generalization bound, ensuring that the optimization process of GeSSL consistently leads to representations that generalize well to unseen tasks. Empirical results on multiple benchmark datasets demonstrate that GeSSL consistently achieves superior performance across diverse downstream tasks, validating its effectiveness.

## 1 INTRODUCTION

Self-supervised learning (SSL) has revolutionized machine learning by enabling models to learn meaningful representations from unlabeled data, thereby significantly reducing reliance on large labeled datasets (Gui et al., 2024). SSL methods are generally divided into two categories: discriminative SSL (D-SSL) and generative SSL (G-SSL). D-SSL approaches, such as SimCLR (Chen et al., 2020a), BYOL (Grill et al., 2020), and Barlow Twins (Zbontar et al., 2021), focus on distinguishing between different augmented views of the same image, learning representations by maximizing the similarity between positive pairs and minimizing it with negative ones. In contrast, G-SSL methods like MAE (He et al., 2022) aim to reconstruct missing or corrupted parts of the input data, learning representations by capturing inherent visual structures and patterns. Both D-SSL and G-SSL have demonstrated remarkable ability in representation learning.

Whether using D-SSL or G-SSL methods, most researches focus on determining which factors, e.g., network architectures (Caron et al., 2021), optimization strategies (Ni et al., 2021), prior assumptions (Ermolov et al., 2021), inductive biases (Grill et al., 2020), etc., lead to good representations or models. However, a fundamental question persists: Why these factors can lead to a "good" representation or model? To address this question, the common practice is to evaluate the learned representations or models on various downstream tasks, that is, if the performance is strong, the representation or model is deemed good. Yet, a key challenge remains in understanding the underlying mechanisms by which these factors yield a good representation or model. In other words, we often lack direct explanations of how specific methodological choices influence the quality of the representation or model. For instance, why does an asymmetric dual-branch network architecture in methods like BYOL enhance performance on downstream tasks? Similarly, why does enforcing a uniform distribution on feature representations serve as an inductive bias for obtaining good representations in methods like SimCLR?

In this paper, we shift focus from designing SSL methods in terms of "which factors should be adopted" to exploring "what directly constitutes a good representation or model". The advantage of this shift is that, once we have identified the key properties that define a good representation, we

can directly incorporate these properties into the optimization objective of SSL. As a result, we no longer need to justify whether a particular "do" operation can implicitly lead to good representations, because the representation itself is explicitly modeled as the target of learning. Thus, we concentrate on the question: What characteristics should a good representation or model possess? Inspired by the evaluation methods of most SSL and unsupervised learning approaches (Chen et al., 2020a; Grill et al., 2020; He et al., 2022), we answer this question by that a good representation or model should satisfy three constraints: 1) Discriminability: For a single task, the model should achieve the expected performance on the training set; 2) Generalizability: For a single task, the trained model should generalize to unseen datasets while maintaining its performance; 3) Transferability: The trained model should generalize to multiple different tasks while guaranteeing its performance. We next consolidate the three dimensions, e.g., discriminability, generalization, and transferability, into a single criterion: universality. When these capabilities are jointly satisfied within one framework, the resulting representation or model is said to possess high universality. Hence, a "good" representation or model can be succinctly defined as one with high universality: it separates classes well on the source task, generalizes robustly to new yet in-distribution data, and transfers efficiently to unfamiliar tasks, thereby furnishing a reliable, reusable foundation for diverse downstream objectives.

Given the definition of Universality, a central challenge is how to formally integrate its properties into the SSL process. To address this, we propose General SSL (GeSSL), a unified framework that explicitly models Universality by embedding its three core components, e.g., discriminability, generalizability, and transferability, into SSL training. For discriminability, GeSSL not only leverages alignment-based objectives commonly used in existing SSL methods but also introduces an additional discriminative loss to improve class separation in an unsupervised setting. For generalizability, GeSSL employs a bi-level optimization mechanism that separates training data into support and query sets, thereby simulating an update-then-evaluate process to directly model generalization behavior. Moreover, it encourages shared feature extractor across multiple tasks, which indirectly enhances generalization by promoting causal consistency. Lastly, for transferability, GeSSL adopts an episodic training paradigm in which multiple mini-batches are treated as distinct tasks, allowing the model to estimate and adapt to the underlying task distribution and generalize to unseen scenarios. In this way, GeSSL provides a principled approach to modeling Universality within the SSL framework. To further establish the soundness of this framework, we provide formal performance guarantees showing that GeSSL's training objective leads to bounded generalization error on novel tasks. This is achieved under smoothness and boundedness assumptions, demonstrating that the jointly optimized representation can be reliably adapted to unseen tasks with good performance.

**Our contributions**: **(i)** We theoretically define SSL universality, encompassing discriminability, generalizability, and transferability (Sections 3.1). **(ii)** We propose GeSSL, a novel framework that models universality through a bi-level learning paradigm (Section 3.2). **(iii)** Theoretical and empirical evaluations on benchmark datasets demonstrate the advantages of GeSSL (Sections 4, 5).

## 2 REVISITING SSL FROM A TASK PERSPECTIVE

During the training phase, the data is organized into mini-batches, i.e., a mini-batch is denoted as $X_{tr} = \{x_i\}_{i=1}^N$, where $x_i$ is the $i$-th sample, and $N$ is the batch size. In D-SSL, each sample $x_i$ undergoes stochastic data augmentation to generate two augmented views, i.e., $x_i^1$ and $x_i^2$. In G-SSL, each sample $x_i$ is partitioned into multiple small blocks, some blocks are masked, and the remaining blocks are reassembled into a new sample $x_i^1$. The original sample is then referred to as $x_i^2$. We can also regard the partition-reassemble operation as augmentation. Each augmented dataset in both D-SSL and G-SSL is represented as $X_{tr}^{aug} = \{x_i^1, x_i^2\}_{i=1}^N$. Each $\{x_i^1, x_i^2\}$ constitutes the $i$-th sample pair, and the SSL objective is to learn a feature extractor $f$ from these pairs.

D-SSL methods typically have two main objectives: alignment and regularization (Chen et al., 2020a; Oord et al., 2018; Hjelm et al., 2018). The alignment objective maximizes the similarity between paired samples in the embedding space, while the regularization objective constrains the learning behavior via inductive biases. For example, SimCLR (Chen et al., 2020a) enforces a uniform distribution over the feature representations. G-SSL methods (He et al., 2022) can also be viewed as implementing alignment within a pair using an encoding-decoding structure: sample $x_i^1$ is input into this structure to generate an output that is made as consistent as possible with sample $x_i^2$. Notably, alignment in D-SSL is often implemented using anchor points, where one sample in a

pair is viewed as the anchor, and the training process gradually pulls the other sample towards this anchor. This concept of an anchor is also applicable to G-SSL, where $x_i^2$ is treated as the anchor, and the training process involves constraining $x_i^1$ to approach $x_i^2$.

Regardless of whether it is G-SSL or D-SSL, the anchor can be regarded as a learning target. Specifically, SSL can be interpreted as follows: In a data augmentation pair, one sample (the anchor) is designated as the target. By constraining the other augmented sample in the feature space to move toward this anchor, consistency in feature representations is achieved. This dynamic adjustment causes samples within the same pair to become tightly clustered, thus, the anchor plays a role similar to that of a clustering center. In other words, for a mini-batch $X_{tr}^{aug}$ in SSL, each pair within the batch can be considered as belonging to a specific class, where the class center serves as the anchor. Thus, $X_{tr}^{aug}$ can be interpreted as a multi-class classification task with $N$ classes. Given the role of the "alignment part" in SSL, the learning process within a single mini-batch can be viewed as performing a classification task. More details are provided in Appendix G.3 and Appendix H.4.

## 3 METHODOLOGY

In this section, we first present the definition and explanation of universality. Then, we propose a way to model universality in SSL, i.e., General SSL (GeSSL). The framework is illustrated in Figure 1. Finally, we give some high-level explanations for the proposed modeling method.

### 3.1 DEFINITION AND EXPLANATION OF UNIVERSALITY

Typically, a learnable task consists of a training dataset and a test dataset, where the training dataset is used to train the model and the test dataset is used to evaluate its performance. Each element in the training or test dataset is usually represented as a tuple consisting of an input sample and its corresponding label. If we treat a task as a basic unit, then each task can be viewed as being sampled from a task distribution $P_t$. Based on this, we present the definition of universality as follows:

**Definition 3.1 (Universality)** *For a set of training tasks and a disjoint set of test tasks, i.e., with no class-level overlap and each sample is with a label, the model $f_\theta$, or the representation extracted by it, is said to exhibit universality if it satisfies: **1) Discriminability**: For a training task with labeled training dataset, a model $f_\theta$ trained on the training dataset can predict the labels of all training samples with high accuracy. **2) Generalizability**: For a training task with training and test datasets, a model $f_\theta$ trained on the training dataset can predict the labels of all test samples with high accuracy. **3) Transferability**: For a training task and a test task, $f_\theta$ trained on the training dataset of the training task can predict the labels of all samples of the test task with high accuracy.*

Discriminability, Generalizability, and Transferability are not new concepts. At first glance, Universality may seem like a simple combination of these three and therefore lacks novelty. However, the core purpose of proposing **Universality** is to more clearly answer a fundamental question: what "directly" constitutes a good representation or model. When we revisit SSL from this perspective, the value of **Universality** becomes evident, because it directly quantifies the essential qualities that a good representation should possess, thus forming a sharp contrast with the motivations of existing SSL methods. We next explain this argument further.

Existing SSL methods typically follow the idea of "do what can lead to a good representation", such as using contrastive learning or masked prediction. However, why these operations lead to good representations often requires strong prior assumptions or extensive empirical validation, making them costly and difficultly to explain. In contrast, **Universality** explicitly incorporates the three key capabilities of a good representation into a unified learning objective, making the "do what" much more straightforward: find an SSL method that directly models **Universality** in its training objective. This not only makes it easier to explain the effectiveness of the "do what" (since the training objective itself is **Universality**), but also greatly simplifies the technical path for identifying truly effective learning strategies. Using causal paths as a metaphor, the difference between the two approaches can be described as follows: 1) Traditional SSL: "do what → universality → good representation"; 2) Our proposed SSL framework: "universality → good representation". Therefore, explicitly proposing and defining **Universality** not only offers a new conceptual perspective for designing SSL meth-

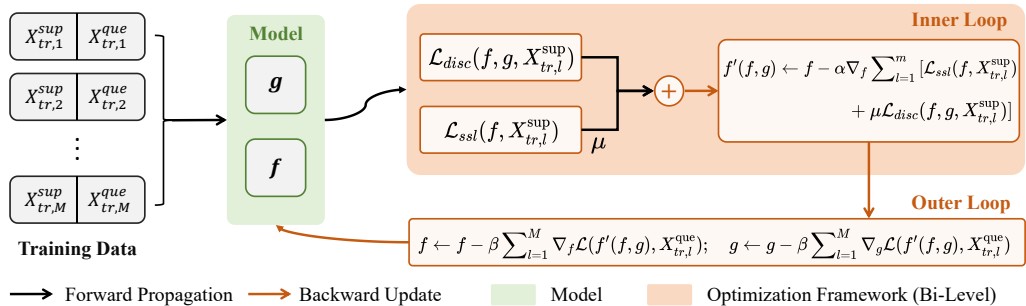

Figure 1: Overview of GeSSL. $X_{tr,l}^{sup}$ and $X_{tr,l}^{que}$ denote the *support* and *query* splits of the $l$-th mini-batch. The black arrows denote forward propagation; orange arrows denote parameter updates (backward propagation); green boxes denote model components ($f, g$); the orange region denotes the bi-level optimization (inner and outer loops).

ods, but also opens up a more direct and interpretable path for practical implementation, e.g., how to model **Universality**. It thus carries both conceptual novelty and practical value.

## 3.2 EXPLICIT MODELING UNIVERSALITY IN SSL

Let the mini-batches be denoted by $X^{\text{aug}} = \{X_{tr,l}^{\text{aug}}\}_{l=1}^m$, where $l$ indexes the $l$-th mini-batch. The $l$-th mini-batch is denoted as $X_{tr,l}^{\text{aug}} = \{x_{i,l}^1, x_{i,l}^2, x_{i,l}^{\text{anchor}}\}_{i=1}^n$, with $\{x_{i,l}^1, x_{i,l}^2, x_{i,l}^{\text{anchor}}\}$ representing the $i$-th pair and $x_{i,l}^{\text{anchor}}$ is the anchor within that pair. For D-SSL methods, all three samples in a pair originate from the same source image and are produced by applying different data-augmentation pipelines to that image. Any of the three augmented views may serve as the anchor. Whichever sample is chosen, the triplet can always be written as $\{x_{i,l}^1, x_{i,l}^2, x_{i,l}^{\text{anchor}}\}$. To keep the notation concise, we do not label which element is the anchor, instead, we treat pairs with different anchor choices as distinct pairs indexed by $i$. For G-SSL methods, the anchor of each pair is the source sample. The remaining two elements, $x_{i,l}^1$ and $x_{i,l}^2$, are generated from that anchor by applying two different masking operations. A list of notations is provided in **Appendix A**.

During training, every mini-batch is divided at the pair level into two disjoint subsets: support set $X_{tr,l}^{\text{sup}} = \{x_{i,l}^1, x_{i,l}^{\text{anchor}}\}_{i=1}^n$ and query set $X_{tr,l}^{\text{que}} = \{x_{i,l}^2, x_{i,l}^{\text{anchor}}\}_{i=1}^n$. At each training step, $m$ such mini-batches are processed in parallel. The GeSSL objective over these mini-batches can be presented as:

$$\min_{f,g} \sum_{l=1}^m \mathcal{L}_{ssl}(f', X_{tr,l}^{\text{que}}), \text{ s.t. } f' = \arg\min_f \sum_{l=1}^m [\mathcal{L}_{ssl}(f, X_{tr,l}^{\text{sup}}) + \mu\mathcal{L}_{disc}(f, g, X_{tr,l}^{\text{sup}})], \quad (1)$$

where $\mathcal{L}_{ssl}(\cdot)$ represents the loss function in SSL method, e.g., the contrastive loss in D-SSL and the MSE loss in G-SSL, $\mu$ is a hyperparameter, and $\mathcal{L}_{disc}(f, g, X_{tr,l}^{\text{sup}})$ is a defined discriminative loss:

$$\mathcal{L}_{disc}(f, g, X_{tr,l}^{\text{sup}}) = \sum_{i=1}^n \sum_{j=1}^n [\mathbb{1}_{\{d_j^i \leq a_i\}} \cdot d_j^i + \mathbb{1}_{\{d_j^i > a_i\}} \cdot (-d_j^i)], \quad (2)$$

where $a_i \in \mathbb{R}$ is the output of the function $g$, the input to $g$ is the mean and covariance matrix of the vector set $\{f(x_{j,l}^1) - f(x_{i,l}^{\text{anchor}})\}_{j=1}^n$, $\mathbb{1}_{\{\cdot\}}$ denotes the indicator function, which evaluates whether a given condition is satisfied, $d_j^i = d(f(x_{j,l}^1), f(x_{i,l}^{\text{anchor}}))$, and $d(\cdot)$ is the cosine distance. As we can see, minimizing $\mathcal{L}_{disc}(f, g, X_{tr,l}^{\text{sup}})$ can be interpreted as: 1) when $d_j^i \leq a_i$, minimize $d_j^i$; 2) when $d_j^i > a_i$, maximize $d_j^i$. Since the indicator function is non-differentiable, optimizing $g$ leads to a zero-gradient problem. Therefore, we replace Equation (2) with a differentiable approximation. Based on (Maddison et al., 2016; Musgrave et al., 2020; Sohn, 2016), this differentiable approximation can be expressed as:

$$\mathcal{L}_{disc}(f, g, X_{tr,l}^{\text{sup}}) = \sum_{i=1}^n \sum_{j=1}^n \left[ w(a_i) \cdot d_j^i + (1 - w(a_i)) \cdot (-d_j^i) \right], \quad (3)$$

where $w(a_i) = \text{Sigmoid}(k \cdot (a_i - d_j^i))$, and $k \in \mathbb{R}^+$ is a hyperparameter. More discussions about the theoretical motivation behind our objective design are provided in **Appendix H.3**. Finally, the specific optimization process of Equation (1) is divided into the following two steps of iteration:

**Inner-Loop Optimization:** In this step, $g$ is fixed, and GeSSL learns a proxy model $f'$ by minimizing the constraint of Equation (1). The update of $f'$ can be obtained by the follows:

$$f'(f,g) \leftarrow f - \alpha \nabla_f \sum_{l=1}^{m} [\mathcal{L}_{ssl}(f, X_{tr,l}^{\text{sup}}) + \mu \mathcal{L}_{disc}(f, g, X_{tr,l}^{\text{sup}})], \qquad (4)$$

where $\alpha$ is the learning rate, and $f'(f,g)$ represents that $f'$ is a function of $f$ and $g$, this is because that Equation (4) explicitly represents the process of updating parameters based on gradient descent. Typically, $f'$ undergoes $\varsigma$ updates by executing Equation (4) $\varsigma$ times, and each resulting $f'$ can be expressed as a function of $f$ and $g$, i.e., $f' = f'(f,g)$. We set $\varsigma = 1$ for computational convenience. It should be noted that the terms $\mathcal{L}_{ssl}$ and $\mathcal{L}_{disc}$ are all calculated based on the support set.

**Outer-Loop Optimization:** In this step, GeSSL learns the optimal model $f$ and $g$, based on the agent model $f'$. The learning process is presented as the follows:

$$f \leftarrow f - \beta \sum_{l=1}^{m} \nabla_f \mathcal{L}_{ssl}(f'(f,g), X_{tr,l}^{\text{que}}); \quad g \leftarrow g - \beta \sum_{l=1}^{m} \nabla_g \mathcal{L}_{ssl}(f'(f,g), X_{tr,l}^{\text{que}}), \quad (5)$$

where $\beta$ is the learning rate, $\mathcal{L}_{ssl}(f'(f,g), X_{tr,l}^{\text{que}})$ is calculated based on the query set and the proxy model $f'$. From Equation (5), $f'$ contains both the parameters of $f$ and the first-order derivatives of $f$, and since $\mathcal{L}_{ssl}(f'(f,g), X_{tr,l}^{\text{que}})$ can be viewed as a function of $f'(f,g)$, the gradient of $\mathcal{L}_{ssl}(f'(f,g), X_{tr,l}^{\text{que}})$ with respect to $f$ involves both the first and second order derivatives of $f$.

**Explanation for Bi-Level Optimization**: Equation (2) belongs to a bi-level optimization objective. The inner constraint aims to learn a proxy model $f'$, while the outer objective is to ultimately learn a better $f$ based on $f'$. The advantage of this design lies in the fact that it enables $\mathcal{L}_{ssl}$ to be minimized twice, thereby potentially yielding a better $f$. Specifically, the first minimization of $\mathcal{L}_{ssl}$ occurs during the learning of $f'$, as obtaining $f'$ involves minimizing $\mathcal{L}_{ssl}$. As discussed in Section 2, a single mini-batch in SSL can be regarded as a task, and dividing a task's dataset into a support set and a query set allows us to reasonably assume that both sets are drawn from the same underlying distribution. Therefore, if $f'$ minimizes $\mathcal{L}_{ssl}$ on the support set, it can also be expected to perform well on the query set. Moreover, in the outer objective of Equation (2), we further adjust $f$ to obtain a new $f'$ such that the value of $\mathcal{L}_{ssl}$ computed using this updated $f'$ is lower than that computed using the previous $f'$, thus, this constitutes the second minimization. Also, this process can be interpreted as modeling the behavior of selecting the best among many local minima. Hence, compared to jointly optimizing the outer and inner objectives in a single stage, the bi-level optimization framework can yield a better $f$, because only a better $f$ leads to a lower $\mathcal{L}_{ssl}$ when propagated through $f'$.

**Explanation for Discriminability**: GeSSL models discriminability from two complementary perspectives. The first dimension is captured via the loss $\mathcal{L}_{ssl}$, while the second dimension is addressed through the loss $\mathcal{L}_{disc}$. In the first dimension, both D-SSL and G-SSL formulations of $\mathcal{L}_{ssl}$ include an "alignment term", which enforces each augmented sample in a pair to align closely with its corresponding anchor. According to Section 2, each mini-batch can be regarded as a task, where samples within the same pair are assumed to belong to the same semantic class, while samples across different pairs correspond to different classes. The anchor sample within each class acts as a proxy for the class center. Therefore, minimizing the alignment term effectively encourages intra-class compactness by pulling together samples of the same class. From this perspective, $\mathcal{L}_{ssl}$ implicitly models discriminability. However, this modeling is limited due to the coarse class assignment strategy in SSL. Specifically, current SSL methods treat augmented views derived from the same source as belonging to the same class, while failing to consider that different sources may actually belong to the same underlying class. As a result, augmented samples from different sources, but sharing semantic similarity, are not encouraged to align. Furthermore, $\mathcal{L}_{ssl}$ lacks an explicit mechanism for pushing apart samples from different classes. Hence, relying solely on $\mathcal{L}_{ssl}$ under this simplistic partitioning is insufficient for capturing a rich notion of discriminability.

To address the above limitation, GeSSL introduces $\mathcal{L}_{disc}$ as a second mechanism to enhance discriminability. Minimizing $\mathcal{L}_{disc}$ can be interpreted as follows: for a given anchor and a learnable threshold, if the distance between the anchor and an augmented sample is less than or equal to the threshold, the sample is pulled closer to the anchor, otherwise, it is pushed farther away. This operation is applied across all anchors, enabling the model to approximately group together semantically similar samples while separating dissimilar ones. In essence, $\mathcal{L}_{disc}$ compensates for the shortcomings of $\mathcal{L}_{ssl}$ by explicitly promoting both intra-class compactness and inter-class separability. However, the effectiveness of $\mathcal{L}_{disc}$ hinges on the quality of the threshold: an inaccurate threshold can

undermine its discriminative power. This is where the bi-level optimization framework in GeSSL plays a crucial role. The threshold, like the model $f$, is learned by adjusting an auxiliary network $g$, such that the proxy model $f'$ computed based on $f$ and $g$ leads to a lower $\mathcal{L}_{ssl}$. In other words, GeSSL optimizes $g$ so that the learned threshold helps find a better local minimum of $\mathcal{L}_{ssl}$ via $f'$. Only when $g$ learns an accurate threshold can this two-stage minimization be effective. Thus, the bi-level framework implicitly regularizes the learning of $g$, ensuring its accuracy. Ultimately, this design guarantees that the introduction of $\mathcal{L}_{disc}$, guided by a well-learned threshold, enables GeSSL to model discriminability in a more comprehensive and principled manner.

**Explanation for Generalizability**: GeSSL models generalizability along two dimensions. The first is direct modeling. During training, each mini-batch task is split into a support set and a query set with no overlap. GeSSL first fine-tunes the base model $f$ on the support set to obtain a task-specific model $f'$, then evaluates $f'$ on the query set and uses this feedback to update $f$ via backpropagation. This involves two gradient updates: the first adapts $f$ into $f'$, and the second refines $f$ based on query-set performance. Through this "update-then-evaluate" mechanism, the query set functions not only as training data but also as a simulated test set. In this sense, the support and query sets mirror the roles of training and test sets in conventional learning: the support set drives task adaptation, while the query set evaluates generalization. Unlike standard training, however, GeSSL incorporates "performing well on the test set" as an explicit training objective, thereby directly modeling generalizability. The second dimension is "indirect modeling". Prior work shows that strong performance across diverse tasks often indicates that a model has learned causal representations. (Ahuja et al., 2020) argue that consistent task performance suggests the capture of stable causal features, while (Schölkopf et al., 2021) and (Ahuja et al., 2023) further assert that causal representations are sufficient for generalization. Building on this foundation, GeSSL's training strategy implicitly enforces causal modeling: in each round, multiple mini-batches corresponding to different tasks are jointly processed. The same $f$ and its adapted version $f'$ are applied across all tasks, and this cross-task consistency constraint encourages the discovery of stable, shared features. As a result, GeSSL indirectly models generalizability by promoting the learning of causal representations.

**Explanation for Transferability**: The training process of GeSSL can be regarded as an episodic learning process like meta-learning. Specifically, each episode of GeSSL consists of $m$ mini-batch tasks, and the entire learning process can be divided into multiple episodes. Based on Section 2, we consider the learning process of GeSSL as estimating the true task distribution from discrete training tasks, which enables the GeSSL model to generalize to new, unseen tasks (i.e., test tasks). Therefore, we conclude that GeSSL achieves model transferability through its learning paradigm. More disucssions are provided in **Appendix H.2** due to space limitations.

**Comparison of GeSSL and Meta-Learning**: Note that the primary goal of GeSSL is to provide an effective method for modeling Universality. In other words, GeSSL seeks to answer the question: how can Universality be modeled? Its design is explicitly inspired by the concept of Universality. At the same time, it is crucial to highlight the differences between GeSSL and traditional meta-learning methods. First, meta-learning relies on explicit supervision, whereas GeSSL operates in a self-supervised manner, constructing its own pseudo-labels without the need for manual annotations. Second, although both methods follow an episodic training paradigm, meta-learning benefits from accurate supervision, which allows it to model discriminability effectively. In contrast, GeSSL relies on heuristically constructed labels that are often noisy, which can hinder its ability to model discriminability accurately. To address this, GeSSL introduces an additional loss term $\mathcal{L}_{disc}$ to mitigate the impact of noisy labels and enhance its capacity to model true class structures. Moreover, in meta-learning, a separate task-specific model is learned for each task, meaning different tasks have different adapted models. GeSSL, on the other hand, learns a single unified adapted model $f'$ for all tasks. This key difference enables GeSSL to better capture shared structures across tasks, thereby learning representations with stronger generalizability. In summary, GeSSL is not only distinct from traditional meta-learning in its methodology and learning signals but also differs fundamentally from approaches that directly transplant meta-learning paradigms into SSL. GeSSL's design reflects a unique theoretical motivation and a novel approach to learning universal representation.

# 4 THEORETICAL ANALYSIS

In this section, we provide performance guarantees. Specifically, we prove that through the objective of GeSSL (Equation (5)), the performance of the SSL model on new tasks is guaranteed. We assume a task distribution $\mathcal{T}$, where each task $\tau \sim \mathcal{T}$ comprises a support set $S_\tau$ (used for rapid task-specific adaptation) and a query set $Q_\tau$ (used to evaluate generalization). For any parameter vector $\theta$ and task $\tau$, we denote the supervised loss incurred on the unseen query set $Q_\tau$ as $\mathcal{L}_{\sup}(\theta; \tau)$; the self-supervised and discriminative losses on the support set $S_\tau$ as $\mathcal{L}_{ssl}(\theta; S_\tau)$ and $\mathcal{L}_{disc}(\theta; S_\tau)$; letting $\theta' = A(\theta, S_\tau)$ be the adapted parameters after applying the adaptation operator $A$ to $\theta$ using $S_\tau$; $\mathcal{L}_{query}(\theta'; Q_\tau)$ be the resulting SSL loss in query set. By jointly optimizing these losses, our goal is to learn representations that are both transferable and generalizable to new tasks while ensuring discriminative performance. Next, we provide the main theorem with proofs in Appendix B.

---

**Performance Guarantee of GeSSL**

**Theorem 4.1** *Let $\theta^*$ denote the parameter after bi-level training over $N$ tasks (mini-batches). For any new task $\tau_{test} \sim \mathcal{T}$, let $\theta^*_{test} = A(\theta^*, S_{\tau_{test}})$ denote the adapted parameter, under Assumption B.1, with probability at least $1 - \delta$, we have:*

$$\mathbb{E}_{\tau_{test}}\left[\mathcal{L}_{sup}(\theta^*_{test}; \tau_{test})\right] \leq \frac{1}{N}\sum_{i=1}^{N}\left[\mathcal{L}_{ssl}(\theta', S_{\tau_i}) + \mathcal{L}_{disc}(\theta', S_{\tau_i}) + \mathcal{L}_{query}(\theta', Q_{\tau_i})\right] + \mathcal{O}\left(\sqrt{\frac{1}{N}\ln\frac{1}{\delta}}\right), \quad (6)$$

*where $\theta'$ is the adapted parameter for training task $\tau_i$ (the $i$-th mini-batch).*

---

*Proof sketch.* We decompose the expected test loss into an adaptation term and a generalization term. Assumptions B.1 ensure that the inner update strictly reduces the SSL with discriminative loss and that its effect on the parameters is small, yielding an adaptation error of order $O(1/\sqrt{m})$. Since tasks are i.i.d. and losses are bounded, Hoeffding's inequality shows that the empirical bi-level objective concentrates around its expectation within $O(\sqrt{\ln(1/\delta)/N})$. Summing both terms gives the final bound: the expected supervised loss on new tasks is controlled by the empirical objective plus these two vanishing error terms. It states that, under standard assumptions, the bi-level training procedure of GeSSL, which jointly optimizes the SSL loss, the discriminative loss (to enhance class separability), and the query loss (to guarantee generalization to new tasks), provides an upper bound of order $\mathcal{O}(\sqrt{\ln(1/\delta)/N})$ on the supervised loss for unseen tasks. This result formally validates both the effectiveness and the broad applicability of the GeSSL strategy.

# 5 EMPIRICAL EVALUATION

In this section, we conduct extensive experiments on various settings to verify the effectiveness of GeSSL. For unsupervised and semi-supervised learning, we select CIFAR-10 (Krizhevsky et al., 2009), CIFAR-100 (Krizhevsky et al., 2009), STL-10 (Coates et al., 2011), Tiny ImageNet (Le & Yang, 2015), ImageNet-100 (Tian et al., 2020a) and ImageNet (Deng et al., 2009a); For transfer learning, we select PASCAL VOC (Everingham et al., 2010), COCO (Lin et al., 2014), Flower102 (Nilsback & Zisserman, 2008), Food101 (Bossard et al., 2014), etc.; For few-shot learning, we select Omniglot (Lake et al., 2019), miniImageNet (Vinyals et al., 2016a), CIFAR-FS (Bertinetto et al., 2018), CUB (Welinder et al., 2010), Cars (Krause et al., 2013), etc., for evaluation. We select both D-SSL and G-SSL baselines for comparison. All results are reported via five runs on NVIDIA 4090 GPUs. More details and additional results are provided in Appendix C-G.

## 5.1 PERFORMANCE COMPARISON

**Unsupervised Learning.** We adopt the most commonly used protocol (Chen et al., 2020a), freezing the feature extractor and training a linear classifier on top of it. We use Adam (Kingma & Ba, 2014) with Momentum and weight decay set at $0.8$ and $10^{-4}$. The linear classifier runs for 500 epochs with a batch size of 128 and a learning rate that starts at $5 \times 10^{-2}$ and decays to $5 \times 10^{-6}$. We use ResNet-18 for small-scale datasets (CIFAR-10, CIFAR-100, STL-10, and Tiny ImageNet) while using ResNet-50 for the medium-scale (ImageNet-100) and large-scale (ImageNet) datasets. Table 1 shows that applying GeSSL significantly outperforms the state-of-the-art (SOTA) methods on all datasets and SSL baselines. The results demonstrate its ability to enhance SSL performance. To

Table 1: The Top-1 and Top-5 classification accuracies of linear classifier on the ImageNet-100 and ImageNet (200 Epochs) with ResNet-50.

| Method | ImageNet-100 | | ImageNet | |
|---|---|---|---|---|
| | Top-1 | Top-5 | Top-1 | Top-5 |
| W-MSE (Ermolov et al., 2021) | 76.01 ± 0.27 | 93.12 ± 0.21 | 70.85 ± 0.31 | 91.57 ± 0.20 |
| RELIC v2 (Tomasev et al., 2022) | 75.88 ± 0.15 | 93.52 ± 0.13 | 70.98 ± 0.21 | 91.15 ± 0.26 |
| LMCL (Chen et al., 2021) | 75.89 ± 0.19 | 92.89 ± 0.28 | 70.83 ± 0.26 | 90.04 ± 0.21 |
| ReSSL (Zheng et al., 2021) | 75.77 ± 0.21 | 92.91 ± 0.27 | 69.92 ± 0.24 | 91.25 ± 0.12 |
| CorInfoMax (Ozsoy et al., 2022) | 75.54 ± 0.20 | 92.23 ± 0.25 | 70.83 ± 0.15 | 91.53 ± 0.22 |
| MEC (Liu et al., 2022) | 75.38 ± 0.17 | 92.84 ± 0.20 | 70.34 ± 0.27 | 91.25 ± 0.38 |
| SimCLR (Chen et al., 2020a) | 70.15 ± 0.16 | 89.75 ± 0.14 | 68.32 ± 0.31 | 89.76 ± 0.23 |
| SimCLR + GeSSL | 72.96 ± 0.24 | 92.50 ± 0.17 | 69.88 ± 0.21 | 91.32 ± 0.25 |
| MoCo (Chen et al., 2020b) | 72.80 ± 0.12 | 91.64 ± 0.11 | 67.55 ± 0.27 | 88.42 ± 0.11 |
| MoCo + GeSSL | 74.35 ± 0.24 | 94.10 ± 0.31 | 69.60 ± 0.30 | 91.28 ± 0.39 |
| SimSiam (Chen & He, 2021) | 73.01 ± 0.21 | 92.61 ± 0.27 | 70.02 ± 0.14 | 88.76 ± 0.23 |
| SimSiam + GeSSL | 75.93 ± 0.24 | 95.51 ± 0.38 | 72.04 ± 0.22 | 89.43 ± 0.40 |
| Barlow Twins (Zbontar et al., 2021) | 75.97 ± 0.23 | 92.91 ± 0.19 | 69.94 ± 0.32 | 88.97 ± 0.27 |
| Barlow Twins + GeSSL | 77.55 ± 0.29 | 93.48 ± 0.30 | 72.84 ± 0.26 | 89.50 ± 0.19 |
| MAE (He et al., 2022) | 76.56 ± 0.16 | 93.24 ± 0.24 | 70.73 ± 0.25 | 91.41 ± 0.27 |
| MAE + GeSSL | 78.45 ± 0.31 | **96.17 ± 0.26** | 71.45 ± 0.24 | 92.19 ± 0.30 |
| DINO (Caron et al., 2021) | 75.43 ± 0.18 | 93.32 ± 0.19 | 70.58 ± 0.24 | 91.32 ± 0.27 |
| DINO + GeSSL | 77.13 ± 0.29 | 95.75 ± 0.30 | 73.52 ± 0.30 | **94.05 ± 0.26** |
| VICRegL (Bardes et al., 2022) | 75.96 ± 0.19 | 92.97 ± 0.26 | 70.24 ± 0.27 | 91.60 ± 0.24 |
| VICRegL + GeSSL | **78.48 ± 0.34** | 95.90 ± 0.17 | **73.91 ± 0.36** | 93.77 ± 0.35 |

Table 2: The semi-supervised learning accuracies (± 95% confidence interval) on ImageNet with the ResNet-50 pre-trained on Imagenet.

| Method | Epochs | 1% | | 10% | |
|---|---|---|---|---|---|
| | | Top-1 | Top-5 | Top-1 | Top-5 |
| MoCo (Chen et al., 2020b) | 200 | 43.8 ± 0.2 | 72.3 ± 0.1 | 61.9 ± 0.1 | 84.6 ± 0.2 |
| MoCo + GeSSL | 200 | 46.6 ± 0.3 | 74.5 ± 0.3 | 63.8 ± 0.2 | 85.9 ± 0.2 |
| BYOL (Grill et al., 2020) | 200 | 54.8 ± 0.2 | 78.8 ± 0.1 | 68.0 ± 0.2 | 88.5 ± 0.2 |
| BYOL + GeSSL | 200 | **57.3 ± 0.2** | **79.8 ± 0.2** | **71.1 ± 0.2** | **90.1 ± 0.3** |
| SimSiam (Chen & He, 2021) | 1000 | 54.9 ± 0.2 | 79.5 ± 0.2 | 68.0 ± 0.1 | 89.0 ± 0.3 |
| RELIC v2 (Tomasev et al., 2022) | 1000 | 55.2 ± 0.2 | 80.0 ± 0.1 | 68.0 ± 0.2 | 88.9 ± 0.2 |
| LMCL (Chen et al., 2021) | 1000 | 54.8 ± 0.2 | 79.4 ± 0.2 | 70.3 ± 0.1 | 89.9 ± 0.2 |
| ReSSL (Zheng et al., 2021) | 1000 | 55.0 ± 0.1 | 79.6 ± 0.3 | 69.9 ± 0.1 | 89.7 ± 0.1 |
| SSL-HSIC (Li et al., 2021) | 1000 | 55.4 ± 0.3 | 80.1 ± 0.2 | 70.4 ± 0.1 | 90.0 ± 0.1 |
| CorInfoMax (Ozsoy et al., 2022) | 1000 | 55.0 ± 0.2 | 79.6 ± 0.3 | 70.3 ± 0.2 | 89.3 ± 0.2 |
| MEC (Liu et al., 2022) | 1000 | 54.8 ± 0.1 | 79.4 ± 0.2 | 70.0 ± 0.1 | 89.1 ± 0.1 |
| VICRegL (Bardes et al., 2022) | 1000 | 54.9 ± 0.1 | 79.6 ± 0.2 | 67.2 ± 0.1 | 89.4 ± 0.2 |
| SimCLR (Chen et al., 2020a) | 1000 | 48.3 ± 0.2 | 75.5 ± 0.1 | 65.6 ± 0.1 | 87.8 ± 0.2 |
| SimCLR + GeSSL | 1000 | 51.1 ± 0.2 | 77.7 ± 0.1 | 67.8 ± 0.3 | 89.8 ± 0.3 |
| MoCo (Chen et al., 2020b) | 1000 | 52.3 ± 0.1 | 77.9 ± 0.2 | 68.4 ± 0.1 | 88.0 ± 0.2 |
| MoCo + GeSSL | 1000 | 54.0 ± 0.3 | 78.8 ± 0.1 | 71.6 ± 0.2 | 89.3 ± 0.2 |
| BYOL (Grill et al., 2020) | 1000 | 56.3 ± 0.2 | 79.6 ± 0.2 | 69.7 ± 0.2 | 89.3 ± 0.1 |
| BYOL + GeSSL | 1000 | **59.6 ± 0.3** | **81.9 ± 0.2** | **71.8 ± 0.2** | 91.3 ± 0.2 |
| Barlow Twins (Zbontar et al., 2021) | 1000 | 55.0 ± 0.1 | 79.2 ± 0.1 | 67.7 ± 0.2 | 89.3 ± 0.2 |
| Barlow Twins + GeSSL | 1000 | 58.1 ± 0.3 | 80.5 ± 0.2 | 68.9 ± 0.2 | **92.3 ± 0.3** |

Table 3: The results of transfer learning on object detection and instance segmentation with C4-backbone as the feature extractor. "AP" is the average precision, "$AP_N$" represents the average precision when the IoU (Intersection and Union Ratio) threshold is $N\%$.

| Method | VOC 07 detection | | | VOC 07+12 detection | | | COCO detection | | | COCO instance segmentation | | |
|---|---|---|---|---|---|---|---|---|---|---|---|---|
| | $AP_{50}$ | AP | $AP_{75}$ | $AP_{50}$ | AP | $AP_{75}$ | $AP_{50}$ | AP | $AP_{75}$ | $AP_{50}^{mask}$ | $AP^{mask}$ | $AP_{75}^{mask}$ |
| Supervised | 74.4 | 42.4 | 42.7 | 81.3 | 53.5 | 58.8 | 58.2 | 38.2 | 41.2 | 54.7 | 33.3 | 35.2 |
| Barlow Twins (Zbontar et al., 2021) | 75.7 | 47.2 | 50.3 | 82.6 | 56.8 | 63.4 | 59.0 | 39.2 | 42.5 | 56.0 | 34.3 | 36.5 |
| MEC (Liu et al., 2022) | 77.4 | 48.3 | 52.3 | 82.8 | 57.5 | 64.5 | 59.8 | 39.8 | 43.2 | 56.3 | 34.7 | 36.8 |
| RELIC v2 (Tomasev et al., 2022) | 76.9 | 48.0 | 52.0 | 82.1 | 57.3 | 63.9 | 58.4 | 39.3 | 42.3 | 56.0 | 34.6 | 36.3 |
| CorInfoMax (Ozsoy et al., 2022) | 76.8 | 47.6 | 52.2 | 82.4 | 57.0 | 63.4 | 58.8 | 39.6 | 42.5 | 56.2 | 34.8 | 36.5 |
| SimCLR (Chen et al., 2020a) | 75.9 | 46.8 | 50.1 | 81.8 | 55.5 | 61.4 | 57.7 | 37.9 | 40.9 | 54.6 | 33.3 | 35.3 |
| SimCLR + GeSSL | 78.1 | 49.4 | 52.1 | 84.5 | 58.2 | 63.3 | 59.1 | 40.1 | 43.4 | 56.8 | 35.9 | 36.5 |
| MoCo (Chen et al., 2020b) | 77.1 | 46.8 | 52.5 | 82.5 | 57.4 | 64.0 | 58.9 | 39.3 | 42.5 | 55.8 | 34.4 | 36.5 |
| MoCo + GeSSL | 78.7 | 50.0 | **54.9** | **85.6** | 60.5 | 65.9 | 61.5 | **42.8** | **44.8** | **58.9** | 37.0 | 39.0 |
| BYOL (Grill et al., 2020) | 77.1 | 47.0 | 49.9 | 81.4 | 55.3 | 61.1 | 57.8 | 37.9 | 40.9 | 54.3 | 33.2 | 35.0 |
| BYOL + GeSSL | 78.7 | 49.8 | 53.2 | 84.9 | 59.0 | 64.8 | 60.7 | 40.9 | 44.0 | 57.5 | 36.2 | 38.1 |
| MAE (He et al., 2022) | 77.4 | 48.6 | 53.0 | 82.9 | 57.8 | 63.9 | 60.2 | 39.1 | 42.9 | 55.9 | 35.2 | 36.4 |
| MAE + GeSSL | 79.1 | **51.0** | 54.4 | 85.4 | **61.2** | 65.9 | 62.1 | 42.1 | 44.8 | 58.2 | **38.3** | 39.1 |
| SimSiam (Chen & He, 2021) | 77.3 | 48.5 | 52.5 | 82.4 | 57.0 | 63.7 | 59.3 | 39.2 | 42.1 | 56.0 | 34.4 | 36.7 |
| SimSiam + GeSSL | **79.3** | 50.5 | 54.1 | 85.0 | 59.4 | 65.8 | 62.0 | 41.5 | 44.3 | 58.4 | 37.5 | **39.6** |
| SwAV (Caron et al., 2020) | 75.5 | 46.5 | 49.6 | 82.6 | 56.1 | 62.7 | 58.6 | 38.4 | 41.3 | 55.2 | 33.8 | 35.9 |
| SwAV + GeSSL | 78.4 | 49.3 | 52.3 | 84.8 | 58.7 | 65.1 | 61.3 | 40.7 | 43.9 | 57.0 | 36.6 | 38.8 |
| VICRegL (Bardes et al., 2022) | 75.9 | 47.4 | 52.3 | 82.6 | 56.4 | 62.9 | 59.2 | 39.8 | 42.1 | 56.5 | 35.1 | 36.7 |
| VICRegL + GeSSL | 78.9 | 50.5 | 54.6 | 85.4 | 59.8 | **66.0** | **62.2** | 42.2 | 44.5 | 58.7 | 37.8 | 39.4 |

verify the generalization and improvement of GeSSL on recent SSL and transfer tasks, we also conduct comparative experiments with the methods that aim to improve the performance of SSL baselines, including methods with ResNet and ViT backbones. The results are provided in Appendix F.1 due to space limitations, demonstrating the advantages of GeSSL.

**Semi-supervised Learning.** We adopt the commonly used protocol (Zbontar et al., 2021) and create two balanced subsets by sampling 1% and 10% of the training set. We fine-tune the models for 50 epochs with learning rates of 0.05 and 1.0 for the classifier, 0.0001 and 0.01 for the backbone on the 1% and 10% subsets. Table 2 shows that the performance with GeSSL is superior to the SOTA methods, e.g., when only 1% labels are available, the improvement reaches more than 3%.

**Transfer Learning.** We use Faster R-CNN (Ren et al., 2015) for VOC detection and Mask R-CNN (He et al., 2017) for COCO detection and segmentation with the same C4-backbone (Wu et al., 2019). We train the Faster R-CNN on the VOC 07+12 set (16K images) and reduce the initial learning rate by 10 at 18K and 22K iterations, while training on the VOC 07 set (5K images) with fewer iterations. For Mask R-CNN, we train it on the COCO 2017 train split and report on the val split. See Appendix F.2 for details. Table 3 shows the great performance improvements achieved by GeSSL. After introducing GeSSL, the models achieve SOTA performance.

**Few-shot Learning.** We adopt the commonly used protocol (Jang et al., 2023) on miniImageNet, Omniglot, and CIFAR-FS. For the few-shot SSL task, we randomly select $N$ samples without class-level overlap for each task, and then apply 2-times data augmentation, obtaining a $N$-way 2-shot task with $N$ classes and $2N$ samples. We use the SGD optimizer, setting the momentum and weight decay values to 0.9 and $10^{-4}$ respectively. We evaluate the trained model's performance in some

Table 4: Few-shot learning accuracies ($\pm$ 95% confidence interval) on miniImageNet, Omniglot, and CIFAR-FS with C4. See Appendix E for the baselines' details, and Appendix F for full results.

| Method | Omniglot | | | *mini*ImageNet | | | CIFAR-FS | | |
|---|---|---|---|---|---|---|---|---|---|
| | (5,1) | (5,5) | (20,1) | (5,1) | (5,5) | (20,1) | (5,1) | (5,5) | (20,1) |
| *Unsupervised Few-shot Learning* | | | | | | | | | |
| CACTUs (Hsu et al., 2018) | 65.29 ± 0.21 | 86.25 ± 0.19 | 49.54 ± 0.21 | 39.32 ± 0.28 | 53.54 ± 0.27 | 31.99 ± 0.29 | 40.02 ± 0.23 | 58.16 ± 0.22 | 35.88 ± 0.25 |
| UMTRA (Khodadadeh et al., 2019) | 83.32 ± 0.37 | 94.23 ± 0.35 | 75.84 ± 0.34 | 39.23 ± 0.34 | 51.78 ± 0.32 | 30.27 ± 0.34 | 41.61 ± 0.40 | 60.55 ± 0.38 | 37.10 ± 0.39 |
| LASIUM (Khodadadeh et al., 2020) | 82.38 ± 0.36 | 95.11 ± 0.36 | 70.23 ± 0.36 | 42.12 ± 0.38 | 54.98 ± 0.37 | 34.26 ± 0.35 | 45.33 ± 0.32 | 62.65 ± 0.33 | 38.40 ± 0.33 |
| SVEBM (Kong et al., 2021) | 87.07 ± 0.28 | 94.13 ± 0.27 | 73.33 ± 0.28 | 44.74 ± 0.29 | 58.38 ± 0.28 | 39.71 ± 0.30 | 47.24 ± 0.25 | 63.10 ± 0.28 | 40.10 ± 0.28 |
| GMVAE (Lee et al., 2021) | 90.89 ± 0.32 | 96.05 ± 0.32 | 81.51 ± 0.33 | 42.28 ± 0.36 | 56.97 ± 0.38 | 39.83 ± 0.36 | 47.45 ± 0.36 | 63.20 ± 0.35 | 41.55 ± 0.35 |
| PsCo (Jang et al., 2023) | 96.18 ± 0.21 | 98.22 ± 0.23 | 89.32 ± 0.23 | 46.35 ± 0.24 | 63.05 ± 0.23 | 40.84 ± 0.27 | 51.77 ± 0.27 | 69.12 ± 0.26 | 45.08 ± 0.27 |
| *Self-supervised Learning* | | | | | | | | | |
| SimCLR (Chen et al., 2020a) | 90.83 ± 0.21 | 97.67 ± 0.21 | 81.67 ± 0.23 | 42.32 ± 0.38 | 51.10 ± 0.37 | 36.36 ± 0.36 | 49.44 ± 0.30 | 60.02 ± 0.29 | 39.29 ± 0.30 |
| SimCLR + GeSSL | 94.35 ± 0.31 | **98.41 ± 0.19** | 90.23 ± 0.24 | 46.51 ± 0.29 | 62.56 ± 0.29 | 39.56 ± 0.12 | **52.72 ± 0.15** | 67.52 ± 0.11 | 46.81 ± 0.14 |
| MoCo (Chen et al., 2020b) | 87.83 ± 0.20 | 95.52 ± 0.19 | 80.03 ± 0.21 | 40.56 ± 0.34 | 49.41 ± 0.37 | 36.52 ± 0.38 | 45.35 ± 0.31 | 58.11 ± 0.32 | 37.89 ± 0.32 |
| MoCo + GeSSL | 93.15 ± 0.15 | 97.84 ± 0.14 | 88.84 ± 0.13 | 47.23 ± 0.14 | 61.05 ± 0.12 | 40.75 ± 0.11 | 51.58 ± 0.09 | 66.25 ± 0.08 | 44.48 ± 0.11 |
| SwAV (Caron et al., 2020) | 91.28 ± 0.19 | 97.21 ± 0.20 | 82.02 ± 0.20 | 44.39 ± 0.36 | 54.91 ± 0.36 | 37.13 ± 0.37 | 49.39 ± 0.29 | 62.20 ± 0.30 | 40.19 ± 0.32 |
| SwAV + GeSSL | **96.18 ± 0.14** | 98.25 ± 0.18 | **91.62 ± 0.22** | **48.60 ± 0.15** | **63.56 ± 0.08** | **41.54 ± 0.23** | 52.33 ± 0.28 | **69.58 ± 0.25** | **47.56 ± 0.15** |

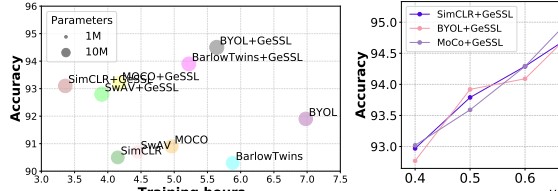

Figure 2: Model efficiency.  Figure 3: Ablation study of $\mu$.

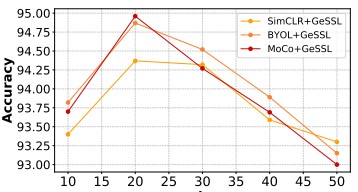

Figure 4: Ablation study of $k$.

unseen samples sampled from a new class. Table 4 shows the standard few-shot learning results of GeSSL compared with the baselines. Our framework still achieves remarkable performance improvement, demonstrating the superiority of GeSSL. See Appendix F.3 for more details.

## 5.2 ABLATION STUDY AND ANALYSIS

We conduct various ablation studies to evaluate GeSSL, including the model efficiency, effects of $\mathcal{L}_{disc}$, parameter sensitivity, the role of bi-level optimization, etc. More details, analyses, and additional experiments are provided in Appendix G and Appendix H.

**Model efficiency.** We evaluate the trade-off performance of baselines using GeSSL on STL-10 (Coates et al., 2011). Figure 2 shows that GeSSL achieves great performance and efficiency improvements with acceptable parameter size. Combining Appendix G.4, although GeSSL brings a larger memory footprint and parameter size costs, it is relatively negligible compared to the improvements.

**Effects of $\mathcal{L}_{disc}$.** To assess the impact of $\mathcal{L}_{disc}$, we visualize t-SNE feature clusters results before and after adding $\mathcal{L}_{disc}$ (see Appendix G.2 for details). Figure 8 shows that incorporating $\mathcal{L}_{disc}$ yields sharper class boundaries on multiple SSL baselines, demonstrating its effect on discriminability.

**Parameter Sensitivity** We evaluate the impact of $\mu$, $k$, and batch size $M$ (see Appendix G.3 for details). We search $\mu$ over $[0.3, 1.0]$ with a step of 0.05 and $k$ over $[10, 50]$ with 10, then refine in the best subranges. Figures 3-4 show that the optimal settings are $\mu = 0.7$ and $k = 30$.

**Evaluation of the bi-level optimization.** To evaluate the benefit of our bi-level optimization, we compare it against two alternatives: (i) optimizing inner and outer objectives jointly in a single stage; and (ii) training a separate $f'$ for each mini-batch. As shown in Figure 11, our bi-level optimization achieves SOTA performance. See Appendix G.4 for details and more experiments.

## 6 RELATED WORK

Self-supervised learning (SSL) learns representations from pretext tasks without labeled data. As summarized by (Jaiswal et al., 2020) and (Kang et al., 2023), SSL methods fall into two broad categories: discriminative and generative. Discriminative methods, such as SimCLR (Chen et al., 2020a) and BYOL (Grill et al., 2020), apply stochastic augmentations to generate two views of the same input and maximize their similarity in embedding space, thereby learning meaningful

representations. Generative methods, such as MAE (He et al., 2022) and VideoMAE (Tong et al., 2022), use an encoder–decoder structure: the input is partitioned into blocks, a subset is masked, and the remaining blocks are reconstructed in their original positions. Despite impressive empirical success, SSL faces several challenges (Jaiswal et al., 2020). Models often fail to generalize (i) under limited data (Krishnan et al., 2022) and (ii) in noisy, real-world environments (Goyal et al., 2021). Performance is also highly sensitive to alignment between pretext and downstream tasks, which can hinder transfer (see Section 5). As a result, the universality of SSL remains difficult to achieve. Prior studies (Oord et al., 2018; Hjelm et al., 2018; Mizrahi et al., 2024; Tian et al., 2020b; Oquab et al., 2023) primarily highlight empirical gains, but rarely ask what constitutes a good representation. In this work, we address this gap by formalizing "good representation" in terms of discriminability, generalizability, and transferability (see Appendices F–H for detailed analysis).

## 7 CONCLUSION

In this study, we explore the universality of SSL. We first unify SSL paradigms, i.e., discriminative and generative SSL, from the task perspective and propose the definition of SSL universality. It is a fundamental concept that involves discriminability, generalizability, and transferability. Then, we propose GeSSL to explicitly model universality into SSL through bi-level optimization, which introduces an auxiliary network to guide the model learn in the best direction. Extensive theoretical and empirical analyses demonstrate the effectiveness of GeSSL.

## ETHICS STATEMENT

All authors have carefully read and fully adhered to the ICLR Code of Ethics. The research presented in this paper was conducted in compliance with the principles of research integrity, fairness, and transparency. Our work does not involve human subjects, personally identifiable information, or other sensitive data, and it does not present foreseeable risks of harm, misuse, or ethical concerns. All experiments are carried out on publicly available datasets or synthetic data, and we have ensured proper documentation and reproducibility. The authors confirm that there are no conflicts of interest or violations of the ICLR Code of Ethics associated with this submission.

## REPRODUCIBILITY STATEMENT

We ensure reproducibility by detailing experimental settings, datasets, and hyperparameters in both Section 5 and Appendices C-G, and by providing full proofs of theoretical results in Appendix B. The source code is included in the supplementary materials to reproduce the proposed method.

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

## APPENDIX

The appendix is organized into several sections:

- Appendix A provides the list of notations.
- Appendix B contains the analyses and proofs of the presented definitions and theorems.
- Appendix C presents the implementation and architecture of our GeSSL.
- Appendix D provides details for all datasets used in the experiments.
- Appendix E provides details for the baselines mentioned in the main text.
- Appendix F showcases additional experiments, full results, and experimental details of the comparison experiments that were omitted in the main text due to space limitations.
- Appendix G provides the additional experiments and full details of the ablation studies that were omitted in the main text due to page limitations.
- Appendix H provides the discussion about the proposed methodology.

Note that before we illustrate the details and analysis, we provide a brief summary about all the experiments conducted in this paper, as shown in Table 5.

Table 5: Illustration of the experiments conducted in this work. Note that all experimental results are obtained after five rounds of experiments.

| Experiments | Location | Results |
|---|---|---|
| Experiments of unsupervised learning on six benchmark dataset | Section 5.1 and Appendix F.1 | Table 1, Table 6, Table 10, and Table 14 |
| Experiments of semi-supervised learning on on ImageNet with two settings | Section 5.1 | Table 2 and Table 15 |
| Experiment of transfer learning | Section 5.1 and Appendix F.2 | Table 3, Table 11, and Table 12 |
| Experiment of few-shot learning on standard and cross-domain scenarios | Section 5.1 and Appendix F.3 | Table 4 and Table 13 |
| Ablation study-Model efficiency | Section 5.2 and Appendix G.1 | Figure 2 and Table 29 |
| Ablation study-Effect of $\mathcal{L}_{disc}$ | Section 5.2 and Appendix G.2 | Figure 8 |
| Ablation study-Parameter sensitivity | Section 5.2 and Appendix G.3 | Figure 4, Figure 3, Figure 9, and Figure 10 |
| Ablation study-Evaluation of the bi-level optimization | Section 5.2 and Appendix G.4 | Figure 12 |
| Universality of existing SSL methods | Appendix F.4 | Figure 5 and Table 16 |
| Evaluation of generative SSL on three scenarios | Appendix F.5 | Figure 6, Table 17, Table 18, and Table 19 |
| Evaluation on more modalities | Appendix F.6 | Table 20 |

## A    LIST OF NOTATIONS

We list the definitions of all notations from the main text as follows:

☐ **Symbols of Data**

- $N$: the batch size.
- $m$: the number of mini-batches processed in parallel.
- $x_{i,l}^1, x_{i,l}^2$: two augmented views of the $i$-th sample in the $l$-th mini-batch.
- $x_{i,l}^{\text{anchor}}$: the anchor sample in pair $i$ of mini-batch $l$.
- $X_{tr,l}^{aug}$: the $l$-th augmented mini-batch.
- $X_{tr,l}^{\text{sup}}$: support set of mini-batch $l$, $\{x_{i,l}^1, x_{i,l}^{\text{anchor}}\}_{i=1}^n$.
- $X_{tr,l}^{\text{que}}$: query set of mini-batch $l$, $\{x_{i,l}^2, x_{i,l}^{\text{anchor}}\}_{i=1}^n$.
- $n$: number of sample pairs in a mini-batch (analogous to batch size).
- $P_t$: task distribution.

☐ **Symbols of Models and Losses**

- $f$: the SSL feature extractor (base encoder).
- $g$: the auxiliary network producing adaptive thresholds $a_i$.
- $f'$: the proxy model obtained by inner-loop adaptation.
- $\mathcal{L}_{ssl}$: SSL loss (contrastive loss in D-SSL, reconstruction loss in G-SSL).
- $\mathcal{L}_{disc}$: discriminative regularization loss in GeSSL.
- $\mu$: the weighting coefficient for $\mathcal{L}_{disc}$.

☐ **Symbols in the Discriminative Loss**

- $a_i$: the adaptive threshold predicted by $g$ for pair $i$.
- $d_j^i$: cosine distance between $f(x_{j,l}^1)$ and $f(x_{i,l}^{\text{anchor}})$.
- $d(\cdot, \cdot)$: cosine distance function.
- $\mathbb{1}_{\{\cdot\}}$: indicator function.
- $w(a_i)$: soft gating weight $\text{Sigmoid}(k(a_i - d_j^i))$.
- $k$: temperature/steepness parameter for the sigmoid gate.

☐ **Symbols of Optimization**

- $\alpha$: learning rate of the inner-loop update.
- $\beta$: learning rate of the outer-loop update.
- $\varsigma$: number of inner-loop update steps.
- $f'(f, g)$: notation emphasizing that $f'$ depends on both $f$ and $g$.

## B  PROOFS

Before giving the main theorem, we first provide the assumptions.

**Assumption B.1** *The following conditions are assumed to hold simultaneously:*

*(A1) IID Tasks: The training tasks $\tau_1, \ldots, \tau_N$ are sampled i.i.d. from the task distribution $\mathcal{T}$.*

*(A2) Bounded Losses: For any parameter $\theta$ and task $\tau$, all loss components satisfy $0 \le \mathcal{L}_{\text{sup}}(\theta; \tau), \mathcal{L}_{ssl}(\theta; S_\tau), \mathcal{L}_{disc}(\theta; S_\tau), \mathcal{L}_{query}(\theta'; Q_\tau) \le \mathcal{L}_{\max}$.*

*(A3) Gradient Lipschitz Continuity: There exists a constant $G > 0$ such that for all $\theta, \theta'$ and any $\tau$, with $\left\| \nabla_\theta \big( \mathcal{L}_{ssl}(\theta; S_\tau) + \mathcal{L}_{disc}(\theta; S_\tau) \big) - \nabla_\theta \big( \mathcal{L}_{ssl}(\theta'; S_\tau) + \mathcal{L}_{disc}(\theta'; S_\tau) \big) \right\| \le G \|\theta - \theta'\|$.*

*(A4) Inner-Loop Reduction: There exists $\Delta > 0$ such that for all $\theta$ and $\tau$, have $\mathcal{L}_{ssl}\big(A(\theta, S_\tau); S_\tau\big) + \mathcal{L}_{disc}\big(A(\theta, S_\tau); S_\tau\big) \le \mathcal{L}_{ssl}(\theta; S_\tau) + \mathcal{L}_{disc}(\theta; S_\tau) - \Delta$.*

*(A5) Fast Adaptation: There exists $C > 0$ and a small sample size $m \ll |Q_\tau|$ such that for all $\theta$ and $\tau$, have $\mathcal{L}_{\text{sup}}\big(A(\theta, S_\tau); \tau\big) \le \frac{C}{\sqrt{m}} \Big( \mathcal{L}_{ssl}(\theta; S_\tau) + \mathcal{L}_{disc}(\theta; S_\tau) \Big)$.*

These assumptions are recognized as mild conditions in both theory and practice. Specifically, (A1) assumes that training and test tasks are drawn i.i.d. to ensure that strategies learned on the training set generalize to new tasks, an assumption ubiquitous in generalization analyses (Baxter, 2000; Finn et al., 2017). In our setting, tasks (mini-batches) are constructed by randomly sampling from the data distribution and applying augmentations, which amounts to independently drawing from an underlying class distribution and sample distribution joint space and thus naturally satisfies the i.i.d. condition. (A2) imposes bounded losses, a requirement extensively validated in practice (Boucheron et al., 2013). When deriving generalization bounds, we typically invoke concentration inequalities such as Hoeffding's or McDiarmid's to obtain exponential tail bounds; also, any residual unboundedness can be handled by simple clipping or adding a small constant without affecting empirical performance (Krizhevsky et al., 2012; Miyato et al., 2018). These conditions make A2 readily satisfied in real-world settings. Then, the gradient Lipschitz continuity in (A3), which is equivalent to a bounded Hessian, is a standard condition in non-convex optimization analysis. Techniques like BatchNorm, weight decay, or spectral normalization in deep networks effectively enforce this smoothness, ensuring stable and convergent updates (Bottou et al., 2018). (A4) and (A5) require that after multiple steps of optimization, the model yields a strictly lower loss, aligning with gradient-based optimization theory (Vapnik, 1998); indeed, many prior works demonstrate that even a few steps achieve substantial loss reduction (Rajeswaran et al., 2019; Wang et al., 2024). Therefore, under these mild assumptions, we analyze the optimization objective to further ensure its reliability.

Review the notations and settings: Given a task distribution: $\mathcal{T}$, each task $\tau \sim \mathcal{T}$ contains a support set $S_\tau$ and a query set $Q_\tau$. For any parameter vector $\theta$ and task $\tau$, we denote the supervised loss on the unseen query set $Q_\tau$ by $\mathcal{L}_{\sup}(\theta; \tau)$, the self-supervised and discriminative losses on the support set $S_\tau$ by $\mathcal{L}_{ssl}(\theta; S_\tau)$ and $\mathcal{L}_{disc}(\theta; S_\tau)$, respectively. Let $\theta' = A(\theta, S_\tau)$ be the adapted parameters after applying the adaptation operator $A$ to $\theta$ using $S_\tau$, the resulting query loss by $\mathcal{L}_{\text{query}}(\theta'; Q_\tau)$. The training tasks are $\{\tau_i\}_{i=1}^N$, independent and identically distributed (A1); all losses are truncated to $[0, \mathcal{L}_{max}]$ (A2). By jointly minimizing these four losses, we aim to learn representations that are both broadly transferable across tasks and rapidly fine-tunable to new tasks, while ensuring robust generalization performance. Next, we provide a detailed proof.

In the bi-level training stage, $\theta^*$ is the parameter obtained by minimizing the following formula on $N$ training tasks, where the empirical risk can be expressed as:

$$R_N(\theta) \equiv \frac{1}{N} \sum_{i=1}^N \Big[ \mathcal{L}_{ssl}(\theta; S_{\tau_i}) + \mathcal{L}_{disc}(\theta; S_{\tau_i}) + \mathcal{L}_{query}\big(A(\theta, S_{\tau_i}); Q_{\tau_i}\big) \Big]. \tag{7}$$

For simplicity, we denote $X_i(\theta) = \mathcal{L}_{ssl}(\theta; S_{\tau_i}) + \mathcal{L}_{disc}(\theta; S_{\tau_i}) + \mathcal{L}_{query}\big(A(\theta, S_{\tau_i}); Q_{\tau_i}\big)$, then we get $R_N(\theta) = \frac{1}{N} \sum_{i=1}^N X_i(\theta)$. To decompose the risk, the expected supervision (query) loss of the new task we want to prove is:

$$\mathcal{R} = \mathbb{E}_{\tau \sim \mathcal{T}}\big[ \mathcal{L}_{\sup}(A(\theta^*, S_\tau); \tau) \big] = \mathbb{E}_\tau \big[ \mathcal{L}_{query}(\theta^*_{\text{test}}; Q_\tau) \big], \tag{8}$$

where $\theta^*_{\text{test}} = A(\theta^*, S_\tau)$. Adding and subtracting $R_N(\theta^*)$ yields:

$$\mathcal{R} = R_N(\theta^*) + \underbrace{\big( \mathbb{E}_\tau[\mathcal{L}_{query}(\theta^*_{\text{test}}; Q_\tau)] - R_N(\theta^*) \big)}_{(*)}. \tag{9}$$

Then, we split $(*)$ into two parts:

$$(*) = \Big( \mathbb{E}_\tau[\mathcal{L}_{query}(\theta^*_{\text{test}})] - \mathbb{E}_\tau[X(\theta^*)] \Big) + \Big( \mathbb{E}_\tau[X(\theta^*)] - \frac{1}{N}\sum_{i=1}^N X_i(\theta^*) \Big)$$
$$\equiv (A) + (B). \tag{10}$$

where A denotes adaptation error, measures the difference between the fine-tuned and initial parameters on the same task; and B refers to generalization error, measures the deviation of estimating the overall expectation using a limited $N$ number of training tasks.

Next, we leverage the conditions in Assumption B.1 to analyze the adaptive error bound A. Assume that A4 guarantees: for any $\theta, \tau$, we have:

$$\mathcal{L}_{ssl}(A(\theta, S_\tau); S_\tau) + \mathcal{L}_{disc}(A(\theta, S_\tau); S_\tau) \leq \mathcal{L}_{ssl}(\theta; S_\tau) + \mathcal{L}_{disc}(\theta; S_\tau) - \Delta. \tag{11}$$

Therefore, let $F(\theta; S_\tau) = \mathcal{L}_{ssl}(\theta; S_\tau) + \mathcal{L}_{disc}(\theta; S_\tau)$, we have $F\big(A(\theta, S_\tau); S_\tau\big) \leq F(\theta; S_\tau) - \Delta$. Take single-step gradient descent as an example, the adaptation operator $\theta' = \theta - \eta \widehat{\nabla} F(\theta; S_\tau)$, where $\widehat{\nabla} F(\theta; S_\tau)$ represents the empirical gradient calculated on the support set $S_\tau$, with a step size of $\eta > 0$. By the triangle inequality and the gradient Lipschitz continuity in (A3), we have:

$$\begin{aligned}
\|\theta' - \theta\| &= \eta \left\| \widehat{\nabla} F(\theta; S_\tau) \right\| \\
&\leq \eta \Big( \|\nabla F(\theta; S_\tau)\| + \left\| \widehat{\nabla} F(\theta; S_\tau) - \nabla F(\theta; S_\tau) \right\| \Big).
\end{aligned} \tag{12}$$

Using the concentration inequality between empirical gradient and true gradient, i.e., Hoeffding's generalization of vector gradient, when the support set size is $m$, we have:

$$\left\| \widehat{\nabla} F(\theta; S_\tau) - \nabla F(\theta; S_\tau) \right\| \leq O\Big( \tfrac{G}{\sqrt{m}} \Big). \tag{13}$$

Thus we have $\|\theta' - \theta\| \leq \eta \Big( \|\nabla F(\theta; S_\tau)\| + O\big(\tfrac{G}{\sqrt{m}}\big) \Big)$. Assume that the query loss for parameters is also $L_{\sup}$-Lipschitz, that is $\big| \mathcal{L}_{query}(\theta'; Q_\tau) - \mathcal{L}_{query}(\theta; Q_\tau) \big| \leq \mathcal{L}_{\sup} \|\theta' - \theta\|$. Therefore, $\mathcal{L}_{query}(\theta'; Q_\tau) \leq \mathcal{L}_{query}(\theta; Q_\tau) + \mathcal{L}_{\sup} \|\theta' - \theta\|$. If we have made the initial query loss of $\theta$ close to zero on all tasks in the bi-level training (or can be regarded as a constant term and merged into the big $O$), then take $\mathcal{L}_{query}(\theta; Q_\tau) \approx 0$. Combine the above formula and substitute it into the bound of $\|\theta' - \theta\|$, we have:

$$\begin{aligned}
\mathcal{L}_{query}(\theta'; Q_\tau) &\leq 0 + \mathcal{L}_{\sup} \eta \Big( \|\nabla F(\theta; S_\tau)\| + O\big(\tfrac{G}{\sqrt{m}}\big) \Big) \\
&= \mathcal{L}_{\sup} \eta \|\nabla F(\theta; S_\tau)\| + O\Big( \tfrac{\eta}{\sqrt{m}} \Big).
\end{aligned} \tag{14}$$

By the smoothness and convexity of self-supervised and discriminant loss, it can be established that $\|\nabla F(\theta; S_\tau)\| = O\big( \sqrt{F(\theta; S_\tau)} \big)$. For example, it is exactly true in the case of quadratic convexity, or $\|\nabla F\|^2 \leq 2L\,F$ in the case of general smooth convexity, substituting in $\mathcal{L}_{query}(\theta'; Q_\tau) \leq \mathcal{L}_{\sup} \eta\, O\big( \sqrt{F(\theta; S_\tau)} \big) + O\Big( \tfrac{\eta}{\sqrt{m}} \Big)$. Take the gradient step size $\eta = \Theta(1/\sqrt{m})$, then the two terms are of the same order, and $\mathcal{L}_{\sup} \eta \sqrt{F} = O\Big( \tfrac{1}{\sqrt{m}} \sqrt{F(\theta; S_\tau)} \Big)$. Consider that Assumption A5 guarantees: for any $\theta, \tau$, have:

$$\mathcal{L}_{\sup}\big( A(\theta, S_\tau); \tau \big) = \mathcal{L}_{query}\big( A(\theta, S_\tau); Q_\tau \big) \leq \frac{C}{\sqrt{m}} F(\theta; S_\tau). \tag{15}$$

The constant $C$ combines factors such as $\mathcal{L}_{\sup}$, asymptotic implicit constants, and possible upper bounds $\sqrt{F} \leq F$ (when $F \leq 1$). Substituting the above steps, we get:

$$\mathcal{L}_{query}(\theta^*_{\text{test}}; Q_\tau) \leq \frac{C}{\sqrt{m}} F(\theta^*; S_\tau) = \frac{C}{\sqrt{m}} \Big( \mathcal{L}_{ssl}(\theta^*; S_\tau) + \mathcal{L}_{disc}(\theta^*; S_\tau) \Big). \tag{16}$$

The $i$-th item in the training phase $R_N(\theta^*)$ contains the sum of these two items, so we can write:

$$\text{(A)} = \mathbb{E}_\tau \big[ \mathcal{L}_{query}(\theta^*_{\text{test}}) \big] - \mathbb{E}_\tau \big[ X(\theta^*) \big] \leq \frac{C}{\sqrt{m}} \mathbb{E}_\tau \big[ F(\theta^*; S_\tau) \big] - \mathbb{E}_\tau \big[ X(\theta^*) \big]. \tag{17}$$

Note that $X(\theta^*) = F(\theta^*; S_\tau) + \mathcal{L}_{query}(A(\theta^*, S_\tau))$, then we get $\text{(A)} \leq \Big( \tfrac{C}{\sqrt{m}} - 1 \Big) \mathbb{E}_\tau [F(\theta^*; S_\tau)] - \mathbb{E}_\tau \big[ \mathcal{L}_{query}(\theta^*_{\text{test}}) \big]$. In common settings, $m$ is chosen so that $\tfrac{C}{\sqrt{m}} \leq 1$, thus $\text{(A)} \leq 0$ (or merged with the constant term into big $O$). Overall, we can let $\text{(A)} = \mathcal{O}\Big( \tfrac{C}{\sqrt{m}} \Big)$.

Next, we discuss the generalization error bound (B) via Hoeffding. Firstly, consider that $X_i(\theta^*) \in [0, 3\mathcal{L}_{max}]$, $\mu \equiv \mathbb{E}_\tau[X(\theta^*)]$, by Hoeffding inequality, we have:

$$\Pr\Big( \big| \tfrac{1}{N} \sum_{i=1}^N X_i(\theta^*) - \mu \big| \geq \epsilon \Big) \leq 2 \exp\Big( -\tfrac{2N\epsilon^2}{(3\mathcal{L}_{max})^2} \Big). \tag{18}$$

Take the right side as $\delta$, and solve $\epsilon = 3\mathcal{L}_{max} \sqrt{\tfrac{\ln(2/\delta)}{2N}} = \mathcal{O}\Big( \sqrt{\tfrac{1}{N} \ln \tfrac{1}{\delta}} \Big)$, with probability at least $1 - \delta$, we have:

$$\text{(B)} = \Big| \tfrac{1}{N} \sum X_i(\theta^*) - \mu \Big| \leq 3\mathcal{L}_{max} \sqrt{\frac{\ln(2/\delta)}{2N}}. \tag{19}$$

Merge each item back into the risk decomposition formula:

$$\mathcal{R} = R_N(\theta^*) + (A) + (B) \leq R_N(\theta^*) + \mathcal{O}\left(\frac{C}{\sqrt{m}}\right) + 3\mathcal{L}_{max}\sqrt{\frac{\ln(2/\delta)}{2N}}. \tag{20}$$

Remove the low-order constants and absorb $\frac{C}{\sqrt{m}}$ into the big $O$, and we get the required conclusion:

$$\mathbb{E}_{\tau_{\text{test}}}\left[\mathcal{L}_{\sup}(\theta^*_{\text{test}}; \tau_{\text{test}})\right] \leq \frac{1}{N}\sum_{i=1}^{N}\left[\mathcal{L}_{ssl}(\theta', S_{\tau_i}) + \mathcal{L}_{disc}(\theta', S_{\tau_i}) + \mathcal{L}_{\text{query}}(\theta', Q_{\tau_i})\right] + \mathcal{O}(\sqrt{\frac{1}{N}\ln\frac{1}{\delta}}) \tag{21}$$

The proof is complete.

## C   IMPLEMENTATION DETAILS

We use C4-backbone, ResNet-18, ResNet-50, and viT backbones as our encoders for a fair comparison with different methods. The convolutional layers are followed by batch normalization, ReLU nonlinearity, and max pooling (strided convolution) respectively. The last layer is fed into a MLP for $\mathcal{L}_{disc}$. These architectures are pre-trained and kept fixed during training. We optimize our model with a Stochastic Gradient Descent (SGD) optimizer, setting the momentum and weight decay values to $0.9$ and $10^{-4}$ respectively. The specific adjustments of the experimental settings corresponding to different experiments are illustrated in Section 5.1 of the main text. All the experiments are apples-to-apples comparisons and performed on NVIDIA RTX 4090 GPUs. We build tasks based on images with a batch size of $B = 16$. For data augmentation, we use the same data augmentation scheme as SimCLR to augment each image in the batch 5 times. In simple terms, we draw a random patch ($224 \times 224$) from the original image, and then apply a random augmentation sequence composed of random horizontal flip, cropping, color jitter, etc.

**How to integrate GeSSL in the ViT setting (how we model $f$, $g$, and $f'$).** For $f$, maps an input image $x$ to a representation vector $z = f(x)$. On ViT, $f$ corresponds to the full Transformer encoder (i.e., patch embedding, positional embedding, and Transformer blocks). The LayerNorm output of the CLS token is used as the fixed-length global representation $z = f(x)$; before feeding $z$ into downstream distance or projection modules we apply LayerNorm followed by $L_2$ normalization. For $g$, the projection head. For each anchor we form an input vector by concatenating the per-dimension mean and variance computed over the differential set; this vector is fed into a small MLP consisting of a linear dimensionality reduction, LayerNorm, and GELU, which outputs a scalar threshold $a_i$. For $f'$, to avoid costly differentiable second-order optimization, we instantiate $f'$ and $g'$ as momentum copies of $f$ and $g$. Their parameters are updated as $\theta' \leftarrow \tau\theta' + (1-\tau)\theta$ and they do not participate in backpropagation, thereby providing a stable one-step proxy for query-set evaluation and pseudo-label generation.

**Experimental Settings.** We evaluate GeSSL under four major settings: unsupervised learning, semi-supervised learning, transfer learning, and few-shot learning. For unsupervised learning, we follow the standard linear-evaluation protocol of SimCLR: the pretrained feature extractor is frozen and a linear classifier is trained on top using Adam with momentum 0.8, weight decay $10^{-4}$, a batch size of 128, and a learning rate decaying from $5 \times 10^{-2}$ to $5 \times 10^{-6}$ over 500 epochs. ResNet-18 is used for small-scale datasets (CIFAR-10/100, STL-10, Tiny ImageNet), ResNet-50 and ViT are used for ImageNet-100 and ImageNet. For semi-supervised learning, following the protocol of Barlow Twins, we construct two balanced subsets by sampling 1% and 10% of the training data, and fine-tune the models for 50 epochs using learning rates of 0.05/0.0001 (classifier/backbone) for the 1% setup and 1.0/0.01 for the 10% setup. For transfer learning, we adopt standard object detection and segmentation pipelines: Faster R-CNN is trained on PASCAL VOC 07+12 and VOC 07 with scheduled learning-rate drops at 18k and 22k iterations, while Mask R-CNN is trained on COCO 2017 using the same C4 backbone. For few-shot learning, following the widely used protocol, we evaluate on miniImageNet, Omniglot, and CIFAR-FS; each task samples $N$ instances from $N$ classes without label overlap, applies two data augmentations to form an $N$-way 2-shot episode, and is optimized using SGD with momentum 0.9 and weight decay $10^{-4}$, with evaluation performed on unseen samples from novel classes. These unified settings allow us to comprehensively assess the representation quality and generalization ability brought by GeSSL. More details are provided in the corresponding experimental section.

## D BENCHMARK DATASETS

In this section, we briefly introduce all datasets used in our experiments. In summary, the benchmark datasets can be divided into four categories: (i) for unsupervised learning, we evaluate GeSSL on six benchmark datasets, including CIFAR-10 (Krizhevsky et al., 2009), CIFAR-100 (Krizhevsky et al., 2009), STL-10 (Coates et al., 2011), Tiny ImageNet (Le & Yang, 2015), ImageNet-100 (Tian et al., 2020a) and ImageNet (Deng et al., 2009a); (ii) for semi-supervised learning, we evaluate GeSSL on ImageNet (Deng et al., 2009a); (iii) for transfer learning, we select three scenarios: instance segmentation (PASCAL VOC (Everingham et al., 2010)) and object detection (COCO (Lin et al., 2014), general transfer learning (CIFAR10 (Krizhevsky et al., 2009), Flower102 (Nilsback & Zisserman, 2008), Food101 (Bossard et al., 2014), and Aircraft (Maji et al., 2013)), and video tracking tasks (UniTrack); (iv) for few-shot learning, we select nine benchmarks for evaluation, including Omniglot (Lake et al., 2019), miniImageNet (Vinyals et al., 2016a), CIFAR-FS (Bertinetto et al., 2018), CUB (Welinder et al., 2010), Cars (Krause et al., 2013), Places (Zhou et al., 2017), CropDiseases (Mohanty et al., 2016), ISIC (Codella et al., 2018), and ChestX (Wang et al., 2017). The composition of the data set is as follows:

- CIFAR-10 (Krizhevsky et al., 2009) is a prevalent image classification benchmark comprising 10 classes, each containing 5000 $32\times32$ resolution images.

- CIFAR-100 (Krizhevsky et al., 2009), another widely used image classification benchmark, consists of 100 classes, each containing 5000 images at a resolution of $32\times32$.

- STL-10 (Coates et al., 2011) encompasses 10 classes with 500 training and 800 test images per class at a high resolution of 96x96 pixels. It also includes 100,000 unlabeled images for unsupervised learning.

- Tiny ImageNet (Le & Yang, 2015), a subset of ImageNet by Stanford University, comprises 200 classes, each with 500 training, 50 verification, and 50 test images.

- ImageNet-100 (Tian et al., 2020a), a subset of ImageNet, includes 100 classes, each containing 1000 images.

- ImageNet (Deng et al., 2009a), organized by the WordNet hierarchy, is a renowned dataset featuring 1.3 million training and 50,000 test images across 1000+ classes.

- PASCAL VOC dataset (Everingham et al., 2010), known for object classification, detection, and segmentation, encompasses 20 classes with a total of 11,530 images split between VOC 07 and VOC 12.

- COCO dataset (Lin et al., 2014), primarily used for object detection and segmentation, comprises 91 classes, 328,000 samples, and 2,500,000 labels.

- Flower102 (Nilsback & Zisserman, 2008) contains 102 flower categories, totaling 8,189 images. Each class has between 40 and 258 images of varying original resolution, typically resized or center-cropped to $224\times224$ pixels for model input.

- Food101 (Bossard et al., 2014) comprises 101 food categories with 1,000 images each (101,000 total). The split is 750 images per class for training and 250 for testing.

- Aircraft (Maji et al., 2013) covers 100 aircraft model variants with approximately 100–200 images per class (over 10,000 images total). Original image resolutions vary; standard practice is to crop or resize them to $224\times224$ pixels for downstream tasks.

- miniImageNet (Vinyals et al., 2016a) is a few-shot learning dataset that consists of 100 classes, each with 600 images. The images have a resolution of 84x84 pixels.

- Omniglot (Lake et al., 2019) is another dataset for few-shot learning, which comprises 1623 different handwritten characters from 50 different alphabets. The 1623 characters were drawn by 20 different people online using Amazon's Mechanical Turk. Each image is paired with stroke data $[x, y, t]$ sequences and time (t) coordinates (ms).

- CIFAR-FS (Bertinetto et al., 2018) is also a dataset for few-shot learning research, derived from the CIFAR-100 dataset. It consists of 100 classes, each with a small training set of 500 images and a test set of 100 images. The images have a resolution of $32 \times 32$ pixels.

- CUB (Welinder et al., 2010) is a dataset of 200 bird species, with 11,788 images in total and about 60 images per species. Each image has detailed annotations, including subcategory labels, 15 part locations, 312 binary attributes, and a bounding box.

- Cars (Krause et al., 2013) is a dataset of 196 car models, with 16,185 images in total and about 80 images per model. Each image has a subcategory label, indicating the manufacturer, model, and year of the car.

- Places (Zhou et al., 2017) is a dataset of 205 scene categories, with 2.5 million images in total and about 12,000 images per category. The scene categories are defined by their functions, representing the entry-level of the environment.

- CropDiseases (Mohanty et al., 2016) is a dataset of 24,881 images of crop pests and diseases, with 22 categories, each including different pests and diseases of 4 crops (cashew, cassava, maize, and tomato).

- ISIC (Codella et al., 2018) is a dataset of over 13,000 dermoscopic images of skin lesions, which is the largest publicly available quality-controlled archive of dermoscopic images. The dataset includes 8 common types of skin lesions, such as melanoma, basal cell carcinoma, squamous cell carcinoma, etc.

- ChestX (Wang et al., 2017) is a dataset of 112,120 chest X-ray images, with 14 common types of chest diseases, such as pneumonia, emphysema, fibrosis, etc. The dataset was collected from 30,805 unique patients (from 1992 to 2015) of the National Institutes of Health Clinical Center (NIHCC).

# E BASELINES

In this section, we briefly introduce all baselines used in the experiments for comparison. We select eighteen representative self-supervised methods as baselines, including discriminative SSL (D-SSL) and generative SSL (G-SSL) methods. These methods cover almost all the classic and SOTA self-supervised methods, including:

- SimCLR (Chen et al., 2020a) learns visual representations by contrastive learning of augmented image pairs. It uses a neural network to maximize the similarity of positive pairs and minimize the similarity of negative pairs.

- MoCo v2 (Chen et al., 2020b) improves MoCo (Chen et al., 2020b), another contrastive learning method for visual representation learning. MoCo v2 introduces a momentum encoder, a memory bank, and a shuffling BN layer to handle limited batch size and noisy negatives. MoCo v2 also adopts SimCLR's data augmentation and loss function to boost the performance.

- BYOL (Grill et al., 2020) does not need negative pairs or a large batch size. It uses two neural networks, an online network and a target network, that learn from each other. The online network predicts the target network's representation of an augmented image, while the target network is updated by a slow-moving average of the online network.

- SimSiam (Chen & He, 2021) simplifies BYOL by removing the momentum encoder and the prediction MLP. It consists of two Siamese networks that map an input image to a feature vector, and a small MLP head that projects the feature vector to the contrastive learning space. SimSiam applies a stop-gradient operation to one of the MLP outputs, and uses a negative cosine similarity loss to maximize the similarity between the two outputs.

- Barlow Twins (Zbontar et al., 2021) learns representations by enforcing that the cross-correlation matrix between the outputs of two identical networks fed with different augmentations of the same image is close to the identity matrix. This encourages the networks to produce similar representations for the positive pair, while reducing the redundancy between the representation dimensions.

- DeepCluster (Caron et al., 2018) is a clustering-based method for self-supervised learning. It iteratively groups the features produced by a convolutional network into clusters, and uses the cluster assignments as pseudo-labels to update the network parameters by supervised learning. DeepCluster can discover meaningful clusters that are discriminative and invariant to transformations, and can learn competitive features for various downstream tasks.

- SwAV (Caron et al., 2020) uses online swapping of cluster assignments between multiple views of the same image to learn visual features. SwAV first computes prototypes (cluster

centers) from a large set of features, and then assigns each feature to the nearest prototype. The assignments are then swapped across the views, and the network is trained to predict the swapped assignments.

- DINO (Caron et al., 2021) learns visual features by using a teacher-student architecture and a distillation loss. The teacher network is an exponential moving average of the student network, and the distillation loss makes the student features similar to the teacher features. DINO also applies a centering and sharpening operation to the teacher features, which prevents feature collapse and increases feature diversity.

- MAE (He et al., 2022) randomly masks a high proportion of image patches and trains the model to reconstruct the missing pixels. By forcing the encoder to infer global structure from partial inputs, MAE learns rich, semantic representations that transfer well to downstream tasks with minimal fine-tuning.

- SeqCLR (Aberdam et al., 2021) extends contrastive frameworks to video by treating successive frames as positive pairs and distant frames (or different clips) as negatives. By maximizing agreement between temporally adjacent representations, SeqCLR learns spatiotemporal features that are effective for downstream video-based tasks.

- W-MSE (Ermolov et al., 2021) learns features by using a weighted mean squared error (MSE) loss, which assigns higher weights to the informative and less noisy features, and lower weights to the less informative and more noisy features.

- RELIC v2 (Tomasev et al., 2022) learns visual features by predicting relative location of image patches. RELIC v2 divides an image into a grid of patches, and randomly selects a query and a target patch. The network is trained to predict the relative location of the target patch with respect to the query patch, using a cross-entropy loss.

- LMCL (Chen et al., 2021) learns visual features by using a large margin cosine loss (LMCL). LMCL is a metric learning loss that makes the features of the same class closer and the features of different classes farther in the cosine space.

- ReSSL (Zheng et al., 2021) learns visual features by using a reconstruction loss and a contrastive loss. ReSSL applies random cropping and resizing to generate two views of the same image, and then feeds them to a reconstruction network and a contrastive network. The reconstruction network is trained to reconstruct the original image from the cropped view, while the contrastive network is trained to maximize the similarity between the features of the two views.

- SSL-HSIC (Li et al., 2021) learns visual features by using a Hilbert-Schmidt independence criterion (HSIC) loss. HSIC is a measure of statistical dependence between two random variables, and can be used to align the features of different views of the same image.

- CorInfoMax (Ozsoy et al., 2022) learns visual features by maximizing the correlation and mutual information between the features of augmented image pairs and the image labels. CorInfoMax aims to learn features that are both discriminative and consistent, and outperform previous methods on image classification and segmentation tasks.

- MEC (Liu et al., 2022) is a clustering algorithm that can handle large-scale data with limited memory by using a memory-efficient clustering (MEC) loss. MEC first samples a subset of features, and then performs k-means clustering on the subset. The cluster assignments are then propagated to the rest of the features by a nearest neighbor search.

- VICRegL (Bardes et al., 2022) learns visual features by using a variance-invariance-covariance regularization loss (VICRegL).

- VoCo (Wu et al., 2024) a simpleyet-effective Volume Contrast (VoCo) framework to leverage the contextual position priors for pre-training.

In addition, for the few-shot learning scenario, we choose six advanced unsupervised few-shot learning methods as comparison baselines.

- CACTUs (Hsu et al., 2018) uses clustering and augmentation to create pseudo-labels for unlabeled data. It then trains a classifier on the labeled data and fine-tunes it on a few labeled examples from the target task.

Table 6: The classification accuracies ($\pm$ 95% confidence interval) of a linear classifier (linear) and a 5-nearest neighbors classifier (5-nn) with a ResNet-18 as the feature extractor. The comparison baselines cover almost all types of methods mentioned in Section 6. The "-" denotes that the results are not reported. More details of the baselines are provided in Appendix E.

| Method | CIFAR-10 | | CIFAR-100 | | STL-10 | | Tiny ImageNet | |
|---|---|---|---|---|---|---|---|---|
| | linear | 5 $-$ nn | linear | 5 $-$ nn | linear | 5 $-$ nn | linear | 5 $-$ nn |
| SimCLR (Chen et al., 2020a) | 91.80 $\pm$ 0.15 | 88.42 $\pm$ 0.15 | 66.83 $\pm$ 0.27 | 56.56 $\pm$ 0.18 | 90.51 $\pm$ 0.14 | 85.68 $\pm$ 0.10 | 48.84 $\pm$ 0.15 | 32.86 $\pm$ 0.25 |
| MoCo (Chen et al., 2020b) | 91.69 $\pm$ 0.12 | 88.66 $\pm$ 0.14 | 67.02 $\pm$ 0.16 | 56.29 $\pm$ 0.25 | 90.64 $\pm$ 0.28 | 88.01 $\pm$ 0.19 | 50.92 $\pm$ 0.22 | 35.55 $\pm$ 0.16 |
| BYOL (Grill et al., 2020) | 91.93 $\pm$ 0.22 | 89.45 $\pm$ 0.22 | 66.60 $\pm$ 0.16 | 56.82 $\pm$ 0.17 | 91.99 $\pm$ 0.13 | 88.64 $\pm$ 0.20 | 51.00 $\pm$ 0.12 | 36.24 $\pm$ 0.28 |
| SimSiam (Chen & He, 2021) | 91.71 $\pm$ 0.27 | 88.65 $\pm$ 0.17 | 67.22 $\pm$ 0.26 | 56.36 $\pm$ 0.19 | 91.01 $\pm$ 0.19 | 88.16 $\pm$ 0.19 | 51.14 $\pm$ 0.20 | 35.67 $\pm$ 0.16 |
| Barlow Twins (Zbontar et al., 2021) | 90.88 $\pm$ 0.19 | 89.68 $\pm$ 0.21 | 66.13 $\pm$ 0.10 | 56.70 $\pm$ 0.25 | 90.38 $\pm$ 0.13 | 87.13 $\pm$ 0.23 | 49.78 $\pm$ 0.26 | 34.18 $\pm$ 0.18 |
| SwAV (Caron et al., 2020) | 91.03 $\pm$ 0.19 | 89.52 $\pm$ 0.24 | 66.56 $\pm$ 0.17 | 57.01 $\pm$ 0.25 | 90.72 $\pm$ 0.29 | 86.24 $\pm$ 0.26 | 52.02 $\pm$ 0.26 | 37.40 $\pm$ 0.11 |
| DINO (Caron et al., 2021) | 91.83 $\pm$ 0.25 | 90.15 $\pm$ 0.33 | 67.15 $\pm$ 0.21 | 56.48 $\pm$ 0.19 | 91.03 $\pm$ 0.12 | 86.15 $\pm$ 0.25 | 51.13 $\pm$ 0.30 | 37.86 $\pm$ 0.19 |
| W-MSE (Ermolov et al., 2021) | 91.99 $\pm$ 0.12 | 89.87 $\pm$ 0.25 | 67.64 $\pm$ 0.16 | 56.45 $\pm$ 0.26 | 91.75 $\pm$ 0.23 | 88.59 $\pm$ 0.15 | 49.22 $\pm$ 0.16 | 35.44 $\pm$ 0.10 |
| RELIC v2 (Tomasev et al., 2022) | 91.92 $\pm$ 0.14 | 90.02 $\pm$ 0.22 | 67.66 $\pm$ 0.20 | 57.03 $\pm$ 0.18 | 91.10 $\pm$ 0.23 | 88.66 $\pm$ 0.12 | 49.33 $\pm$ 0.13 | 35.52 $\pm$ 0.22 |
| LMCL (Chen et al., 2021) | 91.91 $\pm$ 0.25 | 88.52 $\pm$ 0.29 | 67.01 $\pm$ 0.18 | 56.86 $\pm$ 0.14 | 90.87 $\pm$ 0.18 | 85.91 $\pm$ 0.25 | 49.24 $\pm$ 0.18 | 32.88 $\pm$ 0.13 |
| ReSSL (Zheng et al., 2021) | 90.20 $\pm$ 0.16 | 88.26 $\pm$ 0.18 | 66.79 $\pm$ 0.12 | 53.72 $\pm$ 0.28 | 88.25 $\pm$ 0.14 | 86.33 $\pm$ 0.17 | 46.60 $\pm$ 0.18 | 32.39 $\pm$ 0.20 |
| SSL-HSIC (Li et al., 2021) | 91.95 $\pm$ 0.14 | 89.99 $\pm$ 0.17 | 67.23 $\pm$ 0.26 | 57.01 $\pm$ 0.27 | 92.09 $\pm$ 0.20 | 88.91 $\pm$ 0.29 | 51.37 $\pm$ 0.15 | 36.03 $\pm$ 0.12 |
| CorInfoMax (Ozsoy et al., 2022) | 91.81 $\pm$ 0.11 | 89.85 $\pm$ 0.13 | 67.09 $\pm$ 0.24 | 56.92 $\pm$ 0.23 | 91.85 $\pm$ 0.25 | 89.99 $\pm$ 0.24 | 51.23 $\pm$ 0.14 | 35.98 $\pm$ 0.09 |
| MEC (Liu et al., 2022) | 90.55 $\pm$ 0.22 | 87.80 $\pm$ 0.10 | 67.36 $\pm$ 0.27 | 57.25 $\pm$ 0.25 | 91.33 $\pm$ 0.14 | 89.03 $\pm$ 0.33 | 50.93 $\pm$ 0.13 | 36.28 $\pm$ 0.14 |
| VICRegL (Bardes et al., 2022) | 90.99 $\pm$ 0.13 | 88.75 $\pm$ 0.26 | 68.03 $\pm$ 0.32 | 57.34 $\pm$ 0.29 | 92.12 $\pm$ 0.26 | 90.01 $\pm$ 0.20 | 51.52 $\pm$ 0.13 | 36.24 $\pm$ 0.16 |
| VoCo (Wu et al., 2024) | 91.19 $\pm$ 0.10 | 89.12 $\pm$ 0.31 | 68.69 $\pm$ 0.39 | 57.81 $\pm$ 0.20 | 92.46 $\pm$ 0.21 | 90.54 $\pm$ 0.20 | 52.37 $\pm$ 0.15 | 37.12 $\pm$ 0.15 |
| SimCLR + GeSSL | 93.45 $\pm$ 0.21 | 91.35 $\pm$ 0.14 | 69.72 $\pm$ 0.15 | 58.80 $\pm$ 0.16 | 93.45 $\pm$ 0.22 | 91.72 $\pm$ 0.14 | 53.92 $\pm$ 0.17 | 37.49 $\pm$ 0.21 |
| MoCo + GeSSL | 93.05 $\pm$ 0.18 | 89.48 $\pm$ 0.20 | 68.48 $\pm$ 0.12 | 59.44 $\pm$ 0.18 | 93.42 $\pm$ 0.15 | 89.16 $\pm$ 0.26 | 52.34 $\pm$ 0.13 | 37.35 $\pm$ 0.11 |
| BYOL + GeSSL | **94.05 $\pm$ 0.19** | **92.60 $\pm$ 0.28** | 69.45 $\pm$ 0.18 | 59.15 $\pm$ 0.14 | 94.55 $\pm$ 0.16 | 90.73 $\pm$ 0.15 | **55.12 $\pm$ 0.16** | 37.76 $\pm$ 0.22 |
| Barlow Twins + GeSSL | 93.18 $\pm$ 0.16 | 91.23 $\pm$ 0.14 | 69.85 $\pm$ 0.16 | 60.12 $\pm$ 0.14 | 93.98 $\pm$ 0.08 | 89.76 $\pm$ 0.23 | 52.85 $\pm$ 0.12 | 35.39 $\pm$ 0.14 |
| SwAV + GeSSL | 93.37 $\pm$ 0.19 | 90.24 $\pm$ 0.12 | 70.28 $\pm$ 0.19 | 59.63 $\pm$ 0.20 | 93.05 $\pm$ 0.26 | **91.92 $\pm$ 0.21** | 52.12 $\pm$ 0.22 | 37.05 $\pm$ 0.30 |
| DINO + GeSSL | 93.08 $\pm$ 0.21 | 92.38 $\pm$ 0.22 | **71.15 $\pm$ 0.16** | **62.03 $\pm$ 0.31** | 94.65 $\pm$ 0.24 | 91.67 $\pm$ 0.18 | 53.74 $\pm$ 0.22 | 38.12 $\pm$ 0.21 |
| VoCo + GeSSL | 93.10 $\pm$ 0.01 | 91.16 $\pm$ 0.37 | 70.53 $\pm$ 0.30 | 59.34 $\pm$ 0.19 | 94.10 $\pm$ 0.22 | 92.01 $\pm$ 0.26 | 54.37 $\pm$ 0.06 | **38.76 $\pm$ 0.10** |

- UMTRA (Khodadadeh et al., 2019) uses random selection and augmentation to create tasks with pseudo-labels from unlabeled data. It then trains a classifier on each task and adapts it to the target task using a few labeled examples.

- LASIUM (Khodadadeh et al., 2020) uses latent space interpolation to generate tasks with pseudo-labels from a generative model. It then trains an energy-based model on each task and adapts it to the target task using a few labeled examples.

- SVEBM (Kong et al., 2021) uses a symbol-vector coupling energy-based model to learn from unlabeled data. It then adapts the model to the target task using a diffusion process.

- GMVAE (Lee et al., 2021) uses a Gaussian mixture variational autoencoder to perform learning, and then adapts the model to the target task using a variational inference process.

- PsCo (Jang et al., 2023) uses a probabilistic subspace clustering model to learn from unlabeled data. It then adapts the model to the target task using a few labeled examples and a subspace alignment process.

# F ADDITIONAL EXPERIMENTS

## F.1 UNSUPERVISED LEARNING

In this section, we present additional results of the unsupervised learning experiments. Specifically, Table 6 shows the results on four small-scale datasets. We can observe that applying the proposed GeSSL framework significantly outperforms the state-of-the-art (SOTA) methods on all four datasets. Table 6 shows the results on four small-scale datasets. The results still demonstrate the proposed GeSSL's ability to enhance the performance of self-supervised learning methods, achieving significant improvements over the original models on all baselines. Moreover, applying our GeSSL framework to all four types of representative SSL models as described in Section 6, including SimCLR, MoCo, BYOL, Barlow Twins, SwAV, and DINO, achieves an average improvement of 3% compared to the original frameworks. Table 10 provides the comparison results of our proposed GeSSL on a large-scale dataset, i.e., ImageNet. The results show that, (i) the self-supervised learning model applying GeSSL achieves the state-of-the-art result (SOTA) performance under all epoch conditions; and (ii) after applying the proposed GeSSL, the self-supervised learning models consistently outperforms the original frameworks in terms of average classification accuracy at 100, 200 and 400 epochs. For 1000 epochs, VICRegL + GeSSL yields the best result among other state-of-the-art methods, with an average accuracy of 78.72%.

Table 7: Results on ImageNet (Accuracy).

| Methods | Original | +$f$-MICL | +RINCE | +PID | +GeSSL |
|---|---|---|---|---|---|
| SigCLR | 70.9 | 72.2 | 72.1 | 73.1 | 73.8 |
| SinSim | 71.9 | 73.1 | 72.7 | 73.4 | 74.0 |
| SimCLR-Cut | 71.0 | 72.9 | 72.8 | 73.2 | 73.7 |
| I-MAE | 74.9 | 75.4 | 75.2 | 76.0 | 76.9 |
| ColorMAE | 74.1 | 74.6 | 74.2 | 74.9 | 75.3 |

Table 8: Results on COCO (AP).

| Methods | Original | +$f$-MICL | +RINCE | +PID | +GeSSL |
|---|---|---|---|---|---|
| SigCLR | 39.0 | 39.5 | 40.0 | 41.2 | 41.6 |
| SinSim | 39.8 | 40.5 | 40.3 | 41.5 | 42.7 |
| SimCLR-Cut | 38.4 | 40.2 | 40.9 | 41.3 | 42.3 |
| I-MAE | 42.0 | 42.9 | 43.2 | 44.6 | 45.5 |
| ColorMAE | 41.2 | 42.0 | 42.7 | 44.4 | 45.4 |

**More recent methods and baselines** The effect of GeSSL is reflected in the performance improvement when applying it to the SSL baselines. The experimental results above have demonstrated that after the introduction of GeSSL, the effects of all SSL baselines have been significantly improved. These results have shown the outstanding effectiveness and robustness of GeSSL. The SSL baselines we use cover all SOTA methods on the leaderboard of the adopted benchmark datasets (before submission). The methods proposed recently are mainly variants of the currently used comparison baselines.

To evaluate the effect of GeSSL on recently proposed methods, we select the two SSL methods published in ICML23 for testing (Baevski et al., 2023; Joshi & Mirzasoleiman, 2023), where we follow the same experimental settings. The results are shown in Tables 14 and 15. The results still prove the effectiveness of GeSSL.

Besides the above results, we expanded our experiments to include recent SSL models, i.e., SigCLR (Çağatan, 2024), SinSim (Sepanj & Fiegth, 2025), SimCLR-Cut (Draganov et al., 2024), I-MAE (Zhang & Shen, 2024), and ColorMAE (Hinojosa et al., 2024), and compared against recently proposed methods that target SOTA SSL improvements, i.e., f-MICL, RINCE, and PID. Following the experimental setup of Section 5.1, we evaluate ResNet-50 and ViT backbones in both the unsupervised setting (ImageNet, 200 epochs) and transfer learning (COCO). The results in **Table 7** and **Table 8** report ImageNet accuracy and COCO AP and demonstrate that integrating GeSSL consistently improves performance across architectures and benchmarks; moreover, the gains from GeSSL are generally larger than those achieved by baselines.

**Performance on more larger datasets** To evaluate the scalability and generalization ability of GeSSL across a broad range of datasets, including data sources that substantially exceed the scale of ImageNet-1K. Beyond pretraining on ImageNet-1K and CIFAR-100, we further assess GeSSL on larger datasets such as ImageNet, the 250M-image DALLE corpus, COCO, consistently observing an average improvement of over 2%. Following the large-scale pretraining protocol of (Al Kader Hammoud et al., 2024), we also trained models on the YFCC100M dataset and evaluated them on the Cars dataset. As shown in Table 9, integrating GeSSL yields steady and significant gains across representative SSL frameworks, demonstrating that GeSSL remains effective when scaled to diverse and extensive visual data.

## F.2 TRANSFER LEARNING

As mentioned in Section 5.1, we construct three sets of transfer learning experiments, including the most commonly used object detection and instance segmentation protocol (Chen et al., 2020a; Zbontar et al., 2021; Grill et al., 2020), transfer to other domains (different datasets), and transfer

Table 9: Results on the Cars dataset after pretraining on YFCC100M.

| Method | Original | +GeSSL |
|--------|----------|--------|
| SimCLR | 37.2 | 39.9 |
| BYOL | 35.0 | 39.7 |
| MAE | 40.9 | 44.6 |

Table 10: The Top-1 and Top-5 classification accuracies of linear classification on the ImageNet dataset with ResNet-50 as the feature extractor. We record the comparison results from 100, 200, 400, and 1000 epochs.

| Method | 100 Epochs | | 200 Epochs | | 400 Epochs | 1000 Epochs |
|--------|------------|------------|------------|------------|------------|------------|
| | Top-1 | Top-5 | Top-1 | Top-5 | Top-1 | Top-1 |
| Supervised | 71.93 | - | 73.45 | - | 74.92 | 76.35 |
| SimCLR (Chen et al., 2020a) | $66.54 \pm 0.22$ | $88.14 \pm 0.26$ | $68.32 \pm 0.31$ | $89.76 \pm 0.23$ | $69.24 \pm 0.21$ | $70.45 \pm 0.30$ |
| MoCo (Chen et al., 2020b) | $64.53 \pm 0.25$ | $86.17 \pm 0.11$ | $67.55 \pm 0.27$ | $88.42 \pm 0.11$ | $69.76 \pm 0.14$ | $71.16 \pm 0.23$ |
| BYOL (Grill et al., 2020) | $67.65 \pm 0.27$ | $88.95 \pm 0.11$ | $69.94 \pm 0.21$ | $89.45 \pm 0.27$ | $71.85 \pm 0.12$ | $73.35 \pm 0.27$ |
| SimSiam (Chen & He, 2021) | $68.14 \pm 0.26$ | $87.12 \pm 0.26$ | $70.02 \pm 0.14$ | $88.76 \pm 0.23$ | $70.86 \pm 0.34$ | $71.37 \pm 0.22$ |
| Barlow Twins (Zbontar et al., 2021) | $67.24 \pm 0.22$ | $88.66 \pm 0.19$ | $69.94 \pm 0.32$ | $88.97 \pm 0.27$ | $70.22 \pm 0.15$ | $73.29 \pm 0.13$ |
| SwAV (Caron et al., 2020) | $66.55 \pm 0.27$ | $88.42 \pm 0.22$ | $69.12 \pm 0.24$ | $89.38 \pm 0.20$ | $70.78 \pm 0.34$ | $75.32 \pm 0.11$ |
| DINO (Caron et al., 2021) | $67.23 \pm 0.19$ | $88.48 \pm 0.21$ | $70.58 \pm 0.24$ | $91.32 \pm 0.27$ | $71.98 \pm 0.26$ | $73.94 \pm 0.29$ |
| W-MSE (Ermolov et al., 2021) | $67.48 \pm 0.29$ | $90.39 \pm 0.27$ | $70.85 \pm 0.31$ | $91.57 \pm 0.20$ | $72.49 \pm 0.24$ | $72.84 \pm 0.18$ |
| RELIC v2 (Tomasev et al., 2022) | $66.38 \pm 0.23$ | $90.89 \pm 0.21$ | $70.98 \pm 0.21$ | $91.15 \pm 0.26$ | $71.84 \pm 0.21$ | $72.17 \pm 0.20$ |
| LMCL (Chen et al., 2021) | $66.75 \pm 0.13$ | $89.85 \pm 0.36$ | $70.83 \pm 0.26$ | $90.04 \pm 0.21$ | $72.53 \pm 0.24$ | $72.97 \pm 0.29$ |
| ReSSL (Zheng et al., 2021) | $67.41 \pm 0.27$ | $90.55 \pm 0.23$ | $69.92 \pm 0.24$ | $91.25 \pm 0.12$ | $72.46 \pm 0.29$ | $72.91 \pm 0.30$ |
| CorInfoMax (Ozsoy et al., 2022) | $70.13 \pm 0.12$ | $91.14 \pm 0.25$ | $70.83 \pm 0.15$ | $91.53 \pm 0.22$ | $73.28 \pm 0.24$ | $74.87 \pm 0.36$ |
| MEC (Liu et al., 2022) | $69.91 \pm 0.10$ | $90.67 \pm 0.15$ | $70.34 \pm 0.27$ | $91.25 \pm 0.38$ | $72.91 \pm 0.27$ | $75.07 \pm 0.24$ |
| VICRegL (Bardes et al., 2022) | $69.99 \pm 0.25$ | $91.27 \pm 0.16$ | $70.24 \pm 0.27$ | $91.60 \pm 0.24$ | $72.14 \pm 0.20$ | $75.07 \pm 0.23$ |
| SimCLR + GeSSL | $68.45 \pm 0.20$ | $89.62 \pm 0.23$ | $69.88 \pm 0.21$ | $91.32 \pm 0.25$ | $71.50 \pm 0.16$ | $72.82 \pm 0.28$ |
| MoCo + GeSSL | $66.78 \pm 0.19$ | $88.41 \pm 0.20$ | $69.60 \pm 0.30$ | $91.28 \pm 0.39$ | $70.82 \pm 0.29$ | $73.04 \pm 0.22$ |
| SimSiam + GeSSL | $70.61 \pm 0.18$ | $88.61 \pm 0.17$ | $72.04 \pm 0.22$ | $89.43 \pm 0.40$ | $72.78 \pm 0.17$ | $74.78 \pm 0.24$ |
| Barlow Twins + GeSSL | $69.62 \pm 0.21$ | $89.55 \pm 0.19$ | $72.84 \pm 0.26$ | $89.50 \pm 0.19$ | $74.10 \pm 0.13$ | $75.02 \pm 0.22$ |
| SwAV + GeSSL | $69.05 \pm 0.18$ | $89.50 \pm 0.17$ | $72.28 \pm 0.19$ | $90.68 \pm 0.30$ | $72.88 \pm 0.18$ | $76.38 \pm 0.19$ |
| DINO + GeSSL | $69.55 \pm 0.20$ | $90.62 \pm 0.22$ | $73.52 \pm 0.30$ | $\mathbf{94.05 \pm 0.26}$ | $74.02 \pm 0.26$ | $76.40 \pm 0.21$ |
| VICRegL + GeSSL | $\mathbf{72.75 \pm 0.21}$ | $\mathbf{91.45 \pm 0.18}$ | $\mathbf{73.91 \pm 0.36}$ | $93.77 \pm 0.35$ | $\mathbf{74.38 \pm 0.23}$ | $\mathbf{78.85 \pm 0.29}$ |

learning on video-based tasks. The results of the first experiment are illustrated in Section 5.1, and the other two sets of experiments are described below.

**Transfer to other domains.** To explore the nature of transfer learning of the proposed framework, we leverage models that had been pre-trained on the CIFAR100 dataset, including SimCLR (Chen et al., 2020a), BYOL (Grill et al., 2020), and Barlow Twins (Zbontar et al., 2021), on the CIFAR100 dataset. We then applied these models to four distinct datasets, including CIFAR10 (Krizhevsky et al., 2009), Flower102 (Nilsback & Zisserman, 2008), Food101 (Bossard et al., 2014), and Aircraft (Maji et al., 2013). We first calculate the classification performance (Top-1) based on the existing self-supervised model on different data sets, recorded as $acc(\text{method}, \text{dataset})$, such as $acc(\text{SimCLR}, \text{Flower102})$. Then, we calculate the model's classification performance by incorporating GeSSL on those data sets, which is recorded as $acc(\text{method} + \text{GeSSL}, \text{dataset})$. Finally, we get the improvement $\Delta(\text{method}, \text{dataset}) = acc(\text{method} + \text{GeSSL}, \text{dataset}) - acc(\text{method}, \text{dataset})$ in classification performance on each dataset, as shown in Table 11. The results show that the migration effect of the model after applying the GeSSL framework has been steadily improved, proving that GeSSL has effectively improved the versatility of the SSL model.

**Video-based Task** In order to assess the performance of our method with video-based tasks, we transition our pre-trained model to handle a variety of video tasks, utilizing the UniTrack evaluation framework (Wang et al., 2021) as our testing ground. The findings are compiled in Table 12, which includes results from five distinct tasks, drawing on the features from [layer3/layer4] of the Resnet-50. The data indicates that existing SSL methods incorporating our GeSSL significantly surpass original SSL approaches, with SimCLR achieving more than a 2% improvement in VOS (Perazzi et al., 2016), and BYOL seeing over a 3% gain in MOT (Milan et al., 2016).

Table 11: The performance of adding task information in self-supervised models on different datasets.

| Evl.dataset | SimCLR+GeSSL | BYOL+GeSSL | Barlow Twins+GeSSL | VICRegL+GeSSL |
|---|---|---|---|---|
| CIFAR10 | +3.56 | +2.51 | +2.17 | +2.80 |
| Flower102 | +4.03 | +2.09 | +2.94 | +3.07 |
| Food101 | +1.85 | +2.31 | +2.01 | +2.02 |
| Aircraft | +2.57 | +2.89 | +2.24 | +2.34 |

Table 12: Transfer learning on video tracking tasks. All methods use the same ResNet-50 backbone and are evaluated based on UniTrack.

| Method | SOT | | VOS | MOT | | MOTS | | PoseTrack |
|---|---|---|---|---|---|---|---|---|
| | $AUC_{XCorr}$ | $AUC_{DCF}$ | $\mathcal{J}$-mean | IDF1 | HOTA | IDF1 | HOTA | IDF1 |
| SimCLR | 47.3 / 51.9 | 61.3 / 50.7 | 60.5 / 56.5 | 66.9 / 75.6 | 57.7 / 63.2 | 65.8 / 67.6 | 67.7 / 69.5 | 72.3 / 73.5 |
| MoCo | 50.9 / 47.9 | 62.2 / 53.7 | 61.5 / 57.9 | 69.2 / 74.1 | 59.4 / 61.9 | 70.6 / 69.3 | 71.6 / 70.9 | 72.8 / 73.9 |
| SwAV | 49.2 / 52.4 | 61.5 / 59.4 | 59.4 / 57.0 | 65.6 / 74.4 | 56.9 / 62.3 | 68.8 / 67.0 | 69.9 /69.5 | 72.7 / 73.6 |
| BYOL | 48.3 / 55.5 | 58.9 / 56.8 | 58.8 / 54.3 | 65.3 / 74.9 | 56.8 / 62.9 | 70.1 / 66.8 | 70.8 / 69.3 | 72.4 / 73.8 |
| Barlow Twins | 44.5 / 55.5 | 60.5 / **60.1** | 61.7 / 57.8 | 63.7 / 74.5 | 55.4 / 62.4 | 68.7 / 67.4 | 69.5 / 69.8 | 72.3 / 74.3 |
| SimCLR+GeSSL | 51.0 / 54.4 | **63.7** / 53.5 | **62.3 / 58.3** | **70.3 / 77.3** | **60.5 / 65.0** | 68.1 / **69.6** | 69.2 / **71.2** | 73.7 / 74.4 |
| BYOL+GeSSL | **52.0 / 57.9** | 60.5 / 59.0 | 61.2 / 57.3 | 68.0 / 77.1 | 58.2 / 64.4 | **73.0** / 68.6 | **73.6** / 71.1 | **75.0 / 75.8** |

## F.3 FEW-SHOT LEARNING

The outstanding performance of GeSSL in the few-shot learning scenario has been confirmed in Section 5.1, where it can produce good results with limited data. However, the situation becomes complicated in scenarios where data collection is infeasible in real life, such as medical diagnosis and satellite imagery (Zheng, 2015; Tang et al., 2012). Therefore, the performance of the model on cross-domain few-shot learning tasks is crucial, as it determines the applicability of the learning model (Guo et al., 2020). To ensure that GeSSL can achieve robust performance in real-world applications, we further conduct comparative experiments on cross-domain few-shot learning.

**Experimental setup.** We compare our proposed GeSSL with the few-shot learning baselines as described in Table 4 on cross-domain few-shot learning. The details of the baselines are illustrated in Appendix E. We adopt six cross-domain few-shot learning benchmark datasets, and divided these datasets into two categories according to their similarity with ImageNet: i) high similarity: CUB (Welinder et al., 2010), Cars (Krause et al., 2013), and Places (Zhou et al., 2017); ii) low similarity: CropDiseases (Mohanty et al., 2016), ISIC (Codella et al., 2018), and ChestX (Wang et al., 2017). The $(N, A)$ in the tables means the $N$-way $A$-shot tasks with $N$ classes and $N \times A$ samples, where each class has $A$ samples augmented from the same image.

**Results**. Table 13 presents the performance of the model trained on miniImageNet and transfer to the six cross-domain few-shot learning benchmark datasets mentioned above. By observation, we further validate the performance of our proposed GeSSL: i) Effectiveness: achieves better results than the state-of-the-art baselines on almost all benchmark datasets; ii) Generalization: achieves nearly a 3% improvement compared to unsupervised few-shot Learning and self-supervised learning on the datasets with significant differences from the training phase; iii) Robustness: achieves better results than the PsCo (Jang et al., 2023) which introduces out-of-distribution samples, even though we do not explicitly consider out-of-distribution samples on datasets with significant differences.

## F.4 UNIVERSALITY OF EXISTING SSL METHODS

Current self-supervised learning (SSL) models overlook the explicit incorporation of universality within their objectives, and the corresponding theoretical comprehension remains inadequate, posing challenges for SSL models to attain universality in practical, real-world applications (Huang et al., 2021; Sun et al., 2020; Ericsson et al., 2022). Therefore, we propose a provable $\sigma-$measure to help evaluate the model universality, and further build GeSSL based on it to explicitly model universality into the SSL's learning objective. In this Section, we specifically quantify the universality scores of

Table 13: The cross-domain few-shot learning accuracies ($\pm 95\%$ confidence interval). We transfer models trained on miniImageNet to six benchmark datasets with the C4-backbone. The best results are highlighted in **bold**. The $(N, A)$ means the $N$-way $A$-shot tasks with $N$ classes and $N \times A$ samples, where each class has $A$ samples augmented from the same image.

| Method | CUB | | Cars | | Places | |
|---|---|---|---|---|---|---|
| | **(5,5)** | **(5,20)** | **(5,5)** | **(5,20)** | **(5,5)** | **(5,20)** |
| *Unsupervised Few-shot Learning* | | | | | | |
| MetaSVEBM | $45.893 \pm 0.334$ | $54.823 \pm 0.347$ | $33.530 \pm 0.367$ | $44.622 \pm 0.299$ | $50.516 \pm 0.397$ | $61.561 \pm 0.412$ |
| MetaGMVAE | $48.783 \pm 0.426$ | $55.651 \pm 0.367$ | $30.205 \pm 0.334$ | $39.946 \pm 0.400$ | $55.361 \pm 0.237$ | $65.520 \pm 0.374$ |
| PsCo | $56.365 \pm 0.636$ | $69.298 \pm 0.523$ | $44.632 \pm 0.726$ | $56.990 \pm 0.551$ | $64.501 \pm 0.780$ | $73.516 \pm 0.499$ |
| *Self-supervised Learning* | | | | | | |
| SimCLR | $51.389 \pm 0.365$ | $60.011 \pm 0.485$ | $38.639 \pm 0.432$ | $52.412 \pm 0.783$ | $59.523 \pm 0.461$ | $68.419 \pm 0.500$ |
| MoCo | $52.843 \pm 0.347$ | $61.204 \pm 0.429$ | $39.504 \pm 0.489$ | $50.108 \pm 0.410$ | $60.291 \pm 0.583$ | $69.033 \pm 0.654$ |
| SwAV | $51.250 \pm 0.530$ | $61.645 \pm 0.411$ | $36.352 \pm 0.482$ | $51.153 \pm 0.399$ | $58.789 \pm 0.403$ | $68.512 \pm 0.466$ |
| SimCLR + GeSSL | $55.922 \pm 0.471$ | $64.723 \pm 0.214$ | $43.892 \pm 0.198$ | $56.100 \pm 0.269$ | $65.125 \pm 0.301$ | $72.892 \pm 0.240$ |
| MoCo + GeSSL | $\mathbf{57.650 \pm 0.221}$ | $65.502 \pm 0.274$ | $\mathbf{45.529 \pm 0.295}$ | $55.354 \pm 0.237$ | $\mathbf{66.602 \pm 0.180}$ | $\mathbf{74.126 \pm 0.243}$ |
| SwAV + GeSSL | $55.421 \pm 0.173$ | $\mathbf{65.927 \pm 0.460}$ | $42.237 \pm 0.296$ | $\mathbf{56.682 \pm 0.380}$ | $64.601 \pm 0.325$ | $72.460 \pm 0.463$ |

| Method | CropDiseases | | ISIC | | ChestX | |
|---|---|---|---|---|---|---|
| | **(5,5)** | **(5,20)** | **(5,5)** | **(5,20)** | **(5,5)** | **(5,20)** |
| *Unsupervised Few-shot Learning* | | | | | | |
| MetaSVEBM | $71.652 \pm 0.837$ | $84.515 \pm 0.902$ | $37.106 \pm 0.732$ | $48.001 \pm 0.723$ | $27.238 \pm 0.685$ | $29.652 \pm 0.610$ |
| MetaGMVAE | $72.683 \pm 0.527$ | $80.777 \pm 0.511$ | $30.630 \pm 0.423$ | $37.574 \pm 0.399$ | $24.522 \pm 0.405$ | $26.239 \pm 0.422$ |
| PsCo | $\mathbf{89.565 \pm 0.372}$ | $95.492 \pm 0.399$ | $43.632 \pm 0.400$ | $54.886 \pm 0.359$ | $21.907 \pm 0.258$ | $24.182 \pm 0.389$ |
| *Self-supervised Learning* | | | | | | |
| SimCLR | $80.360 \pm 0.488$ | $89.161 \pm 0.456$ | $44.669 \pm 0.510$ | $51.823 \pm 0.411$ | $26.556 \pm 0.385$ | $30.982 \pm 0.422$ |
| MoCo | $81.606 \pm 0.485$ | $90.366 \pm 0.377$ | $44.328 \pm 0.488$ | $52.398 \pm 0.396$ | $24.198 \pm 0.400$ | $27.893 \pm 0.412$ |
| SwAV | $80.055 \pm 0.502$ | $89.917 \pm 0.539$ | $43.200 \pm 0.356$ | $50.109 \pm 0.350$ | $21.252 \pm 0.439$ | $28.270 \pm 0.417$ |
| SimCLR + GeSSL | $84.526 \pm 0.413$ | $94.572 \pm 0.332$ | $\mathbf{47.310 \pm 0.389}$ | $55.710 \pm 0.312$ | $\mathbf{30.876 \pm 0.259}$ | $\mathbf{34.492 \pm 0.398}$ |
| MoCo + GeSSL | $85.852 \pm 0.358$ | $\mathbf{95.540 \pm 0.335}$ | $46.437 \pm 0.339$ | $\mathbf{56.466 \pm 0.270}$ | $29.216 \pm 0.332$ | $31.545 \pm 0.279$ |
| SwAV + GeSSL | $85.355 \pm 0.327$ | $94.785 \pm 0.339$ | $46.521 \pm 0.288$ | $55.268 \pm 0.312$ | $27.462 \pm 0.340$ | $32.237 \pm 0.199$ |

Table 14: Top-1 validation accuracy on ImageNet-1K dataset for ViT-B and ViT-L.

| Method | Epoch | ViT-B | ViT-L |
|---|---|---|---|
| data2vec 2.0 | 200/150 | 80.5 | 81.8 |
| data2vec 2.0 + GeSSL | 200/150 | 85.9 | 88.2 |

Table 15: Downstream classification accuracy of SimCLR-SAS on CIFAR-10.

| Method | Subset Size | Top-1 Accuracy (%) |
|---|---|---|
| SimCLR-SAS | 10% | 79.7 |
| SimCLR-SAS + GeSSL | 10% | 84.1 |

existing SSL methods based on $\sigma-$measure, and verify that our proposed GeSSL actually improves the model universality.

Specifically, the $\sigma$-measurement score assesses the difference in performance between the learned model and the optimal model for each task. The optimal model is assumed to output the ground truth, and the performance difference is quantified using the KL divergence between the predicted and true class probability distributions. It compares the predicted class probabilities produced by classifier $\pi$ to the true labels across SSL tasks, such as comparing the predicted values $[0.81, 0.09, 0.03, 0.07]$ to the true labels $[1, 0, 0, 0]$. Take LIN task with SimCLR as an example, we train SimCLR and SimCLR+GeSSL on the COCO dataset for 200 epochs, then add a MLP after the feature extractor. A new mini-batch is input into both SimCLR and SimCLR+GeSSL to generate class probability distributions for each sample, and the KL divergence between these predicted and true distributions is calculated. After normalization, the scores for the LIN task are obtained, with similar evaluations conducted for other baselines and tasks.

**Why this metric is sensible.** For convenience, we convert the normalized $\sigma$ into a "goodness" score $S = 1 - \mathrm{norm}(\sigma)$. Intuitively, $S$ captures the three components of universality: (i) Discriminability: On the training task, we attach a light-weight classification head to the encoder and compute the KL between its predictions and the ground-truth one-hot labels. If the model concentrates probability mass on the correct class (i.e., behaves like an optimal model with high accuracy and confidence), the KL tends to zero ($\sigma \rightarrow 0$) and $S \rightarrow 1$. Thus, $S$ directly measures the ability to form clear class boundaries on the training data. (ii) Generalizability: Applying the same KL computation on the

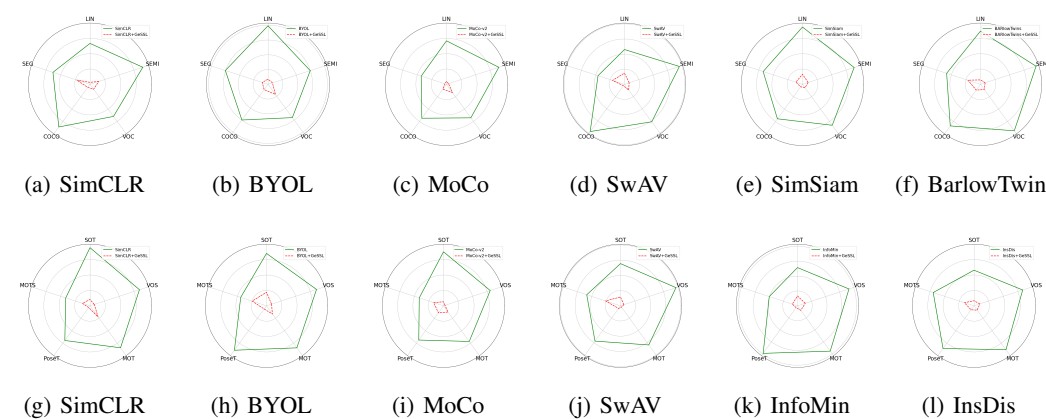

(a) SimCLR  (b) BYOL  (c) MoCo  (d) SwAV  (e) SimSiam  (f) BarlowTwins

(g) SimCLR  (h) BYOL  (i) MoCo  (j) SwAV  (k) InfoMin  (l) InsDis

Figure 5: Universality performance of different models on five image-based tasks (top row) and five video-based tasks (bottom row). We choose $\sigma-$measure as the measurement. It is worth noting that the smaller the $\sigma-$measurefen score, the better the effect. Meanwhile, we normalize the results of $\sigma-$measurefen scores on different datasets and compare the performance between baselines by comparing the corresponding branch of the fan chart.

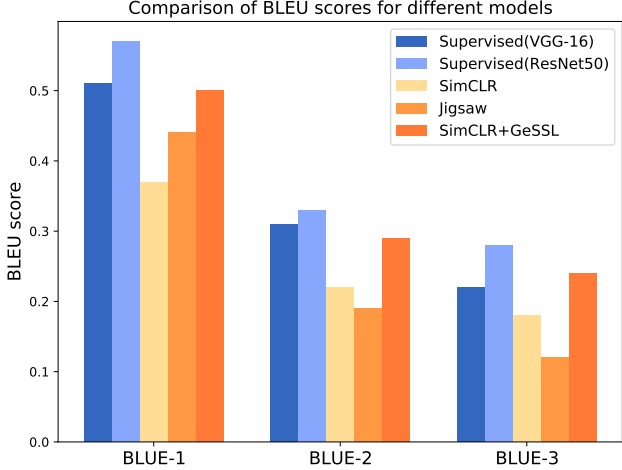

Figure 6: Comparison of BLEU scores for different models, comparing 2 fully supervised and 3 self-supervised pre-text tasks, trained on the Flickr8k.

task's test split yields a test $\sigma$. If the model produces near one-hot predictions on unseen samples (small KL), then $S$ is high, indicating a small train, then test performance drop (small generalization gap). Hence, the train/test $\sigma$ values and their difference directly quantify the paper's requirement of "maintaining high accuracy on test". (iii) Transferability: For a collection of new test tasks whose label sets do not overlap with the training task, we use the same encoder, train/evaluate a light-weight head per new task, compute the task-specific $\sigma$ values, and aggregate them (e.g., by averaging) to obtain a cross-task $\sigma$. A small aggregated $\sigma$ indicates that the representation approaches the per-task optimal models across many new tasks, i.e., it has strong transferability.

In the experiments, we chose two scenarios based on images and videos to evaluate the model versatility following (Liu et al., 2022). The image-based tasks include linear probing (top-1 accuracy) with 800-epoch pre-trained models (LIN), semi-supervised classification (top-1 accuracy) using 1% subset of training data (SEMI), object detection (AP) on VOC dataset (VOC) and COCO dataset (COCO), instance segmentation ($AP^{mask}$) on COCO dataset (SEG). For video-based tasks, we compute rankings in terms of AUC for SOT, $\mathcal{J}$-mean for VOS, IDF-1 for MOT, IDF-1 for PoseTracking, and IDF-1 for MOTS, respectively. Next, we evaluate the $\sigma$-measurement scores of different base-

Table 16: The performance of introducing GeSSL during training. All results are recorded during training using the $\sigma$-measurement.

| Metric | Training Epochs | | | | | | | | | |
|---|---|---|---|---|---|---|---|---|---|---|
| | 20 | 40 | 60 | 80 | 100 | 120 | 140 | 160 | 180 | 200 |
| Accuracy of SimCLR | 20.1 | 43.6 | 51.2 | 60.2 | 70.3 | 77.2 | 82.3 | 86.1 | 88.7 | 88.6 |
| Accuracy of SimCLR + GeSSL | 42.4 | 67.1 | 83.0 | 92.9 | 93.0 | 94.4 | 94.1 | 93.2 | 94.1 | 94.2 |
| Performance Ratio $r$ | 0.474 | 0.650 | 0.617 | 0.648 | 0.756 | 0.818 | 0.875 | 0.924 | 0.943 | 0.941 |

Table 17: Comparison between models.

| Method | scratch, original | scratch, our impl. | baseline MAE | MAE + Our |
|---|---|---|---|---|
| Top 1 | 76.5 | 82.7 | 85.3 | 88.1 |

lines before and after the introduction of GeSSL and after training for 200 epochs. Among them, the better model is set to the result of ground truth, and the calculation of $\sigma$-measurement score is performed on a series of randomly sampled tasks.

Figure 5 shows the comparison results. Note that the lower $\sigma-$measure denotes the better performance. From the results, we can observe that: (i) the $\sigma$-measurement score of the existing SSL model is low and it is difficult to achieve good results in multiple domains and tasks; (ii) after the introduction of GeSSL, the $\sigma$-measurement score of the SSL models are significantly decreased. The results demonstrate that the existing SSL model has limited universality (proves the description in Section 1), and the performance improvement brought by GeSSL is achieved by improving the universality.

Considering that the above experiments evaluate the evaluation universality of SSL models, here, we construct the following numerical experiments to evaluate learning universality: In the first 20-200 epochs of training (each epoch contains multiple tasks), we evaluate the average performance of multiple $f'$ in each epoch. Each $f_\theta^l$ is obtained by updating $f_\theta$ on the corresponding support set. We calculate the accuracy of SimCLR before and after the introduction of GeSSL and the ratio $r$ of their effects on the CIFAR-10 data set. If $r < 1$, it means that the representation effect learned by the model in each epoch of training is better when introducing GeSSL. The results for every 20 epochs are shown in Table 16. The results show that: (i) $r$ is always less than 1, which proves that the representation effect learned after the introduction of GeSSL is significantly improved; (ii) after the introduction of GeSSL, the accuracy of the model is significantly improved, and it becomes stable after 80 epochs, i.e., great results can be achieved for even based on just one iteration and few data. These results show that "the model $f_\theta$ achieves comparable performance on each task quickly with few data during training" after introducing GeSSL.

### F.5 EVALUATION ON GENERATIVE SELF-SUPERVISED LEARNING

In this Section, we evaluate the effectiveness of the proposed GeSSL on the generative self-supervised learning paradigm. We conduct experiments on three scenarios, including image generation, image captioning, and object detection and segmentation.

**Evaluation on Image Generation** To explore the effect of GeSSL on generative SSL, we conduct a set of experiments on ImageNet-1K dataset (Deng et al., 2009b). Specifically, we begin by conducting self-supervised pre-training on the ImageNet-1K (IN1K) training set. Following this, we carry out supervised training to assess the representations using either (i) end-to-end fine-tuning or (ii) linear probing. The results are reported as the top-1 validation accuracy for a single 224×224 crop. For this process, we utilize ViT-Large (ViT-L/16) (Dosovitskiy et al., 2020) as the backbone. Note that ViT-L is very big (an order of magnitude bigger than ResNet-50 (He et al., 2016)) and tends to overfit, as shown in Table 17. The comparison results are shown in Table 18. We can observe that GeSSL achieves stable performance improvements

Table 18: Comparisons with previous results on ImageNet-1K. The ViT models are B/16, L/16, H/14 (Dosovitskiy et al., 2020). The pre-training data is the ImageNet-1K training set (except the tokenizer in BEiT was pre-trained on 250M DALLE data (Ramesh et al., 2021)). All results are on an image size of 224, except for ViT-H with an extra result of 448.

| Method | pre-train data | ViT-B | ViT-L | ViT-H | ViT-H$_{448}$ |
|--------|----------------|-------|-------|-------|---------------|
| DINO | IN1K | 82.8 | - | - | - |
| MoCo | IN1K | 83.2 | 84.1 | - | - |
| BEiT | IN1K+DALLE | 83.2 | 85.2 | - | - |
| MAE | IN1K | 83.6 | 85.9 | 86.9 | 87.8 |
| MAE+Ours | IN1K | 87.6 | 88.5 | 89.2 | 89.7 |

Table 19: COCO object detection and segmentation using a ViT Mask R-CNN baseline. All self-supervised entries use IN1K data without labels, and Mask AP follows a similar trend as box AP.

| Method | pre-train data | AP$^{box}$ | | AP$^{mask}$ | |
|--------|----------------|------------|-------|-------------|-------|
| | | ViT-B | ViT-L | ViT-B | ViT-L |
| supervised | IN1K w/ labels | 47.9 | 49.3 | 42.9 | 43.9 |
| MoCo v3 | IN1K | 47.9 | 49.3 | 42.7 | 44.0 |
| BEiT | IN1K+DALLE | 49.8 | 53.3 | 44.4 | 47.1 |
| MAE | IN1K | 50.3 | 53.3 | 44.9 | 47.2 |
| MAE + Our | IN1K | 54.9 | 57.3 | 47.9 | 53.0 |

**Evaluation on Image Captioning**  We use the commonly used protocol following (Mohamed et al., 2022). The dataset we use to train the pretext task is the unlabeled part of MSCOCO dataset (Vinyals et al., 2016b), which contains 123K images with an average resolution of $640 \times 480$ pixels. This dataset contains color and grayscale images. For downstream tasks, we use the Flicker8K dataset (Hodosh et al., 2013). Next, we train it using pre-trained pre-text tasks supervised by VGG-16 and ResNet-50, as well as self-supervised pre-text tasks from SimCLR and Jigsaw Puzzle solutions. In the next step, to evaluate the results, we use the BLEU (Bilingual Evaluation Research) score as the evaluation metric, which evaluates the generated sentences against the reference sentences, where a perfect match is 1 and a perfect mismatch is 0, calculating scores for 1, 2, 3 and 4 cumulative n-grams. The results are shown in Figure 6. From the results, we can observe that after introducing the GeSSL framework we proposed, the model effect has been further improved, stably exceeding the SOTA of the SSL method, and even approaching the supervised learning results. The results show that our proposed GeSSL can still achieve good results in generative self-supervised learning.

**Evaluation on Object Detection and Segmentation**  For object detection and segmentation, we fine-tune Mask R-CNN (He et al., 2017) end-to-end on COCO (Lin et al., 2014). The ViT backbone is adapted for use with FPN (Lin et al., 2017). We report box AP for object detection and mask AP for instance segmentation. The results are shown in Table 19. Compared to supervised pre-training, our MAE performs better under all configurations. Our method still achieves optimal results, demonstrating its effectiveness.

Table 20: Performance on for text recognition.

| Methods | IIIT5K | IC03 |
|---------|--------|------|
| SimCLR (Chen et al., 2020a) | 1.7 | 3.8 |
| SeqCLR (Aberdam et al., 2021) | 35.7 | 43.6 |
| SimCLR + GeSSL | 21.4 | 20.8 |
| SeqCLR + GeSSL | 41.3 | 50.6 |

Table 21: Multi-modal performance of image-to-text retrieval tasks on COCO.

| Method | Original | +GeSSL |
|--------|----------|--------|
| MoCo-v2 | 51.6 | 52.9 |
| VirTex | 58.1 | 58.7 |

Table 22: Multi-modal performance of vision–language classification tasks on ColoredMNIST.

| Method | Accuracy |
|--------|----------|
| SimCLR | 82.1 |
| SimCLR+GeSSL | 85.9 |

## F.6 EVALUATION ON MORE MODALITIES

GeSSL proposed in this work can be applied in various fields and domains, e.g., instance segmentation, video tracking, sample generation, etc., as mentioned before. Here, we provide the experiments of GeSSL on text modality-based datasets, i.e., IC03 and IIIT5K (Yasmeen et al., 2020), which we have conducted before. We follow the same experimental settings as mentioned in (Aberdam et al., 2021). The results shown in Table 20 demonstrate that GeSSL achieves stable effectiveness and robustness in various modalities combined with the above experiments.

Besides the above results, to evaluate whether GeSSL extends beyond unimodal visual pretraining, we further assess its effectiveness on multimodal tasks. Following the standard protocols in VirTex (Yuan et al., 2021) and MoCo-v2 multimodal extensions (Huang et al., 2024), we conduct experiments on two representative benchmarks: (i) COCO image-to-text retrieval, where we report Recall@1 (R@1), and (ii) ColoredMNIST, a distribution-shifted vision–language classification benchmark. For COCO, we integrate GeSSL into two representative multimodal feature learners, i.e., MoCo-v2 and VirTex, and compare their retrieval performance before and after applying GeSSL. For ColoredMNIST, we fine-tune SimCLR and SimCLR+GeSSL under the same training protocol and report test accuracy. As shown in **Table 21** and **Table 22**, GeSSL consistently improves multimodal representations, yielding +1.3 R@1 on MoCo-v2, +0.6 R@1 on VirTex, and a substantial +3.8% gain on ColoredMNIST. These results demonstrate that GeSSL enhances cross-modal alignment quality and generalizes effectively to multimodal learning scenarios.

## F.7 ROBUSTNESS UNDER DISTRIBUTION SHIFTS

To further evaluate the universality and robustness of GeSSL-pretrained representations, we assess model performance under explicit distribution shifts. In addition to the cross-dataset and cross-task transfer benchmarks reported above, we also conduct dedicated robustness experiments on two widely used corruption and distortion benchmarks, ImageNet-C Hendrycks & Dietterich (2019) and ImageNet-PD Chhipa et al. (2024), to examine how GeSSL handles common corruptions, perturbations, and geometric distortions.

For ImageNet-C, we compute the mean Corruption Error (mCE following Hendrycks & Dietterich (2019), where lower values indicate better robustness. We evaluate SimCLR, BYOL, and MAE-ViT pretrained on ImageNet-1K, both in their original form and with GeSSL integrated. For ImageNet-PD, using the PD generation and evaluation protocol of Chhipa et al. (2024), we measure accuracy under multiple distortion severities. Higher accuracy indicates stronger robustness to geometric perspective changes. Both benchmarks evaluate models without fine-tuning, using their ImageNet-pretrained representations directly. The results are provided in Table 23 and Table 24. From the results, we can observe that across all methods and both benchmarks, GeSSL consistently improves robustness, i.e., around 4% on ImageNet-PD and 3% mCE reduction on ImageNet-C. These results demonstrate that GeSSL not only strengthens in-distribution discriminability and generalization but also enhances stability under severe corruptions and distortions.

**Comparison with non-GeSSL strong baselines.** To further evaluate the robustness and transferability of GeSSL-pretrained representations, we extend our distribution-shift experiments by incorporating a meta-learning–based baseline. While prior results on ImageNet-C and ImageNet-PD demonstrate that GeSSL improves corruption and distortion robustness, it remains important to compare GeSSL against methods that explicitly introduce meta-learning structures into SSL. Following the meta-learning episode construction described in Section 2, we implement an Unsupervised MAML baseline that adapts the MAML-style inner-outer loop to a label-free setting.

Table 23: Performance on ImageNet-PD using accuracy (higher is better).

| Method | Original | +GeSSL |
|--------|----------|--------|
| SimCLR | 75.6 | 79.4 |
| BYOL | 75.3 | 79.2 |
| MAE-ViT | 78.9 | 81.2 |

Table 24: Performance on ImageNet-C using mCE (lower is better).

| Method | Original | +GeSSL |
|--------|----------|--------|
| SimCLR | 77.9 | 74.2 |
| BYOL | 78.2 | 74.3 |
| MAE-ViT | 74.2 | 70.8 |

Table 25: Comparison between GeSSL, unsupervised meta-learning, and non-GeSSL strong SSL baselines on ImageNet-PD.

| Method | Accuracy |
|--------|----------|
| SimCLR | 75.6 |
| SimCLR+GeSSL | 79.4 |
| UnsupervisedMAML | 76.7 |
| DINO | 76.0 |
| CorInfoMax | 75.9 |

Each episode is treated as a pseudo-task formed from augmented sample pairs, enabling the baseline to perform gradient-based meta-updates in an unsupervised manner. We evaluate the following models on ImageNet-PD, using the same distortion generation and evaluation protocol described previously. The results in Table 25 show that GeSSL yields the strongest robustness, outperforming all non-GeSSL baselines, including methods specifically designed for stability and invariance, e.g., DINO, CorInfoMax.

## F.8 COMPARISON WITH LARGER-SCALE PRE-TRAINED MODEL

GeSSL is a pre-training learning mechanism whose core question is how to learn a good representation. In this paper, we define a "good" representation as universality. Indeed, larger-scale pre-trained models (e.g., large language models or multimodal large language models) demonstrate strong general-purpose capabilities. However, these models also suffer from issues such as hallucinations and weak performance on reasoning tasks (e.g., mathematical and code reasoning), which indicates they have not fully learned to generalize and transfer. Current large-model training paradigms are largely based on G-SSL (Touvron et al., 2023; Brown et al., 2020; Hoffmann et al., 2022; Zhang et al., 2022); in other words, these models are effectively trained to model discriminative signals during pretraining without explicitly accounting for generalization and transferability. GeSSL proposes an alternative pretraining scheme that aims to produce better large models. Training large-scale models with GeSSL therefore benefits their downstream generalization and transfer performance. To further validate this claim, we conduct experiments centered on CLIP, which can be regarded as a large-scale model (Radford et al., 2021). Following (Radford et al., 2021; Mu et al., 2022; Zhai et al., 2022), we compare CLIP, SLIP (Mu et al., 2022), LiT (Zhai et al., 2022), and CLIP-GeSSL on ImageNet using the ViT-S/16 backbone, and we adopt the three evaluation protocols used in SLIP. The results in **Table 26** show that, without changing model capacity or data volume, CLIP-GeSSL yields consistent and stable improvements, achieving better overall performance.

## F.9 STABILITY ANALYSES

To investigate whether GeSSL improves not only accuracy but also the stability of the training process, we conduct a comprehensive stability analysis inspired by prior works (Keskar et al., 2016; Hardt et al., 2016; Chaudhari et al., 2019; Foret et al., 2020) linking optimization stability, gradient noise, and generalization. By the gradient-noise-scale framework of (McCandlish et al., 2018), gradient variance can be regarded as a measurable indicator of training stability. Using SimCLR as the base framework, we compare four meta-regularization baselines, i.e., ICL-MSR (Qiang et al., 2022), MetAug (Li et al., 2022a), MCR (Guo et al., 2024), and our method by training all models on ImageNet-1K while recording per-step gradient norms, their variances, seed-to-seed performance variability, and final linear-probe accuracy. The results are shown in **Table 27** and **Figure 7**. Across

Table 26: Comparison with Larger-scale Pre-trained Model.

| Method | 0-shot | Linear | Finetuned |
|---|---|---|---|
| CLIP [5] | 32.7 | 59.3 | 78.2 |
| SLIP [6] | 38.3 | 66.4 | 80.3 |
| LiT [8] | 39.0 | 65.9 | 81.4 |
| CLIP-GeSSL | 39.2 | 67.1 | 81.9 |

Table 27: Training stability analysis and ImageNet linear-probe accuracy.

| Method | Mean GradNorm | GradNorm Var | Linear Probe (%) |
|---|---|---|---|
| SimCLR | $0.045 \pm 0.012$ | 2.1e-4 | 69.9 |
| SimCLR + ICL-MSR | $0.042 \pm 0.010$ | 1.8e-4 | 70.8 |
| SimCLR + MetAug | $0.041 \pm 0.009$ | 1.7e-4 | 71.0 |
| SimCLR + MCR | $0.039 \pm 0.008$ | 1.4e-4 | 71.6 |
| SimCLR + GeSSL | $\mathbf{0.033 \pm 0.006}$ | **9.2e-5** | **72.8** |

all evaluations, GeSSL consistently exhibits the lowest gradient means and variances throughout training, indicating smoother and more stable optimization dynamics, and also shows the smallest sensitivity to random initialization. Correspondingly, GeSSL achieved the best performance. The loss trajectory of GeSSL also decreases steadily, further supporting its robustness and stability. Altogether, these observations demonstrate that GeSSL produces more stable training behavior and is less susceptible to poor local minima, aligning with theoretical insights and yielding superior downstream generalization.

# G  DETAILS OF ABLATION STUDY

In this section, we introduce the experimental details and more comprehensive analysis of the ablation studies (Subsection 5.2).

## G.1  MODEL EFFICIENCY

This ablation study explores the efficiency of self-supervised models before and after applying GeSSL. Specifically, we choose five baselines, including SimCLR (Chen et al., 2020a), MOCO (Chen et al., 2020b), BYOL (Grill et al., 2020), Barlow Twins (Zbontar et al., 2021), and SwAV (Caron et al., 2020). Then, we evaluate the accuracy, training hours, and parameter size of these models on STL-10 before and after applying our proposed GeSSL. We use the same linear evaluation setting as in Section 5.1 of the main text. Finally, we plot the trade-off scatter plot by recording the average values of five runs. The results are shown in Figure 2 of the main text, where the horizontal axis represents the training hours and the vertical axis represents the accuracy. The center

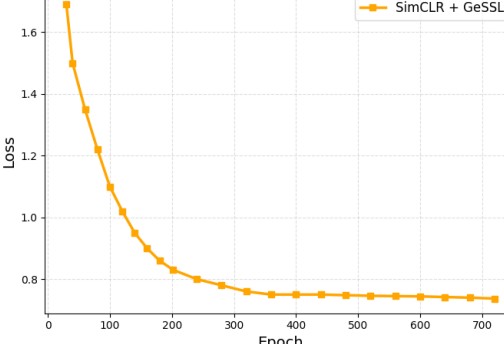

Figure 7: Training loss curve of SimCLR with GeSSL.

Table 28: Training cost per epoch of SSL models.

| Methods | Training Cost per Epoch (s) |
|---|---|
| SimCLR (Chen et al., 2020a) | 12.8 |
| MOCO (Chen et al., 2020b) | 16.9 |
| SimCLR + GeSSL | 9.6 |
| MOCO + GeSSL | 12.0 |

Table 29: Model analysis including parameter size, training time, and performance.

| Methods | Memory Footprint (MiB) | Parameter Size (M) | Training Time (h) | Accuracy (%) |
|---|---|---|---|---|
| SimCLR | 2415 | 23.15 | 4.15 | 90.5 |
| MOCO | 2519 | 24.01 | 4.96 | 90.9 |
| BYOL | 2691 | 25.84 | 6.98 | 91.9 |
| BarlowTwins | 2477 | 23.15 | 5.88 | 90.3 |
| SwAV | 2309 | 22.07 | 4.45 | 90.7 |
| SimCLR+GeSSL | 2784 | 26.21 | 3.36 | 93.4 |
| MOCO+GeSSL | 2912 | 27.20 | 4.23 | 94.6 |
| BYOL+GeSSL | 2875 | 28.01 | 5.70 | 94.8 |
| BarlowTwins+GeSSL | 2856 | 27.11 | 5.39 | 94.2 |
| SwAV+GeSSL | 3012 | 28.61 | 3.96 | 93.2 |

of each circle represents the result of the training time and accuracy of each model, and the area of the circle represents the parameter size. The numerical results of this experiment are shown in Table 29. From the results, we can see that: (i) GeSSL can significantly improve the performance and computational efficiency of self-supervised learning models; (ii) our designed self-motivated target achieves the goal of guiding the model update toward universality with few samples and fast adaptation; (iii) although GeSSL optimizes based on bi-level optimization, the impact of the increased parameter size of GeSSL is negligible.

Note that although the optimization method used by GeSSL is more complex, one of its core goals is to accelerate model convergence, i.e., achieve greater performance improvement per unit of time. This does not imply that GeSSL always requires fewer epochs to reach the optimal result. In fact, GeSSL uses approximate implicit differentiation with finite difference (AID-FD) for updates instead of conventional explicit second-order differentiation (as mentioned in Appendix G.4). Moreover, GeSSL constructs a self-motivated target that guides the model to optimize more effectively in a specific task. Therefore, the efficiency improvement is reflected in the computational efficiency and effectiveness of updates per epoch, rather than simply reducing the total number of epochs. Furthermore, to verify whether the efficiency improvement is attributable to a single epoch, we separately measured the computational overhead of SSL baseline algorithms after integrating GeSSL for a single epoch. The results, presented in Table 28, demonstrate that with a consistent batch size, GeSSL enhances the computational efficiency and the effectiveness of updates per epoch for the SSL baseline algorithms.

## G.2 ABLATION STUDY OF $\mathcal{L}_{disc}$

To evaluate the impact of $\mathcal{L}_{disc}$, we design a series of experiments. $\mathcal{L}_{disc}$ is intended to enhance discriminative power by enforcing constraints that sharpen the SSL model's decision boundaries. We therefore visualize classification performance before and after adding $\mathcal{L}_{disc}$. Using SimCLR, BYOL, MoCo, and MAE as baselines, we randomly select 10 classes from ImageNet-100 (100 samples per class) and compare each model with and without $\mathcal{L}_{disc}$. As shown in Figure 8, introducing $\mathcal{L}_{disc}$ produces noticeably sharper class boundaries, demonstrating its effectiveness in improving model discriminability.

Besides the above results, to isolate and better understand the effect of the discriminative regularization term $\mathcal{L}_{disc}$, we also conduct ablation experiments that incorporate $\mathcal{L}_{disc}$ directly into the training objective without bi-level optimization. This allows us to assess whether $\mathcal{L}_{disc}$ alone can improve representation quality across different SSL paradigms. Specifically, we conduct ablation experiments using SimCLR, BYOL, and MAE as baselines on ImageNet (for the unsupervised learning setting) and COCO (for the transfer learning setting) to evaluate $\mathcal{L}_{disc}$ without bi-level

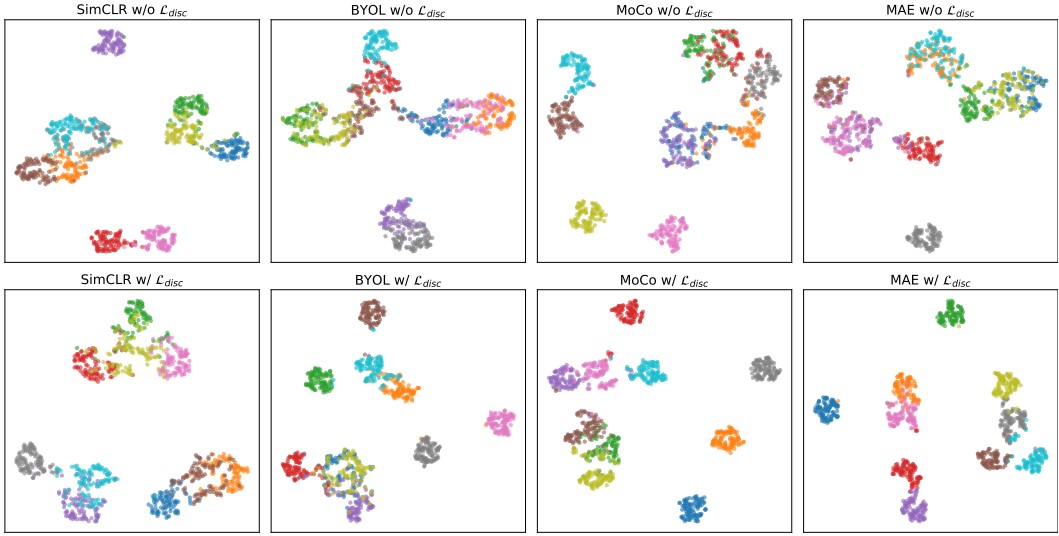

Figure 8: Ablation study of $\mathcal{L}_{disc}$. We perform t-SNE visualization to evaluate the classification performance of the SSL model before and after introducing $\mathcal{L}_{disc}$.

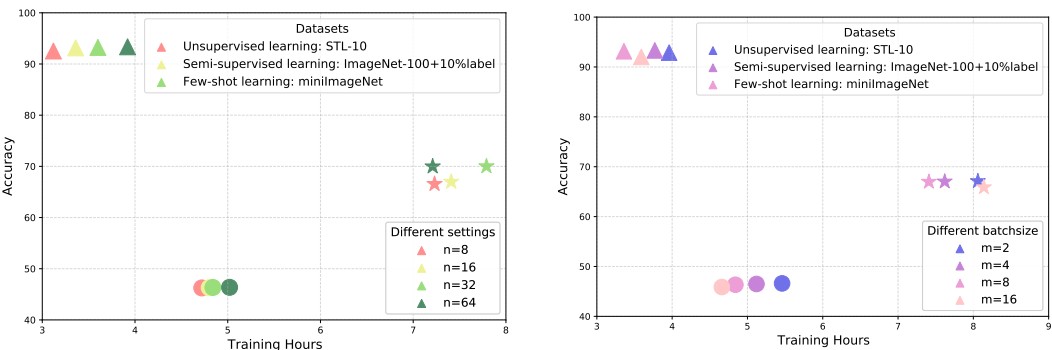

Figure 9: Ablation study of the number of pairs.

Figure 10: Ablation study of the batchsize.

optimization. Concretely, we incorporate $\mathcal{L}_{disc}$ as a regularization term into the loss computed for each mini-batch, and optimize this term across multiple mini-batches. The results in **Tables 30 and 31** show that adding $\mathcal{L}_{disc}$ produces consistent performance improvements, thereby validating its effectiveness.

### G.3 MORE EXPERIMENTS OF PARAMETER SENSITIVITY

Considering that our framework updates the self-supervised model $f_\theta$ in GeSSL based on $M$ tasks simultaneously, the number of sampled samples per batch of self-supervised learning directly determines the class diversity of the data in the task. In this section, we further conduct ablation experiments on the number of pairs within each batch and the batch size (the number of tasks) that are learned simultaneously.

Specifically, we choose the commonly used STL-10 for unsupervised learning, ImageNet with 10% label for semi-supervised learning, and miniImageNet for few-shot learning, and evaluate the performance of SimCLR + GeSSL under different batch sizes and different $n$ values. Figure 9 shows the impact of different number of pairs for SSL. The results show that SimCLR + GeSSL always outperforms SimCLR under any batch size. A larger batch size leads to a slightly larger performance improvement for SimCLR + GeSSL, but also increases the computational resource consumption.

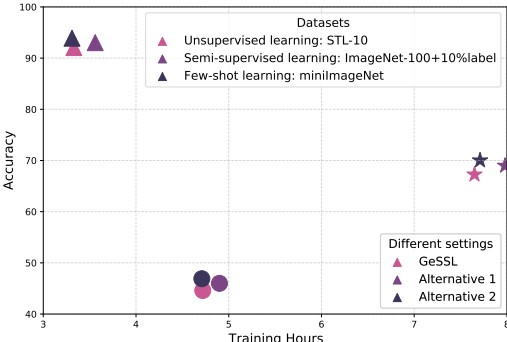 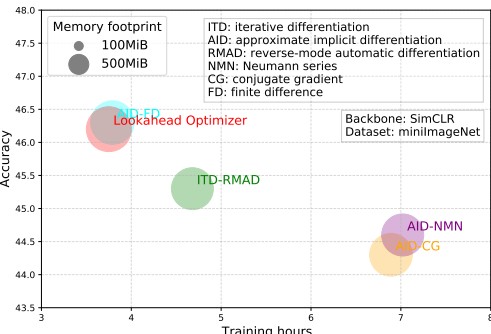

Figure 11: Evaluation of bi-level optimization.   Figure 12: Implementation of optimization.

Table 30: Effect (Accuracy) of $\mathcal{L}_{disc}$ (without bi-level optimization) on ImageNet.

| Method | w/o $\mathcal{L}_{disc}$ | w/ $\mathcal{L}_{disc}$ |
|--------|----------|----------|
| SimCLR | 70.6 | 71.4 |
| BYOL | 73.4 | 74.9 |
| MAE | 75.2 | 76.9 |

Therefore, in this study, we build tasks based on images with a batch size of $n = 16$ or $n = 32$. Figure 10 shows the impact of the batchsize for the outer-loop optimization. The results indicate that $m = 8$ is a better trade-off between model accuracy and time consumption. In the setting of our GeSSL, we also choose $m = 8$ as the hyperparameter setting.

In addition, considering that GeSSL updates every $m$ mini-batches, we evaluate the baseline performance under $m\times$ the original batch size. Specifically, we adopt the same experimental setup as in Figure 2, with the only difference being that we increase the batch size of the SimCLR baseline by a factor of $m$ and record the results. The results are shown in Table 32, which indicates that the performance of SimCLR, after converging with the larger training data, remains largely unchanged and still inferior to GeSSL.

G.4   EVALUATION AND IMPLEMENTATION OF THE BI-LEVEL OPTIMIZATION

As mentioned in Subsection 5.1, to assess the advantages of our bi-level optimization, we compare its performance against two alternatives: (i) jointly optimizing the inner and outer objectives in a single stage; and (ii) training a distinct $f'$ for each mini-batch. The results in Figure 11 demonstrate that our bi-level optimization (Subsection 3.2) achieves state-of-the-art performance.

The model of GeSSL is updated based on bi-level optimization, and the model gradients for each level are obtained by combining the optimal response Jacobian matrices through the chain rule. In practical applications, multi-level gradient computation requires a lot of memory and computation (Choe et al., 2022), so we hope to introduce a more concise gradient backpropagation and update method to reduce the computational complexity. Specifically, we consider two types of gradient update methods, including iterative differentiation (ITD) (Finn et al., 2017) and approximate implicit differentiation (AID) (Grazzi et al., 2020). We provide implementations of four popular ITD/AID algorithms, including ITD with reverse-mode automatic differentiation (ITD-RMAD) (Finn et al., 2017), AID with Neumann series (AID-NMN) (Lorraine et al., 2020), AID with conjugate gradient (AID-CG) (Rajeswaran et al., 2019), and AID with finite difference (AID-FD) (Liu et al., 2018). We also choose the recently proposed optimizer, i.e., Lookahead (Zhang et al., 2019) for comparison. We denote the the upper-level parameters and the lower-level parameters as $\theta$ and $\phi$, respectively. All the way of gradient update of the bi-level optimization are as follows:

**ITD-RMAD** (Finn et al., 2017), ITD with reverse-mode automatic differentiation applies the implicit function theorem to the lower-level optimization problem and computes the gradients of the

Table 31: Effect (AP) of $\mathcal{L}_{disc}$ (without bi-level optimization) on COCO.

| Method | w/o $\mathcal{L}_{disc}$ | w/ $\mathcal{L}_{disc}$ |
|--------|--------------------------|-------------------------|
| SimCLR | 38.1 | 39.0 |
| BYOL | 37.9 | 38.6 |
| MAE | 39.2 | 40.8 |

Table 32: Performance on for a large batchsize.

| Methods | Accuracy | Training Cost |
|---------|----------|---------------|
| SimCLR (Chen et al., 2020a) | 90.8 | 5.2 |
| SimCLR + GeSSL | 93.8 | 3.9 |

upper-level objective with respect to the upper-level parameters using reverse-mode automatic differentiation. The update process is as follows:

- Solve the lower-level optimization problem $\phi^* = \arg\min_\phi L(\phi, \theta)$ using gradient descent.
- Compute the gradient of the upper-level objective $g(\theta) = F(\phi^*, \theta)$ with respect to $\theta$ using reverse-mode automatic differentiation:

$$\nabla_\theta g(\theta) = \nabla_\theta F(\phi^*, \theta) - \nabla_\phi F(\phi^*, \theta)^T (\nabla_\phi L(\phi^*, \theta))^{-1} \nabla_\theta L(\phi^*, \theta) \tag{22}$$

- Update the upper-level parameters using gradient descent or other methods: $\theta \leftarrow \theta - \alpha \nabla_\theta g(\theta)$.

**AID-NMN** (Lorraine et al., 2020), AID with Neumann series, approximates the inverse of the Hessian matrix of the lower-level objective using a truncated Neumann series expansion and computes the gradients of the upper-level objective with respect to the upper-level parameters using forward-mode automatic differentiation. The update process is as follows:

- Solve the lower-level optimization problem $\phi^* = \arg\min_\phi L(\phi, \theta)$ using gradient descent.
- Compute the gradient of the upper-level objective $g(\theta) = F(\phi^*, \theta)$ with respect to $\theta$ using forward-mode automatic differentiation:

$$\nabla_\theta g(\theta) = \nabla_\theta F(\phi^*, \theta) - \nabla_\phi F(\phi^*, \theta)^T (\nabla_\phi L(\phi^*, \theta))^{-1} \nabla_\theta L(\phi^*, \theta)$$
$$\approx \nabla_\theta F(\phi^*, \theta) - \nabla_\phi F(\phi^*, \theta)^T \sum_{k=0}^K (-1)^k (\nabla_\phi^2 L(\phi^*, \theta))^k \nabla_\theta L(\phi^*, \theta) \tag{23}$$

where $K$ is the truncation order of the Neumann series.

- Update the upper-level parameters using gradient descent or other methods: $\theta \leftarrow \theta - \alpha \nabla_\theta g(\theta)$.

**AID-CG** (Rajeswaran et al., 2019), AID with conjugate gradient, solves a linear system involving the Hessian matrix of the lower-level objective using the conjugate gradient algorithm and computes the gradients of the upper-level objective with respect to the upper-level parameters using forward-mode automatic differentiation. The update process is as follows:

- Solve the lower-level optimization problem $\phi^* = \arg\min_\phi L(\phi, \theta)$ using gradient descent or other methods.
- Compute the gradient of the upper-level objective $g(\theta) = F(\phi^*, \theta)$ with respect to $\theta$ using forward-mode automatic differentiation:

$$\nabla_\theta g(\theta) = \nabla_\theta F(\phi^*, \theta)$$
$$-\nabla_\phi F(\phi^*, \theta)^T (\nabla_\phi L(\phi^*, \theta))^{-1} \nabla_\theta L(\phi^*, \theta) \approx \nabla_\theta F(\phi^*, \theta) \tag{24}$$
$$-\nabla_\phi F(\phi^*, \theta)^T v$$

where $v$ is the solution of the linear system $(\nabla_\phi^2 L(\phi^*, \theta))v = \nabla_\theta L(\phi^*, \theta)$ obtained by the conjugate gradient algorithm.

- Update the upper-level parameters using gradient descent or other methods: $\theta \leftarrow \theta - \alpha \nabla_\theta g(\theta)$.

**AID-FD** (Liu et al., 2018), AID with finite difference, approximates the inverse of the Hessian matrix of the lower-level objective using a finite difference approximation and computes the gradients of the upper-level objective with respect to the upper-level parameters using forward-mode automatic differentiation. The update process is as follows:

- Solve the lower-level optimization problem $\phi^* = \arg\min_\phi L(\phi, \theta)$ using gradient descent or other methods.
- Compute the gradient of the upper-level objective $g(\theta) = F(\phi^*, \theta)$ with respect to $\theta$ using forward-mode automatic differentiation:

$$
\begin{aligned}
\nabla_\theta g(\theta) &= \nabla_\theta F(\phi^*, \theta) \\
&\quad - \nabla_\phi F(\phi^*, \theta)^T (\nabla_\phi L(\phi^*, \theta))^{-1} \nabla_\theta L(\phi^*, \theta) \\
&\approx \nabla_\theta F(\phi^*, \theta) \\
&\quad - \nabla_\phi F(\phi^*, \theta)^T \frac{\nabla_\theta L(\phi^* + \epsilon \nabla_\theta L(\phi^*, \theta), \theta) - \nabla_\theta L(\phi^*, \theta)}{\epsilon}
\end{aligned}
\tag{25}
$$

where $\epsilon$ is a small positive constant for the finite difference approximation.

- Update the upper-level parameters using gradient descent or other methods: $\theta \leftarrow \theta - \alpha \nabla_\theta g(\theta)$.

**Lookahead** (Zhang et al., 2019) introduces a novel approach to optimization by maintaining two sets of weights: the fast and the slow weights. The fast weights, $\theta_{\text{fast}}$, are updated frequently through standard optimization techniques, while the slow weights, $\theta_{\text{slow}}$, are updated at a lesser frequency. The key formula that updates the slow weights is given by:

$$
\theta_{\text{slow}} \leftarrow \theta_{\text{slow}} + \alpha(\theta_{\text{fast}} - \theta_{\text{slow}})
\tag{26}
$$

where $\alpha$ is a hyperparameter controlling the step size. This method aims to stabilize training and ensure consistent convergence.

The results shown in Figure 4 of the main text demonstrate that approximate implicit differentiation with finite difference also achieves optimal results on the SSL model. Our optimization process is also based on this setting.

# H MORE DISCUSSION

## H.1 DIFFERENCES BETWEEN GESSL AND META-LEARNING

In the main text, we have illustrated the differences between GeSSL and meta-learning and the advantages of GeSSL. In this section, we further elaborate on this and list different meta-learning methods for comparison.

Meta-learning (Finn et al., 2017; Wang et al., 2024; Snell et al., 2017), often referred to as "learning to learn", has emerged as a prominent approach to improve the efficiency and adaptability of machine learning models, especially in scenarios with limited data. The fundamental idea behind meta-learning is to train models that can rapidly adapt to new tasks with minimal data by leveraging prior experiences gained from a range of related tasks.

Few-shot Learning (Khodadadeh et al., 2019; Jang et al., 2023): One of the primary areas where meta-learning has demonstrated substantial impact is in few-shot learning. Methods like Model-Agnostic Meta-Learning (MAML) (Finn et al., 2017) aim to find a set of model parameters that are sensitive to changes in the task, allowing for quick adaptation to new tasks with just a few examples. Variants of MAML, such as First-Order MAML (FOMAML) and Reptile (Nichol & Schulman, 2018), reduce the computational complexity of the original algorithm while maintaining competitive performance.

Metric-based Approaches: Metric-based meta-learning methods, such as Matching Networks (Sung et al., 2018) and Prototypical Networks (Snell et al., 2017), learn an embedding space where similar tasks are closer together. These models perform classification by comparing the distance between new examples and a few labeled instances (support set) in this learned space, achieving remarkable results in few-shot classification tasks.

Memory-augmented Networks: Another line of research in meta-learning explores the use of external memory structures to facilitate rapid adaptation. Santoro et al introduced Memory-Augmented Neural Networks (MANNs) (Santoro et al., 2016) that use an external memory to store and retrieve information about past tasks, enabling the model to perform well even in tasks with highly variable distributions.

Gradient-based Meta-learning: Beyond MAML, other gradient-based methods such as Meta-SGD (Li et al., 2017) and Learning to Learn with Gradient Descent have been proposed. These methods modify the way gradients are used during the training of the model, either by learning the initial parameters (as in MAML) or by learning the learning rates for different parameters, allowing for more efficient adaptation.

Bayesian Meta-learning: Bayesian approaches to meta-learning, such as Bayesian MAML (Zhang et al., 2021), offer a probabilistic framework for capturing uncertainty and improving generalization to new tasks. These methods have been particularly useful in scenarios where task distributions are diverse, and the model needs to account for uncertainty in task inference.

Meta-learning for Reinforcement Learning: Meta-learning has also been successfully applied in the domain of reinforcement learning (RL). Methods such as Meta-RL (Yu et al., 2020) aim to train agents that can quickly adapt to new environments by leveraging the experience gained in previous tasks. These approaches have shown promise in enabling RL agents to solve tasks with minimal exploration, a crucial aspect for real-world applications where exploration can be costly or risky.

In summary, meta-learning has rapidly evolved as a versatile framework that enhances the ability of models to adapt quickly to new tasks, and operate efficiently in dynamic environments. Compared meta-learning with the proposed GeSSL, we can see that the main difference between them located in the way to model discriminability and generalizability. For more details, please refer to the last paragraph of Section 3.2.

**How GeSSL differs fundamentally from approaches that directly transplant meta-learning paradigms into SSL?** We would like to provide an outline for illustration: (i) Objective: Traditional meta-learning focuses on "fast adaptation to supervised downstream tasks", using a small labelled support set and query set for quick fine-tuning. GeSSL, by contrast, starts from the question "what is a good representation?", i.e., Universality, which explicitly encompasses discriminability, generalizability, and transferability, and encodes these properties directly into the training objective. The design motivations differ: meta-learning is built for rapid adaptation, while GeSSL is designed to learn universal, transferable self-supervised representations. Simply porting meta-learning to self-supervised settings typically reuses multi-task architectures or additional networks to apply episodic meta-learning Liu et al. (2019); Lin et al. (2021), but lacks dedicated mechanisms and theoretical guarantees for "stable learning of transferable representations under no labels". (ii) Task construction: Meta-learning usually relies on a small number of accurately labelled samples; existing attempts to apply meta-learning to SSL therefore construct labels and tasks explicitly via label-generation networks Liu et al. (2019). GeSSL, however, operates entirely within the self-supervised paradigm and must generate pseudo-labels itself (i.e., anchors from augmentations or reconstructions). Because GeSSL runs in an unlabeled regime, a naive application of episodic meta-learning is vulnerable to pseudo-label noise. To address this, GeSSL introduces the discriminative term $\mathcal{L}_{disc}$ and integrates it into training in a learnable way to suppress noise, a unique implementation to GeSSL. (iii) Method and architecture: Many meta-learning methods learn task-specific adapters (different adapted models per task). GeSSL instead learns a single, unified proxy adapter $f'$ that is shared across all mini-batches. This shared design better captures task-invariant shared structure, i.e., causal-like representations, thereby improving cross-task generalization and transfer. (iv) Empirical results: In Table 4 and Table 13, we compare GeSSL against methods that apply meta-learning to SSL (e.g., MetaSVEBM, MetaGMVAE, PsCo, etc.). These results demonstrate GeSSL's effectiveness, where it achieves SOTA performance.

**Comparison of GeSSL and unsupervised meta-learning and episodic SSL.** The unsupervised meta-learning Ni et al. (2021); Jang et al. (2023) and episodic SSL Lee et al. (2023) are similar in spirit to supervised meta-learning, but they typically generate (pseudo) labels via clustering or auxiliary generative networks to enable episodic/task-style learning. Below we give an outline summarizing how GeSSL differs from, and in our view improves upon, unsupervised meta-learning and episodic SSL: (i) Objective. Episodic SSL and unsupervised meta-learning primarily target discriminability or fast few-shot adaptation (i.e., task-style training to boost few-shot accuracy or high performance on specific transfers). GeSSL, by contrast, explicitly optimizes Universality, jointly modelling discriminability, generalizability, and transferability, and embeds these properties directly into the pretraining objective. (ii) Task construction. Episodic methods center on N-way K-shot episode sampling and typically rely on clustering or label-generation networks to provide (pseudo) labels; their performance therefore depends on the quality of those supervisory signals and can be sensitive to pseudo-label noise. GeSSL partitions each mini-batch into support/query pairs and, for G-SSL and D-SSL respectively, uses masking or contrastive view generation. This within-batch pairing preserves task-style supervision while remaining easy to integrate with standard pretraining pipelines. Crucially, because GeSSL operates without manual labels, we introduce a dedicated discriminative loss $\mathcal{L}_{disc}$ together with a bi-level constraint to suppress pseudo-label noise and robustly extract discriminative signals. (iii) Model architecture. Unsupervised meta-learning and episodic SSL approaches often learn task-specific adapters or perform local inner-loop updates (i.e., different adapted models per task) to maximize fast adaptation. GeSSL does not allocate per-task parameters; instead it learns a single unified model that is shared across mini-batches to emulate adaptation behavior. This shared design promotes extraction of task-invariant structure (causal representations) (Sheth et al., 2022; Li et al., 2022b; Mahajan et al., 2021), thereby improving cross-task generalization and transfer. (iv) In Section 5.1 and Appendix F.3, we empirically compare GeSSL against unsupervised meta-learning and episodic SSL baselines (e.g., PsCo, UMTRA, MetaSVEBM, and CACTUs). The results in Table 4 and Table 13 show that, across both standard and cross-domain transfer scenarios, GeSSL consistently yields performance gains and achieves SOTA results.

## H.2 UNDERSTANDING TASK STRUCTURE AND TRANSFERABILITY IN GeSSL

our view of SSL is not organized around domains or ground-truth labels, but adopts a finer-grained perspective. We explain this in steps:

**Step 1: For SSL, different augmentations of the same sample belong to the same class, while augmentations of different samples belong to different classes, this is distinct from the sample's ground-truth label.** First, as stated in Section 2 (first and second paragraphs), both D-SSL and G-SSL reorganize data into pairs: each pair contains an augmented sample and an anchor, and each pair is produced from the same original example. Second, also in Section 2 (second paragraph), the objective of D-SSL methods (e.g., SimCLR, BYOL, Barlow Twins) can be written as an alignment term plus a regularization term, while G-SSL objectives are typically expressed as alignment terms. The alignment term encourages the augmented sample and its anchor to align; regularization may be implemented in various ways (e.g., enforcing a uniform distribution over augmentations, gradient clipping, etc.). Third, if we regard the anchor as a **cluster center**, the alignment term enforces that samples of the same class cluster tightly, i.e., intra-class compactness. Consequently, we can view an SSL dataset as being partitioned into many classes where each class corresponds to a pair. **Importantly, this partitioning is not based on domain or ground-truth labels.** This viewpoint motivates mining representational universality via SSL: instead of being tied to a particular domain or ground-truth label, SSL assigns labels at the pair level, where each pair's anchor serves as the class center. This is a reasonable assumption since no two raw samples are identical; such pair-level labeling is broader than semantic labels from ground truth.

**Step 2: SSL training can be interpreted as task-based training, where each mini-batch constitutes a task.** From Step 1, each pair defines a class and different mini-batches contain different sets of pairs; thus each mini-batch can be regarded as a distinct task.

**Step 3: How the proposed GeSSL models transferability.** Please see "Explanation for Transferability" and "Comparison of GeSSL and Meta-Learning" of Section 3.2 for details.

**Step 4: Why we introduce an additional term to enhance GeSSL's discriminability.** From Step 2, treating every pair in a mini-batch as a different class is a very strong constraint. We relax this

constraint by allowing several pairs within a mini-batch to belong to the same class, please see "Explanation for Discriminability" of Section 3.2 for details.

**Step 5: Empirical validation.** We already performed transfer experiments across multiple domains and tasks where training and test domains exhibit clear distribution shifts. Examples include: training on CIFAR-100 and testing on CIFAR-10, Flower102, Food101, and Aircraft (Table 11); standard VOC and COCO detection/segmentation protocols (Table 3); using pretrained features for five video-tracking tasks on UniTrack (Table 12); and training on miniImageNet then testing on CUB, Cars, Places, etc. (Table 13). The results show that GeSSL achieves an average improvement of over 3%, demonstrating its effectiveness. We have also compared GeSSL against meta-learning baselines based on standard meta-learning and multi-dataset pretraining (i.e., standard and cross-domain few-shot learning in Section 5.1 and Appendix F.3). The nine datasets and benchmarks involved are inherently multi-task. Experimental results in Table 4 and Table 13) confirm GeSSL's superiority, achieving SOTA performance.

### H.3 THEORETICAL MOTIVATION OF $\mathcal{L}_{disc}$ AND $\mathcal{L}_{ssl}$

The intention of this paper is not to derive one single, unique objective function from theory, but to use theory to justify our definition of what constitutes a good representation.

More concretely, firstly, as stated in the Introduction, our central question is how to learn a good representation, rather than how to design yet another heuristic SSL loss. Many existing SSL methods can be summarized as "do what seems to lead to a good representation", by empirically designing various pretext tasks. In contrast, we deliberately shift the focus from "which factors or tricks to adopt in SSL" to "how to explicitly characterize what a good representation or model should be".

Then, based on this perspective, we propose an explicit definition of a good representation: it should be universal, meaning that it jointly satisfies discriminability, generalizability, and transferability. This definition is, in essence, our modeling assumption about good representations and is to some extent subjective. To make this definition more solid and convincing, we provide a theoretical analysis (Theorem 4.1). The theorem shows that, under certain conditions, as long as the training objective induces representations that are discriminative, generalizable, and transferable, the test error can be reduced. This establishes the following logical link: "good representation = discriminability + generalizability + transferability" to "such representations lead to smaller test error". In this way, the theorem does not fix a specific loss form, but it supports the reasonableness and usefulness of our definition of a good representation.

Therefore, the role of the theoretical section is to answer the question "Is our definition of a good representation reasonable and theoretically supported?", rather than to derive a single closed-form objective. Any objective that can induce representations with discriminability, generalizability, and transferability is admissible under our framework. The specific objective used in our method (including $\mathcal{L}_{ssl}$ and $\mathcal{L}_{disc}$) should be viewed as one concrete instantiation that follows these principles, rather than the only possible choice.

### H.4 INTERPRETING SSL OBJECTIVES AS IMPLICIT CLASSIFICATION PROCESSES

Understanding how different self-supervised learning paradigms induce task structure is essential for analyzing the behavior of representation learning methods. While contrastive approaches naturally resemble classification through positive–negative sample discrimination, we show that self-distillation methods (e.g., BYOL, SimSiam) and generative reconstruction methods (e.g., MAE) can also be interpreted through the unified lens of alignment and regularization, which in turn leads to an implicit classification view at the mini-batch level. We would like to explain this through the following three steps.

**Step 1: Reforming SSL from the perspective of alignment and regularization.** Besides the contrastive SSL (MoCo, SimCLR), the self-distillation methods such as BYOL of D-SSL employ an online and a target network, where the training loss aligns the normalized representations of augmented views. Although expressed as a regression loss without explicit negatives, its objective encourages the online representation to match the target representation, while the exponential moving average update constrains gradient propagation. This structure can be reformulated as an optimization problem consisting of an alignment term paired with a constraint that regularizes the evolution of the

target network. Generative SSL methods such as MAE follow a similar alignment–regularization structure. The heavy masking of input tokens encourages the encoder to produce latent representations that capture essential structure, while the reconstruction objective aligns these representations with the original sample through decoder prediction. Masking thus acts as a strong regularizer, forcing the model to encode useful and compressive information rather than trivial pixel-level patterns. In the following, we provide three examples, i.e., SimCLR, BYOL and MAE.

**SimCLR** can be considered a learning method based on anchors. Firstly, SimCLR assumes that augmented samples sharing the same ancestor form a positive pair, e.g., $x_i^1$ and $x_i^2$ constitute such a pair. Secondly, the core modeling idea of SimCLR is as follows: one augmented sample is selected as the anchor; if another sample is a positive example relative to the anchor, it is pulled closer to the anchor, whereas if it is a negative example, it is pushed away from the anchor. Typically, the sample that belongs to the same pair as the anchor is considered a positive sample, while all others are treated as negative samples. Finally, SimCLR implements the above idea by minimizing the Noise Contrastive Estimation (NCE) loss Gutmann & Hyvärinen (2010), whose specific objective is formulated as follows:

$$\mathcal{L}_{\text{NCE}} = \sum_{i=1,l=1}^{i=N,l=2} - \log \frac{\exp\left[\frac{\text{sim}\left(z_i^l, z_i^{3-l}\right)}{\tau}\right]}{\exp\left[\frac{\text{sim}\left(z_i^l, z_i^{3-l}\right)}{\tau}\right] + \sum_{j=1,j\neq i,k=1}^{j=N,k=2} \exp\left[\frac{\text{sim}\left(z_i^l, z_j^k\right)}{\tau}\right]}, \tag{27}$$

where $\tau$ represents the temperature hyperparameter, $z_i^j = f_p(z_i^j)$, $f_p$ represents the projection head, and $\text{sim}(x,y) = x^{\text{T}}y/\|x\|\|y\|$ represents the $l_2$-normalized cosine similarity between $x$ and $y$. According to Theorem 1 in Wang & Isola (2020), as $N \to \infty$, Equation (27) can be rewritten as:

$$L_{\text{NCE}} = -\frac{1}{\tau} \mathbb{E}_{(z_i,z_j)\sim p_{pos}} \text{sim}\left(z_i, z_j\right) + \mathbb{E}_{z^i \sim p_{data}} \left[\log \mathbb{E}_{z^j \sim p_{data}} \exp\left(\frac{\text{sim}\left(z^i, z^j\right)}{\tau}\right)\right], \tag{28}$$

where $p_{pos}$ represents the distribution of positive pairs, e.g., the distribution of pair $(z_i^1, z_i^j)$, and $p_{data}$ represents the data distribution. Then, based on the von Mises-Fisher (vMF) kernel density estimation (KDE) Cohn & Kumar (2007); Borodachov et al. (2019), the second term in Equation (28) can empirically be considered equal to:

$$\begin{aligned} \mathbb{E}_{x\sim p_{data}} &\left[\log \mathbb{E}_{x^- \sim p_{data}} \left[e^{-\text{F}(x)^{\text{T}}\text{F}\left(x^-\right)/\tau}\right]\right] \\ &= \frac{1}{N} \sum_{i=1}^{N} \log\left(\frac{1}{N} \sum_{j=1}^{N} e^{-\text{F}(x_i)^{\text{T}}\text{F}(x_j)/\tau}\right) \\ &= \frac{1}{N} \sum_{i=1}^{N} \log p_{\text{vMF-KDE}}\left(\text{F}\left(x_i\right)\right) + \log Z_{\text{vMF}} \\ &\triangleq -H\left(\text{F}\left(x\right)\right), \end{aligned} \tag{29}$$

where $\text{F}(x) = f_p(f(x))$, $p_{\text{vMF-KDE}}$ is the KDE-based vMF kernel with $\kappa = \tau^{-1}$, $Z_{\text{vMF}}$ is the vMF normalization constant for $\kappa = \tau^{-1}$, and $H(\cdot)$ represents the entropy estimator. Therefore, from Equation (28), the first term can be interpreted as enforcing alignment between the two samples in a pair. From Equation (29), the second term in Equation (28) can be understood as maximizing the entropy of the data distribution, effectively encouraging all augmented samples to follow a uniform distribution.

**BYOL** is a D-SSL method that does not take negative samples into account. It employs two networks: the online network and the target network, both equipped with the same feature extraction module $f$ and projection head module $f_p$. However, the online network has an additional regression module $f_r$ that the target network lacks. The objective of BYOL can be expressed as:

$$\mathcal{L}_{\text{BYOL}} = \sum_{i=1}^{N} \sum_{j=1}^{2} \|f_r(\bar{z}_i^j) - \bar{z}_i^{3-j}\|_2^2, \tag{30}$$

where $z_i^j = f_p(f(x_i^j))$ and $\bar{z}_i^j = z_i^j / \|z_i^j\|_2$. The target network is referred to as $f_{target}$, while the online network is referred to as $f_{online}$. Subsequently, $f_{online}$ is updated through backpropagation, while $f_{target}$ is updated as follows:

$$f_{target} \leftarrow \pi f_{target} + (1-\pi)f_{online}, \tag{31}$$

where $\pi \in [0, 1]$ denotes the target decay rate. Equation (30) can be interpreted as aligning, while Equation (31) can be considered as constraining the gradient backpropagation based on the training data. Hence, from the perspective of aligning and constraining, the objective of BYOL can be rewritten as:

$$\begin{aligned}
&\min_{f_{online}, f_{target}} \quad \mathcal{L}_{\text{BYOL}} + \text{Sg}(\Phi(f_{target})) \\
&s.t. \quad f'_{online} = \min_{f_{online}} \mathcal{L}_{\text{BYOL}} \\
&\quad f_{online} = \{f_p, f\}, f_{target} = \{f'_p, f'\} \\
&\quad \Phi(f_{target}) = (1 - \pi)f'_{online} + \pi f_{target}
\end{aligned} \tag{32}$$

where $\text{Sg}$ expresses the operator that prevents gradient backpropagation, $f'$ denotes the feature extractor in the target network, and $f'_p$ denotes the projection head in the target network.

**MAE** is a representative G-SSL method, distinguished by its reconstruction-based approach. It leverages masking strategies to reconstruct missing parts of input samples, particularly effective in the context of vision transformers. Specifically, given an input sample $x_i$, MAE first generates a masked version $x_i^1$ by randomly masking a significant proportion of patches from $x_i$. The masked sample is then processed by an encoder network $f$ to produce latent representations $z_i^1$, e.g., $z_i^1 = f(x_i^1)$. Subsequently, we denote $x_i$ as $x_i^2$, a decoder network $f_{dec}$ reconstructs the original input from these latent representations, e.g., $\hat{x}_i^1 = f_{dec}(z_i^1)$. The objective function for MAE involves minimizing the reconstruction loss, typically measured using the mean squared error (MSE) between the original and reconstructed patches:

$$\mathcal{L}_{\text{MAE}} = \sum_{i=1}^{N} \left\| \hat{x}_i^1 - x_i^2 \right\|_2^2. \tag{33}$$

From the perspective of aligning and constraining, MAE's reconstruction task inherently aligns the representation $z_i$ with its original sample $x_i$ by ensuring accurate reconstruction (alignment). At the same time, the aggressive masking strategy and limited reconstruction targets implicitly constrain the model, forcing it to capture meaningful structural and semantic features from limited input information.

**Step 2: Mechanism for forming classification tasks** As shown in Appendix H.2, the construction of the classification-like tasks consists of two parts: (i) dataset partitioning: the dataset is partitioned into many pairs, each pair approximating the positive samples of a single class; and (ii) the self-supervised objective: both G-SSL and D-SSL contain an alignment term that enforces alignment between the positive sample in a pair and its corresponding anchor. The anchor can be viewed as the class centroid for the class defined by that pair. In Step 1 we explicitly described how BYOL and MAE implement this alignment term.

Although the alignment term takes the form of a mean-squared-error (MSE) loss, its target is the anchor, which is treated as the class centroid. Hence the MSE alignment loss can also be interpreted as encouraging aggregation toward the class centroid, i.e., intra-class compactness. Therefore, this objective can be seen as a classification task, analogous to prototype or nearest-neighbor classification (Peterson, 2009; Chang, 1974).

**Step 3: Empirical evidence** In the original submission, our experiments demonstrate that the "anchor-alignment" induced by different SSL paradigms indeed produces within-batch clustering and benefits downstream tasks: (i) If the view that anchors act as class/cluster centroids were incorrect, then explicitly adding a discriminative term $\mathcal{L}_{disc}$ to the training objective should not markedly improve decision boundaries or transfer performance. However, in Appendix G.2 we observe the opposite: adding the discriminative term produces clearer class boundaries in feature space (Figure 8). (ii) Experiments across more than 25 baselines show that, consistent with the above statement, GeSSL reliably improves model performance, with average gains exceeding 2%, including improvements over self-distillation methods (e.g., BYOL, SimSiam) and generative methods (e.g., MAE).

## H.5 UNDERSTANDING WHY GESSL INTEGRATES WITH CONTRASTIVE, DISTILLATION, AND GENERATIVE SSL PARADIGMS

Self-supervised learning encompasses multiple paradigms, including contrastive, self-distillation, and generative reconstruction (D-SSL and G-SSL), whose loss functions differ substantially in form.

This section aims to analyze and experimentally validate why GeSSL can be integrated with these different objectives in a unified manner. Specifically, we investigate whether these paradigms share a common structural mechanism that allows GeSSL to treat each mini-batch as a pseudo-task, enabling compatibility across diverse SSL losses.

**Step 1: For SSL, different augmentations of the same sample belong to the same class, while augmentations of different samples belong to different classes, this is distinct from the sample's ground-truth label.** First, as described in Section 2 (first and second paragraphs) of the original submission, both D-SSL and G-SSL reorganize data into pairs. Each pair contains an augmented view and an anchor, and every pair is produced from the same original example. Second, also in Section 2, the objectives of D-SSL methods (e.g., SimCLR, BYOL, Barlow Twins) can be written as an alignment term plus a regularization term, whereas G-SSL objectives are typically expressed as alignment terms. The alignment term enforces that the augmented view aligns with its anchor; the regularizer may be implemented in various ways (e.g., encouraging a uniform distribution over augmentations, gradient clipping, etc.). Third, if we regard the anchor as a cluster center, the alignment term drives intra-class compactness by pulling same-class samples toward that center. Consequently, we can view an SSL dataset as partitioned into many classes, where each class corresponds to a pair. **Importantly, this partitioning is not based on domain or ground-truth labels.** Finally, this perspective suggests a way to mine representational universality with SSL: rather than being tied to a particular domain or ground-truth label, SSL assigns labels at the pair level, with the anchor serving as the class centroid. This is a plausible assumption since no two raw samples are identical; such pair-level labeling is therefore more general than semantic ground-truth labels.

**Step 2: SSL training can be interpreted as task-based training, where each mini-batch constitutes a task.** From Step 1, each pair defines a class and different mini-batches contain different sets of pairs; hence each mini-batch can be regarded as a distinct task.

**Step 3: Rethink G-SSL and D-SSL from the perspective of alignment and regularization.** Please see the step 1 of Appendix H.4.

**Step 4: Mechanism for forming classification tasks.** As shown in Steps 1–3, the construction of these classification-like tasks has two components: (i) dataset partitioning, the dataset is partitioned into many pairs, each pair approximating the positive samples of a single class; and (ii) the self-supervised objective, both G-SSL and D-SSL include an alignment term that enforces alignment between the positive sample in a pair and its corresponding anchor. The anchor can be viewed as the class centroid for the class defined by that pair. In Step 1 we gave concrete examples of how BYOL and MAE realize this alignment term. Although the alignment term often takes the form of a mean-squared-error, its target is the anchor, which is treated as the class centroid. Thus the MSE alignment loss can be interpreted as encouraging aggregation toward the class centroid, i.e., intra-class compactness. Consequently, this objective can be seen as a classification task, analogous to prototype or nearest-neighbor classifiers (Peterson, 2009; Chang, 1974).

**Step 5: Why GeSSL can integrate with different SSL losses.** From the above perspective, SSL losses can be viewed as different implementations of discriminative objectives; therefore, our framework can incorporate a variety of SSL loss functions. Moreover, as shown in Section 4 and Appendix B, we prove that if the basic SSL alignment guarantees a within-batch distance $\leq \varepsilon$ and $\mathcal{L}_{disc}$ maintains an inter-cluster margin $\geq \Delta$, then the GeSSL-learned representation admits a tighter upper bound on downstream generalization error. Empirically, we validated GeSSL across more than 25 baselines, covering contrastive, distillation, and generative SSL methods, and the results in Tables 1–20 demonstrate consistent gains, indirectly confirming that the pairwise-alignment modeling adapts across different loss structures.

## H.6 MORE ILLUSTRATION OF $f$, $f'$ AND $g$

To make the bi-level optimization in GeSSL fully transparent, we explicitly define the three key components, i.e., $f$, $f'$, and $g$, and describe how they interact during training.

Specifically, $f$ corresponds to the SSL model, and also is our optimization target: it maps an input sample $x$ to a representation $f(x)$, and in GeSSL serves as the main set of parameters being optimized (used to compute $\mathcal{L}_{ssl}$ and $\mathcal{L}_{disc}$), i.e., the inner- and outer-objectives minimized over $f, g$ are both based on the base model $f$. Then, $f'$ corresponds to the proxy model, gen-

erated from $f$ and $g$, i.e., $f'(f,g)$: it is defined as the solution of the inner minimization problem, responsible for fast adaptation on the support set and serving as the outer-loop evaluator, i.e., $f'(f,g) \leftarrow f - \alpha \nabla_f \sum_{l=1}^{m} [\mathcal{L}_{ssl}(f, X_{tr,l}^{\text{sup}}) + \mu \mathcal{L}_{disc}(f, g, X_{tr,l}^{\text{sup}})]$ and in practice $f'$ is obtained by one (or several) gradient steps from $f$ in the inner loop. $g$ corresponds to the mean and covariance matrix of the vector set $\{f(x_{j,l}^1) - f(x_{i,l}^{\text{anchor}})\}_{j=1}^n$: $g$ takes batch-wise difference-vector statistics (mean and covariance matrix) as input and outputs a scalar threshold $a_i \in \mathbb{R}$ for each anchor. In the discriminative loss $\mathcal{L}_{disc}$, if the distance from a sample to an anchor $d_{ij} \leq a_i$ it is pulled closer; otherwise it is pushed away. For optimization convenience, the indicator function is replaced by a differentiable approximation $w(a_i) = \text{Sigmoid}(k(a_i - d_{ij}))$. In other words, $g$ learns a discriminative threshold per anchor to guide the behavior of $\mathcal{L}_{disc}$. GeSSL alternates between two optimization stages: (i) In the inner-loop, it fix $g$, perform fast updates of $f$ on the support set using $\mathcal{L}_{ssl} + \mu \mathcal{L}_{disc}$ to obtain $f'$. (ii) In the outer loop, it simultaneously update $f$ and $g$ using $\mathcal{L}_{ssl}(f'(f,g), X_{tr,l}^{\text{que}})$ computed on the query set.

## H.7    BROADER IMPACTS AND LIMITATIONS

In this subsection, we briefly illustrate the broader impacts and limitations of this work.

**Broader Impacts**    This work advances SSL by explicitly modeling "universality", which refers to the capacity of representations to discriminate, generalize, and transfer, through a unified bi-level optimization that balances task-specific adaptation with cross-task consistency. By deriving a theoretical generalization bound, we provide formal guarantees that GeSSL's learned features will perform robustly on unseen tasks. Empirically, GeSSL delivers SOTA results across diverse benchmarks, demonstrating its advantages across various settings. This work benefits the field of SSL and machine learning, and also opens up exciting new avenues for future research.

**Limitations**    This work includes analyses under a variety of settings and presents extensive empirical evidence of its effectiveness. However, it does not offer a dedicated examination of large language models, e.g., models exceeding 7B parameters such as LLaMA, Qwen, and Mistral, despite demonstrating GeSSL's effectiveness on large-scale vision-language models such as CLIP. We will investigate additional case studies to extend this work in the future.

## H.8    USE OF LARGE LANGUAGE MODELS

Large language models (LLMs) were used solely as a general-purpose tool for grammar and language polishing. They were not involved in research ideation, methodology, analysis, or substantive writing. The authors take full responsibility for all contents of the paper.

