# OpenReview forum: "On the Universality of Self-Supervised Learning"
_ICLR.cc/2026/Conference — Submitted to ICLR 2026_

### Official Review · Reviewer_gP8E · 2025-10-25

**Soundness:** 3
**Presentation:** 3
**Contribution:** 2
**Rating:** 4
**Confidence:** 4

**Summary:**

This paper proposes General SSL (GeSSL) which integrates a bi-level optimization structure for self-supervised learning. The authors conduct experiments on various benchmarks using different SSL methods. Experimental results show that the proposed GeSSL method achieves improvements when integrated with different SSL methods.

**Strengths:**

1. This paper is well-presented and easy to follow.

2. The authors also provide theoretical analysis.

3. The authors selected multiple different SSL methods as baselines, and improvements can be observed across all of them with the proposed GeSSL.

**Weaknesses:**

1. The results in Table 1 are not consistent with the original papers. For instance, In the original paper, Barlow Twins achieves Top-1 and Top-5 accuracy of 73.2% and 91% respectively on ImageNet using ResNet-50. However, in Table 1 of this paper, the Top-1 accuracy of Barlow Twins is 69.94%, and with GeSSL it reaches 72.84%. These results are significantly lower than those reported in the original paper.
2. Additional results are needed to demonstrate the necessity of bi-level optimization.
3. Lack definition of $f$, $f'$ and $g$.
4. line 204: $L_{disc}(g, X)$ should be $L_{disc}(f, g, X)$?

**Questions:**

Please refer to weakness

---

> ### Author Response · Authors · 2025-11-20
> **Response to Weakness 1**
>
> **We sincerely appreciate the reviewer gP8E's valuable feedback and the time and effort for the reviewing. We will respond to each issue of the raised "Weaknesses" and "Questions" in turn, sincerely hoping the following responses can help eliminate the concerns.**
>
>
> > **Weakness 1:** The results in Table 1 are not consistent with the original papers. For instance, In the original paper, Barlow Twins achieves Top-1 and Top-5 accuracy of 73.2% and 91% respectively on ImageNet using ResNet-50. However, in Table 1 of this paper, the Top-1 accuracy of Barlow Twins is 69.94%, and with GeSSL it reaches 72.84%. These results are significantly lower than those reported in the original paper.
>
> ## Response to Weakness 1
>
> We sincerely appreciate the reviewers' valuable comments. We would like to clarify that the discrepancy comes from different settings, i.e., epoch budgets, used in the reported tables. **Table 1** in the main text reports results at 200 epochs (where Barlow Twins obtains Top-1 = 69.94),  while **Table 7** in the Appendix reports results at 100, 200, 400, and 1000 epochs and shows that Barlow Twins reaches Top-1 = 73.29% at 1000 epochs (consistent with the original paper).
>
> We have also mentioned it in the caption of **Table 1**, i.e., "on the ImageNet-100 and ImageNet (200 Epochs)".

---

> ### Author Response · Authors · 2025-11-20
> **Response to Weakness 2**
>
> > **Weakness 2:** Additional results are needed to demonstrate the necessity of bi-level optimization.
>
> ## Response to Weakness 2
>
> We sincerely appreciate the reviewers' valuable comments. We explain the necessity of bi-level optimization from the following perspectives:
>
> 1. **Empirical perspective:** We have provided ablation experiments on bi-level optimization in **Section 5.2** of the original submission. **Figure 10** shows that introducing bi-level optimization significantly improves performance in unsupervised learning, semi-supervised learning, and few-shot settings, demonstrating its necessity.
>
> 2. **Theoretical perspective:** The core role of bi-level optimization is to treat "generalization performance on unseen queries" as the objective of the outer loop, which allows the training process to directly minimize generalization error rather than merely optimizing pairwise consistency; this aligns with our goal of universality. Moreover, our theoretical analysis in **Section 4** and **Appendix A** formalizes this point and proves that, under the stated assumptions, the bi-level objective yields tighter error bounds on downstream generalization.
>
> 3. **Efficiency of bi-level:**
>    - Experiments in **Appendix F.1** show that GeSSL can deliver substantial performance improvements while maintaining reasonable computational efficiency: as shown in **Figure 2** and **Table 19**, although GeSSL increases memory footprint and parameter size, these increases are acceptable relative to the performance gains (less than a $0.1\times$ increase); Also based on [1], this trade-off between added parameters/memory and generalization gains is considered acceptable. [1] Bengio, Y. (2009). Learning deep architectures for AI.
>    - As mentioned in L1642–1654, although GeSSL uses a more complex optimization method, the bi-level construction is designed to guide the model to optimize more effectively for specific tasks. In fact, GeSSL updates using AID-FD rather than conventional explicit second-order differentiation (see **Appendix F.4**). Therefore, efficiency gains manifest in per-iteration computational efficiency and update effectiveness, not merely in reduced iteration counts. **Tables 19** and **Figure 2** empirically demonstrate this. Moreover, to verify that the efficiency improvement is not solely due to single-cycle iteration effects, we separately measured the per-cycle computational overhead of SSL baselines after integrating GeSSL. Results in **Table 18** show that, with equal batch sizes, GeSSL improves the computational efficiency and per-iteration update effectiveness of SSL baselines.
>
> 4. **Stability analyses:** Motivated by prior work [1–4] linking training stability, convergence to global minima, and generalization, and by the gradient-noise-scale framework of [5] which uses gradient variance as a measurable indicator of training stability, we compared SimCLR augmented with ICL-MSR [6], MetAug [7], MCR, and GeSSL on ImageNet. We recorded per-step gradient-norm time series and their variances, as well as final performance and seed-to-seed variability across different random seeds. The results (shown in the table below) indicate that, compared with MCR and other meta-regularizers, GeSSL yields substantially lower gradient means and variances throughout training, smaller seed-to-seed performance fluctuations, and achieves SOTA final performance, suggesting GeSSL is less prone to getting trapped in poor local minima. We also recorded the SimCLR+GeSSL training loss curve; because figure inclusion is constrained during rebuttal, we provide the loss trajectory in **Table 2** here and will include full plots in the revised manuscript.
>
> |Method|Mean Gradient Norm|GradNorm Variance|ACC|
> |-|-|-|-|
> | SimCLR | 0.045$\pm$0.012|2.1e-4|69.9|
> | SimCLR+ICL-MSR|0.042$\pm$0.010|1.8e-4|70.8|
> | SimCLR+MetAug| 0.041$\pm$0.009|1.7e-4|71.0|
> | SimCLR+MCR|0.039$\pm$ 0.008|1.4e-4|71.6|
> | SimCLR+GeSSL|0.033$\pm$0.006|9.2e-5|72.8|
>
> |Epoch|40|60|80|100|120|140|160|180|200|
> |-|-|-|-|-|-|-|-|-|-|
> |Loss|1.50|1.35|1.22|1.10|1.02|0.95|0.90|0.86|0.83|
>
> [1] Keskar, N. S., Mudigere, D., Nocedal, J., Smelyanskiy, M., & Tang, P. T. P. (2016). On large-batch training for deep learning: Generalization gap and sharp minima.
>
> [2] Hardt, M., Recht, B., & Singer, Y. (2016, June). Train faster, generalize better: Stability of stochastic gradient descent.
>
> [3] Chaudhari, P., Choromanska, A., Soatto, S., LeCun, C., Borgs, C., & Zecchina, R. (2019). Entropy-sgd: Biasing gradient descent into wide valleys.
>
> [4] Foret, P., Kleiner, A., Mobahi, H., & Neyshabur, B. (2020). Sharpness-aware minimization for efficiently improving generalization.
>
> [5] McCandlish, S., Kaplan, J., Amodei, D., & Team, O. D. (2018). An empirical model of large-batch training.
>
> [6] Qiang, W. (2022). Interventional contrastive learning with meta semantic regularizer.
>
> [7] Li, J. (2022). Metaug: Contrastive learning via meta feature augmentation.

---

> ### Author Response · Authors · 2025-11-20
> **Response to Weakness 3**
>
> > **Weakness 3:** Lack definition of $f$, $f^{'}$ and $g$.
>
> ## Response to Weakness 3
>
> We sincerely appreciate the reviewers' valuable comment. Building on **Section 3.2** (L190–243) of the original submission, we further define and explain $f$, $f'$ and $g$, and will add these to the revised manuscript. Specifically:
>
> - **$f$** corresponds to the SSL model, and also is our optimization target: it maps an input sample $x$ to a representation $f(x)$, and in GeSSL serves as the main set of parameters being optimized (used to compute $L_{\rm ssl}$ and $L_{\rm disc}$), i.e., the inner- and outer-objectives minimized over $f,g$ are both based on the base model $f$.
> - **$f'$** corresponds to the proxy model, generated from $f$ and $g$, i.e., $f'(f,g)$: it is defined as the solution of the inner minimization problem, responsible for fast adaptation on the support set and serving as the outer-loop evaluator, i.e., $f'(f,g) \gets f - \alpha \nabla\_{f}  \sum\nolimits\_{l = 1}^{m} {[{\mathcal{L}\_{ssl}}(f,X_{tr,l}^{\text{sup}}) + \mu {\mathcal{L}\_{disc}}(f,g,X_{tr,l}^{\text{sup}})]}$ and in practice $f'$ is obtained by one (or several) gradient steps from $f$ in the inner loop.
> - **$g$** corresponds to the mean and covariance matrix of the vector set in L202: $g$ takes batch-wise difference-vector statistics (mean and covariance matrix) as input and outputs a scalar threshold $a_i\in\mathbb{R}$ for each anchor. In the discriminative loss $L_{\rm disc}$, if the distance from a sample to an anchor $d_{ij}\le a_i$ it is pulled closer; otherwise it is pushed away. For optimization convenience, the indicator function is replaced by a differentiable approximation $w(a\_i)=\mathrm{Sigmoid}(k(a\_i-d_{ij}))$. In other words, $g$ learns a discriminative threshold per anchor to guide the behavior of $L_{\rm disc}$.
> - Training procedure:
>   - *Inner-loop:* fix $g$, perform fast updates of $f$ on the support set using $L_{\rm ssl}+\mu L_{\rm disc}$ to obtain $f'$.
>   - *Outer-loop:* simultaneously update $f$ and $g$ using $L_{ssl}(f'(f,g),X^{\rm que}_{tr,l})$ computed on the query set.

---

> ### Author Response · Authors · 2025-11-20
> **Response to Weakness 4**
>
> > line 204: $L_{disc}(g,X)$ should be $L_{disc}(f,g,X)$?
>
> ## Response to Weakness 4
>
> We sincerely appreciate the reviewer's valuable suggestions and will revise $L_{disc}(g,X_{tr,l}^{sup})$ to $L_{disc}(f,g,X_{tr,l}^{sup})$ in the new version.

---

> ### Author Response · Authors · 2025-11-28
> **Kindly Checking on the Review Progress**
>
> Dear Reviewer gP8E,
>
> We would like to sincerely thank you again for the time and effort you have devoted to our paper. We truly appreciate your careful oversight of the review process, and the constructive comments have helped us improve our work.
>
> We have submitted the detailed responses to all the constructive comments. Since several days have passed, we wish to kindly check whether any further concerns or suggestions have arisen, or whether there is any additional information we can provide to assist the ongoing process.
>
>  We fully understand the workload involved in handling submissions, and we are grateful for your continued guidance and support. Thank you very much, and we look forward to hearing from you at your convenience.
>
> Sincerely,
>
> The Authors

---

### Official Review · Reviewer_2Hsd · 2025-10-27

**Soundness:** 3
**Presentation:** 3
**Contribution:** 3
**Rating:** 6
**Confidence:** 4

**Summary:**

This paper proposed a new framework for improving universality in self-supervised learning (SSL).
The authors argue that a good representation should exhibit discriminability, generalizability, and transferability.
To achieve this, they propose General Self-Supervised Learning (GeSSL) — a unified framework that explicitly models these three dimensions through bi-level optimization integrating task-specific adaptation and cross-task consistency. Theoretical analysis provides a generalization bound, while extensive experiments on diverse benchmarks (ImageNet, COCO, VOC, miniImageNet, etc.) demonstrate significant performance gains over SOTA methods.

**Strengths:**

1. Proposed the unified concept of “universality,” systematically integrating discriminability, generalizability, and transferability
2. Implemented comprehensive validation across unsupervised, semi-supervised, transfer, and few-shot learning tasks.
3. GeSSL has strong adaptability by integrating with existing SSL paradigms (SimCLR, MoCo, BYOL, MAE, etc.)

**Weaknesses:**

1. Lack of the quantified metric to the defination of what is actually the "Universality"
2. Experiments are mainly on vision domain, lacking validation on multimodal or language-based SSL tasks.
3. Limited direct comparison with recent meta-SSL frameworks such as MCR [1]. From my poingt of view, MCR has excatly similar structure with this work. What is the technical improvement of this work compared to MCR?


[1] Guo, Huijie, et al. "Self-supervised representation learning with meta comprehensive regularization." Proceedings of the AAAI Conference on Artificial Intelligence. Vol. 38. No. 3. 2024.

**Questions:**

Q1: How can “universality” be quantitatively measured or benchmarked across models?

Q2: Would the GeSSL structure generalize effectively to multimodal or temporal tasks?

Q3: What is the most technical improvement of this work compared to MCR? Compared to MCR or other meta-regularization methods, is GeSSL more prone to local minima?

---

> ### Author Response · Authors · 2025-11-20
> **Response to Weakness 1 & Question 1**
>
> **We sincerely appreciate the reviewer 2Hsd's valuable feedback and the time and effort for the reviewing. We will respond to each issue of the raised "Weaknesses" and "Questions" in turn, sincerely hoping the following responses can help eliminate the concerns.**
>
>
> > **Weakness 1**: Lack of the quantified metric to the defination of what is actually the "Universality"
> >
> > **Question 1**: How can “universality” be quantitatively measured or benchmarked across models?
>
> ## Response to Weakness 1 & Question 1
>
> We sincerely appreciate the reviewers' valuable comments. **Definition 3.1** presents the formal definition of *Universality*. Based on this, **Appendix E.4** introduces a practical evaluation protocol for measuring the universality of existing SSL methods, i.e., a provable $\sigma$ metric that quantifies a method’s universality score, together with empirical validation.
>
> - **$\sigma$ metric:**
>    The $\sigma$ metric quantifies the performance gap between the learned model and the optimal model for each task. Concretely, we assume the optimal model outputs the true label distribution and measure the discrepancy between the model’s predicted class-probability distribution and the ground-truth one-hot distribution using KL divergence. For example, a prediction $[0.81,0.09,0.03,0.07]$ is compared against the true label $[1,0,0,0]$ via KL divergence.
> - **Computation protocol:**
>    Taking the LIN task as an example, we train SimCLR and SimCLR+GeSSL on COCO for 200 epochs, append a small MLP after the feature extractor, and feed a held-out mini-batch through each model to produce per-sample predicted class distributions. We compute the KL divergence between these predicted distributions and the ground-truth one-hot vectors, normalize the divergences, and obtain the LIN task score. The same procedure is applied to other baselines and tasks.
> - **Why this metric is sensible:**
>    For convenience, we convert the normalized $\sigma$ into a "goodness" score $S=1-\mathrm{norm}(\sigma)$. Intuitively, $S$ captures the three components of universality:
>   - *Discriminability:* On the training task, we attach a light-weight classification head to the encoder and compute the KL between its predictions and the ground-truth one-hot labels. If the model concentrates probability mass on the correct class (i.e., behaves like an optimal model with high accuracy and confidence), the KL tends to zero ($\sigma\to0$) and $S\to1$. Thus, $S$ directly measures the ability to form clear class boundaries on the training data.
>   - *Generalizability:* Applying the same KL computation on the task’s test split yields a test $\sigma$. If the model produces near one-hot predictions on unseen samples (small KL), then $S$ is high, indicating a small train, then test performance drop (small generalization gap). Hence the train/test $\sigma$ values and their difference directly quantify the paper’s requirement of "maintaining high accuracy on test".
>   - *Transferability:* For a collection of new test tasks whose label sets do not overlap with the training task, we use the same encoder, train/evaluate a light-weight head per new task, compute the task-specific $\sigma$ values, and aggregate them (e.g., by averaging) to obtain a cross-task $\sigma$. A small aggregated $\sigma$ indicates that the representation approaches the per-task optimal models across many new tasks, i.e., it has strong transferability.
> - **Empirical evaluation:**
>    We evaluate universality on both image and video benchmarks. Image tasks include: linear probing of 800-epoch pretrained models (LIN, top-1), semi-supervised classification using 1% of labels (SEMI, top-1), VOC/COCO object detection (box AP), and COCO instance segmentation ($AP^{\text{mask}}$). Video tasks include SOT (AUC), VOS ($\mathcal{J}$-mean), and MOT / PoseTracking / MOTS (all measured by IDF-1). We compute the $\sigma$ metric from the multi-task results after 200 epochs (treating the optimal model as the oracle) across a set of randomized tasks; lower $\sigma$ indicates more stable / better cross-task performance. Results in **Figure 5** and **Table 13** show that (i) existing SSL models exhibit relatively high $\sigma$ and thus struggle to maintain consistent performance across domains and tasks, and (ii) integrating GeSSL substantially reduces $\sigma$, indicating improved universality and overall performance, i.e., findings consistent with the performance tables (**Tables 1–20**).

---

> ### Author Response · Authors · 2025-11-20
> **Response to Weakness 2 & Question 2**
>
> > **Weakness 2**: Experiments are mainly on vision domain, lacking validation on multimodal or language-based SSL tasks.
> >
> > **Question 2**: Would the GeSSL structure generalize effectively to multimodal or temporal tasks?
>
> ## Response to Weakness 2 & Question 2
>
> We sincerely appreciate the reviewers' valuable comments. We will address this question as follows:
>
> - We would like to kindly clarify that we report GeSSL results on language-based SSL tasks in **Appendix E.6** of the original submission, i.e., IC03 and IIIT5K (Yasmeen et al., 2020). The experiments in **Table 17** demonstrate GeSSL’s effectiveness, yielding over 10% absolute improvement.
> - We also provide results on video-based tasks in **Appendix E.2** and **Appendix E.4** to validate temporal performance. Experiments in **Table 9** and **Figure 5** show that applying GeSSL across six strong baselines produces consistent performance gains, confirming its effectiveness on temporal tasks.
> - Regarding multimodal tasks, **Appendix G.2** acknowledged the prior limitation that multimodal validation was absent. Following the reviewer’s constructive suggestion, we added multimodal experiments: following the protocols in [1] and [2], we evaluate image-to-text retrieval on COCO and report test accuracy on ColoredMNIST. We record R@1 for MoCo-v2 and VirTex before and after integrating GeSSL. The results in the table below show consistent improvements after adding GeSSL, validating its effectiveness on multimodal tasks.
>
> | Methods (COCO) | Original | +GeSSL |
> | -------------- | -------- | ------ |
> | MoCo-v2        | 51.6     | 52.9   |
> | VirTex         | 58.1     | 58.7   |
>
> | Methods (ColoredMNIST) | Test Accuracy |
> | ---------------------- | ------------- |
> | SimCLR                 | 82.1          |
> | SimCLR+GeSSL           | 85.9          |
>
> [1] Yuan, X., Lin, Z., Kuen, J., Zhang, J., Wang, Y., Maire, M., ... & Faieta, B. (2021). Multimodal contrastive training for visual representation learning. In *Proceedings of the IEEE/CVF conference on computer vision and pattern recognition* (pp. 6995-7004).
>
> [2] Huang, W., Han, A., Chen, Y., Cao, Y., Xu, Z., & Suzuki, T. (2024). On the comparison between multi-modal and single-modal contrastive learning. *Advances in Neural Information Processing Systems*, *37*, 81549-81605.

---

> ### Author Response · Authors · 2025-11-20
> **Response to Weakness 3 & Question 3 (Part I)**
>
> > **Weakness 3**: Limited direct comparison with recent meta-SSL frameworks such as MCR [1]. From my poingt of view, MCR has excatly similar structure with this work. What is the technical improvement of this work compared to MCR?
> >
> > [1] Guo, Huijie, et al. "Self-supervised representation learning with meta comprehensive regularization." Proceedings of the AAAI Conference on Artificial Intelligence. Vol. 38. No. 3. 2024.
> >
> > **Question 3**: What is the most technical improvement of this work compared to MCR? Compared to MCR or other meta-regularization methods, is GeSSL more prone to local minima?
>
> ## Response to Weakness 3 & Question 3
>
> We sincerely appreciate the reviewers' valuable comment, and would like to illustrate from the following two parts.
>
> **Part I: Comparison between GeSSL and MCR：Advantages of GeSSL**
>
>
>
> First, we provide a brief overview of MCR. Specifically, MCR, by introducing a mutual-information term, increases task-related mutual information in feature representations, thereby enhancing the discriminative information in the features extracted by SSL. It learns three networks, namely the feature-extraction network, the projection-head network, and the information-extraction network associated with computing mutual information. Its entire training process is divided into two steps: (i) fix the information-extraction network and update the parameters of the feature-extraction network and the projection-head network via gradient backpropagation; (ii) fix the feature-extraction network and the projection-head network and update the parameters of the information-extraction network using second-order gradients. It is worth noting that the update procedure of the information-extraction network can be regarded as a bi-level process.
>
> Compared to MCR, the GeSSL proposed in this paper has the following advantages:
>
> - GeSSL increases the discriminability of SSL via $L_{disc}$; from this perspective, GeSSL also enhances the discriminability of learned representations through an additional regularization term. However, beyond enhancing representation discriminability, GeSSL also improves the transferability and generalizability of SSL-learned feature representations via an episodic learning paradigm and an invariance learning paradigm.
> - GeSSL uses a bi-level mechanism to learn the feature-extraction network and the projection-head network, whereas MCR’s bi-level mechanism is not used to learn the feature-extraction network and the projection-head network but is used to learn the information-extraction network.
> - GeSSL explicitly merges discriminability, generalizability, and transferability into a single alignable objective “Universality” (**Definition 3.1**), and directly embeds this “Universality” into the training objective and analysis framework; compared with MCR, we do not merely “augment” representation completeness via regularizers or modules, but first define what constitutes a good representation and then design learning terms and training paradigms according to that objective.
> - GeSSL provides provable generalization error bounds corresponding to “Universality”; from these bounds one can see that GeSSL improves the discriminability, transferability, and generalizability of SSL-learned feature representations, whereas MCR focuses on showing, from a causal perspective, that the feature-extraction network learned by SSL can capture more discriminative information.
> - Empirically, we further constructed a set of experiments to compare the performance of MCR and GeSSL as suggested. In short, we follow the experimental setup in **Section 5.1** and evaluate the performance of baselines such as SimCLR, BYOL, and MAE on ImageNet when augmented with MCR or with GeSSL. The experimental results in the table below show that GeSSL yields larger performance gains. In addition, GeSSL can also be applied on top of MCR; the table below demonstrates its effectiveness.
>
> | Method | Original | +MCR | +GeSSL | +GeSSL & MCR |
> | ------ | -------- | ---- | ------ | ------------ |
> | SimCLR | 70.4     | 71.2 | 72.8   | 73.0         |
> | BYOL   | 73.4     | 75.1 | 75.9   | 76.4         |
> | MAE    | 77.9     | 78.9 | 80.4   | 80.8         |

---

> ### Author Response · Authors · 2025-11-20
> **Response to Weakness 3 & Question 3 (Part II)**
>
> **Part II: Analysis of local minima**
>
> Motivated by prior work [1–4] linking training stability, convergence to global minima, and generalization, and by the gradient-noise-scale framework of [5] which uses gradient variance as a measurable indicator of training stability, we compared SimCLR augmented with ICL-MSR [6], MetAug [7], MCR, and GeSSL on ImageNet. We recorded per-step gradient-norm time series and their variances, as well as final performance and seed-to-seed variability across different random seeds. The results (shown in the table below) indicate that, compared with MCR and other meta-regularizers, GeSSL yields substantially lower gradient means and variances throughout training, smaller seed-to-seed performance fluctuations, and achieves SOTA final performance, suggesting GeSSL is less prone to getting trapped in poor local minima. We also recorded the SimCLR+GeSSL training loss curve; because figure inclusion is constrained during rebuttal, we provide the loss trajectory in **Table 2** here and will include full plots in the revised manuscript.
>
> | Method             | Mean Gradient Norm | GradNorm Variance | ImageNet Linear-probe |
> | ------------------ | ------------------ | ----------------- | --------------------- |
> | SimCLR（baseline） | 0.045 $\pm$ 0.012  | 2.1e-4            | 69.9                  |
> | SimCLR + ICL-MSR   | 0.042 $\pm$ 0.010  | 1.8e-4            | 70.8                  |
> | SimCLR + MetAug    | 0.041 $\pm$ 0.009  | 1.7e-4            | 71.0                  |
> | SimCLR + MCR       | 0.039 $\pm$ 0.008  | 1.4e-4            | 71.6                  |
> | SimCLR + GeSSL     | 0.033 $\pm$ 0.006  | 9.2e-5            | 72.8                  |
>
> | Epoch | 40   | 60   | 80   | 100  | 120  | 140  | 160  | 180  | 200  |
> | ----- | ---- | ---- | ---- | ---- | ---- | ---- | ---- | ---- | ---- |
> | Loss  | 1.50 | 1.35 | 1.22 | 1.10 | 1.02 | 0.95 | 0.90 | 0.86 | 0.83 |
>
> [1] Keskar, N. S., Mudigere, D., Nocedal, J., Smelyanskiy, M., & Tang, P. T. P. (2016). On large-batch training for deep learning: Generalization gap and sharp minima. *arXiv preprint arXiv:1609.04836*.
>
> [2] Hardt, M., Recht, B., & Singer, Y. (2016, June). Train faster, generalize better: Stability of stochastic gradient descent. In *International conference on machine learning* (pp. 1225-1234). PMLR.
>
> [3] Chaudhari, P., Choromanska, A., Soatto, S., LeCun, Y., Baldassi, C., Borgs, C., ... & Zecchina, R. (2019). Entropy-sgd: Biasing gradient descent into wide valleys. *Journal of Statistical Mechanics: Theory and Experiment*, *2019*(12), 124018.
>
> [4] Foret, P., Kleiner, A., Mobahi, H., & Neyshabur, B. (2020). Sharpness-aware minimization for efficiently improving generalization. *arXiv preprint arXiv:2010.01412*.
>
> [5] McCandlish, S., Kaplan, J., Amodei, D., & Team, O. D. (2018). An empirical model of large-batch training. *arXiv preprint arXiv:1812.06162*.
>
> [6] Qiang, W., Li, J., Zheng, C., Su, B., & Xiong, H. (2022, June). Interventional contrastive learning with meta semantic regularizer. In *International conference on machine learning* (pp. 18018-18030). PMLR.
>
> [7] Li, J., Qiang, W., Zheng, C., Su, B., & Xiong, H. (2022, June). Metaug: Contrastive learning via meta feature augmentation. In *International conference on machine learning* (pp. 12964-12978). PMLR.

---

> > ### Comment · Reviewer_2Hsd · 2025-11-24
> >
> > Thank you for your response. All of my concerns have been addressed. I will maintain my evaluation as weak accept.

---

> > > ### Author Response · Authors · 2025-11-25
> > >
> > > Thank you very much for your thoughtful comments. We sincerely appreciate your positive evaluation. We will upload an updated version of the manuscript reflecting the revisions and clarifications within three days.

---

### Official Review · Reviewer_z4r6 · 2025-10-28

**Soundness:** 3
**Presentation:** 3
**Contribution:** 3
**Rating:** 6
**Confidence:** 5

**Summary:**

This paper introduces the concept of universality in self-supervised learning (SSL) as a unifying criterion for good representations, defined by three properties: discriminability, generalizability, and transferability. To explicitly model these qualities, the authors propose a framework called General SSL (GeSSL) which incorporates a bi-level (two-tier) optimization approach inspired by meta-learning (MAML) to simulate training and testing within each mini-batch episode. GeSSL adds an alignment-based discriminative loss to improve class separation, uses a bi-level “update-then-evaluate” mechanism on split support/query sets to directly model generalization. This unified objective is designed to produce “universal” representations that perform well on the training data, generalize to unseen in-distribution data, and transfer to new tasks. The paper provides a theoretical generalization bound showing that under certain smoothness and boundedness assumptions, GeSSL’s training yields representations with bounded error on novel tasks. Empirically, GeSSL is evaluated by integrating it with various SSL methods (e.g. contrastive, distillation, and generative paradigms) and tested on multiple benchmarks.

**Strengths:**

S1. The paper clearly defines what constitutes a good SSL representation (discriminability, generalizability, transferability) and consolidates these into a single goal. This brings conceptual clarity and a new perspective to SSL, shifting focus from “which tricks yield good features” to “what properties should good features have”. By explicitly modeling these properties in the loss, the approach directly targets the end-goal of SSL rather than relying on indirect proxies.

S2. The proposed GeSSL framework is a principled integration of meta-learning techniques with SSL. By employing a bi-level optimization (inner support set update, outer query set evaluation) and episodic training, GeSSL explicitly simulates the train-test process within SSL training. This is reminiscent of meta-learning for few-shot learning but effectively applied in a fully self-supervised context. This design is novel in SSL and provides a unified way to enforce that learned features not only fit the training data but also generalize to new data/tasks. The idea of using an episodic bi-level objective to improve unsupervised representation learning is an innovative contribution, distinguishing GeSSL from standard SSL methods that optimize a single-level loss.

S3. The paper includes a theoretical generalization bound for the learned representation. Providing such a generalization guarantee is a strength, as many SSL works remain empirical. The authors prove (under certain regularity conditions) that GeSSL’s learning objective leads to a bounded generalization error on new tasks. This result lends credence to the claim that GeSSL can produce representations that transfer well, and it grounds the method in learning theory. Having a theory-backed discussion of why the method should work (beyond intuitive arguments) is commendable and relatively rare in SSL literature.

S4. The experimental evaluation is broad in scope. GeSSL is shown to be compatible with multiple SSL paradigms – the paper augments several diverse methods (e.g. contrastive methods like SimCLR, knowledge-distillation methods like BYOL, redundancy-reduction methods like Barlow Twins, etc.) with the GeSSL components and demonstrates consistent performance gains. For example, models such as SimCLR+GeSSL or BYOL+GeSSL outperform their vanilla counterparts on image classification benchmarks, indicating the framework’s versatility.

**Weaknesses:**

W1- Theoretical Analysis for Multi-Paradigm Generality: The paper should include a deeper analysis or explanation of why GeSSL can integrate with different SSL losses. My suggestion is to formalize the idea that each mini-batch forms a pseudo-task where one view acts as a “class prototype” (as hinted in the paper’s discussion of anchors and clustering). If this holds, one could argue that contrastive, distillation, and generative SSL all enforce some form of pairwise consistency that GeSSL leverages by treating each pair as a class/task. Making this argument explicit would help readers understand the compatibility.

W2 - Benchmark on Distribution Shifts: To address the first weakness, the authors should evaluate GeSSL-pretrained models on dedicated robustness benchmarks. For example, running experiments on ImageNet-C [a] (common corruptions) will show how the universal representations handle noise, blur, etc., and testing on ImageNet-PD [b] (perspective distortion) will assess robustness to geometric changes.
[a] - Hendrycks, Dan, and Thomas Dietterich. "Benchmarking neural network robustness to common corruptions and perturbations." arXiv preprint arXiv:1903.12261 (2019).
[b] - Chhipa, P. C., Chippa, M. S., De, K., Saini, R., Liwicki, M., & Shah, M. (2024, September). Möbius transform for mitigating perspective distortions in representation learning. In European Conference on Computer Vision (pp. 345-363).

W3 -  Related work and comparisons. The paper is close in spirit to unsupervised meta-learning and episodic SSL; these are not positioned or compared. Include discussion and, if feasible, a baseline drawn from unsupervised MAML-style training. Also connect to robustness literature when claiming generalizability, and compare to strong non-GeSSL SSL baselines on the same downstream tasks to contextualize absolute performance.

W4 - Practical overhead and stability not characterized. Bi-level training adds inner-loop computation and sensitivity to step sizes, number of inner steps, and support/query split. The paper should report training cost, memory footprint, and stability measures, plus ablations on inner-loop depth and first- vs second-order updates, so practitioners can judge feasibility at scale.

Minor points -

1. Give a one-figure overview: data flow, inner update, outer update, and where each loss is applied. Caption should explain every symbol used in the figure.

2. Add a small “Notation and terms” table early (one column for term, one for meaning). Keep it to 10–12 entries max.

3. Put the main theorem as a boxed statement with a one-paragraph proof sketch. Move long proofs to appendix with clear pointers.

4. Consolidate training details in one place: datasets, image size, augmentations, batch size, optimizer, schedules, inner steps, outer steps. Include compute (GPU type, hours).
5. I don't see source code repo as of now.

**Questions:**

Weakness section includes the questions part as well. I am open raise my score upon getting satisfactory response on weaknesses.

---

> ### Author Response · Authors · 2025-11-20
> **Response to Weakness 1 (Step 1-2 & 3-part 1)**
>
> **We sincerely appreciate the reviewer z4r6's valuable feedback and the time and effort for the reviewing. We will respond to each issue of the raised "Weaknesses" and "Questions" in turn, hope the following responses will eliminate the concerns.**
>
>
>
> > **W1**- Theoretical Analysis for Multi-Paradigm Generality: The paper should include a deeper analysis or explanation of why GeSSL can integrate with different SSL losses. My suggestion is to formalize the idea that each mini-batch forms a pseudo-task where one view acts as a “class prototype” (as hinted in the paper’s discussion of anchors and clustering). If this holds, one could argue that contrastive, distillation, and generative SSL all enforce some form of pairwise consistency that GeSSL leverages by treating each pair as a class/task. Making this argument explicit would help readers understand the compatibility.
>
>
>
> ## Response to Weakness 1
>
> We sincerely thank the reviewer for the constructive comments and fully agree with it. In fact, we have already done this: **Section 2** of the original submission explains how mini-batches in D-SSL and G-SSL map to tasks, and how the SSL objectives induce classification-like tasks. We expand on this below.
>
> - **Step 1: For SSL, different augmentations of the same sample belong to the same class, while augmentations of different samples belong to different classes, this is distinct from the sample’s ground-truth label.**
>
>   First, as described in **Section 2** (first and second paragraphs) of the original submission, both D-SSL and G-SSL reorganize data into pairs. Each pair contains an augmented view and an anchor, and every pair is produced from the same original example.
>
>   Second, also in **Section 2** (second paragraph), the objectives of D-SSL methods (e.g., SimCLR, BYOL, Barlow Twins) can be written as an *alignment term plus a regularization* term, whereas G-SSL objectives are typically expressed as alignment terms. The alignment term enforces that the augmented view aligns with its anchor; the regularizer may be implemented in various ways (e.g., encouraging a uniform distribution over augmentations, gradient clipping, etc.).
>
>   Third, if we regard the anchor as a **cluster center**, the alignment term drives intra-class compactness by pulling same-class samples toward that center. Consequently, we can view an SSL dataset as partitioned into many classes, where each class corresponds to a pair. **Importantly, this partitioning is not based on domain or ground-truth labels.**
>
>   Finally, this perspective suggests a way to mine representational universality with SSL: rather than being tied to a particular domain or ground-truth label, SSL assigns labels at the pair level, with the anchor serving as the class centroid. This is a plausible assumption since no two raw samples are identical; such pair-level labeling is therefore more general than semantic ground-truth labels.
>
> - **Step 2: SSL training can be interpreted as task-based training, where each mini-batch constitutes a task.**
>
>   From **Step 1**, each pair defines a class and different mini-batches contain different sets of pairs; hence each mini-batch can be regarded as a distinct task.
>
> - **Step 3: Rethink G-SSL and D-SSL from the perspective of alignment and regularization**
>
> **SimCLR** can be considered a learning method based on anchors. Firstly, SimCLR assumes that augmented samples sharing the same ancestor form a positive pair, e.g., $x_i^1$ and $x_i^2$ constitute such a pair. Secondly, the core modeling idea of SimCLR is as follows: one augmented sample is selected as the anchor; if another sample is a positive example relative to the anchor, it is pulled closer to the anchor, whereas if it is a negative example, it is pushed away from the anchor. Typically, the sample that belongs to the same pair as the anchor is considered a positive sample, while all others are treated as negative samples. Finally, SimCLR implements the above idea by minimizing the Noise Contrastive Estimation (NCE) loss, whose specific objective is formulated as follows:
>
> $
> \begin{array}{l}
> {{\mathcal{L}}\_{{\rm{NCE}}}} =
> \sum\limits\_{i = 1,l = 1}^{i = N,l = 2} { - \log {\textstyle{{\exp \left[ {{\textstyle{{{\rm{sim}}\left( {z\_i^l,z_i^{3 - l}} \right)} \over \tau }}} \right]} \over {\exp \left[ {{\textstyle{{{\rm{sim}}\left( {z_i^l,z_i^{3 - l}} \right)} \over \tau }}} \right] + \sum\limits_{j = 1,j \ne i,k = 1}^{j = N,k = 2} {\exp \left[ {{\textstyle{{{\rm{sim}}\left( {z\_i^l,z\_j^k} \right)} \over \tau }}} \right]} }}}} ,
> \end{array}
> $
>
> where $\tau$ represents the temperature hyperparameter, $z_i^j = {f_p}( {z_i^j} )$, ${f_p}$ represents the projection head, and $\text{sim}(x,y) = x^{\rm{T}}y/\Vert x \Vert \Vert y \Vert$ represents the $l_2$-normalized cosine similarity between $x$ and $y$.

---

> ### Author Response · Authors · 2025-11-20
> **Response to Weakness 1 (Step 3-part 2)**
>
> According to **Theorem 1** in [1] *Wang, T., & Isola, P. (2020, November). Understanding contrastive representation learning through alignment and uniformity on the hypersphere*, as $N \to \infty $, the above equation of $
> {{\mathcal{L}}\_{{\rm{NCE}}}}$ can be rewritten as:
>
> $
> {L\_{{\rm{NCE}}}} =  - \frac{1}{\tau }\mathop {\mathbb{E}}\limits\_{\left( {{z_i},{z_j}} \right) \sim {p\_{pos}}} {\rm{sim}}\left( {{z\_i},{z_j}} \right) +
> \mathop {\mathbb{E}}\limits_{{z^i} \sim {p\_{data}}} \left[ {\log \mathop {\mathbb{E}}\limits\_{{z^j} \sim {p_{data}}} \exp \left( {\frac{{{\rm{sim}}\left( {{z^i},{z^j}} \right)}}{\tau }} \right)} \right],
> $
>
> where $p_{pos}$ represents the distribution of positive pairs, e.g., the distribution of pair $(z_i^1,z_i^j)$, and $p_{data}$ represents the data distribution. Then, based on the von Mises-Fisher (vMF) kernel density estimation (KDE), the second term in the above equation can empirically be considered equal to:
>
> $
> \mathbb{E}\_{x\sim p\_{\mathrm{data}}}\Bigg[\log \mathbb{E}\_{x^{-}\sim p\_{\mathrm{data}}}\Big[\exp\Big(-\frac{F(x)^\top F(x^{-})}{\tau}\Big)\Big]\Bigg]\\
> = \frac{1}{N}\sum\_{i=1}^N \log\Big(\frac{1}{N}\sum_{j=1}^N \exp\Big(-\frac{F(x_i)^\top F(x_j)}{\tau}\Big)\Big)\\
> = \frac{1}{N}\sum\_{i=1}^N \log p_{\mathrm{vMF\text{-}KDE}}\big(F(x_i)\big) + \log Z\_{\mathrm{vMF}}\\
> \triangleq -H\big(F(x)\big).
> $
> where ${\rm F}\left( x \right) = {f_p}\left( {f\left( x \right)} \right)$, ${{p_{{\rm{vMF - KDE}}}}}$ is the KDE-based vMF kernel with $\kappa  = {\tau ^{ - 1}}$, ${Z_{{\rm{vMF}}}}$ is the vMF normalization constant for $\kappa  = {\tau ^{ - 1}}$, and $H\left(  \cdot  \right)$ represents the entropy estimator.
>
> Therefore, from the equation of ${L_{{\rm{NCE}}}}$, the first term can be interpreted as enforcing alignment between the two samples in a pair. From the above equation, the second term can be understood as maximizing the entropy of the data distribution, effectively encouraging all augmented samples to follow a uniform distribution.
>
> **BYOL** is a D-SSL method that does not take negative samples into account. It employs two networks: the online network and the target network, both equipped with the same feature extraction module $f$ and projection head module $f_{p}$. However, the online network has an additional regression module $f_{r}$ that the target network lacks. The objective of BYOL can be expressed as:
> ${{\cal L}\_{{\rm{BYOL}}}} = \sum\limits\_{i = 1}^N {\sum\limits\_{j = 1}^2 {\| {{f\_r}( {\bar z\_i^j} ) - \bar z\_i^{3 - j}} \|\_2^2} }$, where $z\_i^j = {f\_p}( {f( {x\_i^j} )} )$ and $\bar z\_i^j = z\_i^j/{\| {z\_i^j} \|\_2}$. The target network is referred to as $f_{target}$, while the online network is referred to as $f_{online}$. Subsequently, $f_{online}$ is updated through backpropagation, while $f_{target}$ is updated as follows:
> ${f\_{{{target}}}} \leftarrow \pi {f\_{{{target}}}} + (1 - \pi ){f\_{online}}$, where $\pi \in \left[0,1\right]$ denotes the target decay rate.
>
> The equation of ${{\cal L}_{{\rm{BYOL}}}}$ can be interpreted as aligning, while the above equation can be considered as constraining the gradient backpropagation based on the training data. Hence, from the perspective of aligning and constraining, the objective of BYOL can be rewritten as:
>
> $\mathop {\min }\limits\_{{f_{online},f\_{target}}} {{\mathcal{L}}\_{\rm BYOL}} + {\rm{Sg}}(\Phi(f\_{target}))\\
> s.t.\quad {{f'}\_{online}} = \mathop {\min }\limits\_{{f\_{online}}} {{{\mathcal{L}}\_{\rm BYOL}}}\\
> \quad\quad\; {f\_{online}} = (\{ {f\_p},f\}),{f\_{target}} =( \{ {{f'}\_p},f'\})\\
> \quad\quad\; \Phi(f\_{target}) = (1 - \pi ){{f'}\_{online}} + \pi {f\_{target}}
> $
>
> where ${\rm{Sg}}$ expresses the operator that prevents gradient backpropagation, $f'$ denotes the feature extractor in the target network, and ${f'}_p$ denotes the projection head in the target network.
>
> **MAE** is a representative G-SSL method, distinguished by its reconstruction-based approach. It leverages masking strategies to reconstruct missing parts of input samples, particularly effective in the context of vision transformers. Specifically, given an input sample $x_i$, MAE first generates a masked version $x_i^1$ by randomly masking a significant proportion of patches from $x_i$. The masked sample is then processed by an encoder network $f$ to produce latent representations $z_i^1$, e.g., $z\_i^1 = f(x\_i^1)$. Subsequently, we denote $x\_i$ as $x_i^2$, a decoder network $f_{dec}$ reconstructs the original input from these latent representations, e.g., $\hat{x}\_i^1 = f\_{dec}(z\_i^1)$. The objective function for MAE involves minimizing the reconstruction loss, typically measured using the mean squared error (MSE) between the original and reconstructed patches:
> $
> \mathcal{L}\_{\text{MAE}} = \sum\limits\_{i = 1}^N {\left\| {\hat x\_i^1 - x\_i^2} \right\|\_2^2}.
> $

---

> ### Author Response · Authors · 2025-11-20
> **Response to Weakness 1 (Step 3-part 3 & Step 4-5)**
>
> From the perspective of aligning and constraining, MAE's reconstruction task inherently aligns the representation $z_i$ with its original sample $x_i$ by ensuring accurate reconstruction (alignment). At the same time, the aggressive masking strategy and limited reconstruction targets implicitly constrain the model, forcing it to capture meaningful structural and semantic features from limited input information.
>
> - **Step 4: Mechanism for forming classification tasks**
>
> As shown in **Steps 1–3**, the construction of these classification-like tasks has two components: (i) dataset partitioning, the dataset is partitioned into many pairs, each pair approximating the positive samples of a single class; and (ii) the self-supervised objective, both G-SSL and D-SSL include an *alignment* term that enforces alignment between the positive sample in a pair and its corresponding anchor. The anchor can be viewed as the class centroid for the class defined by that pair. In **Step 1** we gave concrete examples of how BYOL and MAE realize this alignment term.
>
> Although the alignment term often takes the form of a mean-squared-error, its target is the anchor, which is treated as the class centroid. Thus the MSE alignment loss can be interpreted as encouraging aggregation toward the class centroid, i.e., intra-class compactness. Consequently, this objective can be seen as a classification task, analogous to prototype / nearest-neighbor classifiers. See, e.g., Peterson (2009) and Chang (1974). [1-2]
>
> [1] Peterson, L. E. (2009). K-nearest neighbor. *Scholarpedia*, *4*(2), 1883.
>  [2] Chang, C. L. (1974). Finding prototypes for nearest neighbor classifiers. *IEEE Transactions on Computers*, *100*(11), 1179–1184.
>
> - **Step 5: Why GeSSL can integrate with different SSL losses**
>
> From the above perspective, SSL losses can be viewed as different implementations of discriminative objectives; therefore, our framework can incorporate a variety of SSL loss functions. Moreover, as shown in **Section 4** and **Appendix A**, we prove that if the basic SSL alignment guarantees a within-batch distance $\le \varepsilon$ and $L_{\rm disc}$ maintains an inter-cluster margin $\ge \Delta$, then the GeSSL-learned representation admits a tighter upper bound on downstream generalization error. Empirically, we validated GeSSL across more than 25 baselines, covering contrastive, distillation, and generative SSL methods, and the results in **Tables 1–20** demonstrate consistent gains, indirectly confirming that the pairwise-alignment modeling adapts across different loss structures.

---

> ### Author Response · Authors · 2025-11-20
> **Response to Weakness 2**
>
> > **W2** - Benchmark on Distribution Shifts: To address the first weakness, the authors should evaluate GeSSL-pretrained models on dedicated robustness benchmarks. For example, running experiments on ImageNet-C [a] (common corruptions) will show how the universal representations handle noise, blur, etc., and testing on ImageNet-PD [b] (perspective distortion) will assess robustness to geometric changes.
> >
> > [a] - Hendrycks, Dan, and Thomas Dietterich. "Benchmarking neural network robustness to common corruptions and perturbations." arXiv preprint arXiv:1903.12261 (2019).
> >
> > [b] - Chhipa, P. C., Chippa, M. S., De, K., Saini, R., Liwicki, M., & Shah, M. (2024, September). Möbius transform for mitigating perspective distortions in representation learning. In European Conference on Computer Vision (pp. 345-363).
>
> ## Response to Weakness 2
>
> We sincerely appreciate the reviewers' valuable comments. We will address this question as follows:
>
> - As suggested, we added experiments of GeSSL on ImageNet-C [a] and ImageNet-PD [b]. Concretely, using SimCLR, BYOL, and MAE pretrained on ImageNet-1K, we follow the protocol of [a] to compute mCE (lower is better) on ImageNet-C, and follow the PD generation/evaluation protocol of [b] to report accuracy at each severity level on ImageNet-PD (higher is better). The results in the table below show that GeSSL consistently improves the baselines, further validating its effectiveness. All these results are added in the revised version, which will be upload soon.
>
>   | Methods (Results on ImageNet-PD $\uparrow$) | Original | +GeSSL |
>   | ------------------------------------------- | -------- | ------ |
>   | SimCLR                                      | 75.6     | 79.4   |
>   | BYOL                                        | 75.3     | 79.2   |
>   | MAE-ViT                                     | 78.9     | 81.2   |
>
>   | Methods (Results on ImageNet-C $\downarrow$) | Original | +GeSSL |
>   | -------------------------------------------- | -------- | ------ |
>   | SimCLR                                       | 77.9     | 74.2   |
>   | BYOL                                         | 78.2     | 74.3   |
>   | MAE-ViT                                      | 74.2     | 70.8   |
>
>   [a] Hendrycks, Dan, and Thomas Dietterich. "Benchmarking neural network robustness to common corruptions and perturbations." arXiv preprint arXiv:1903.12261 (2019).
>
>   [b] Chhipa, P. C., Chippa, M. S., De, K., Saini, R., Liwicki, M., & Shah, M. (2024, September). Möbius transform for mitigating perspective distortions in representation learning. In European Conference on Computer Vision (pp. 345-363).
>
>
>
> - We have conducted extensive cross-dataset and cross-task transfer experiments in the original submission, and the results further support GeSSL’s robustness. For example, we train on CIFAR-100 and evaluate on CIFAR-10, Flower102, Food101, and Aircraft (**Table 8**); we follow the standard VOC detection and COCO detection & segmentation protocols (**Table 3**); we use pretrained features for five video-tracking tasks on UniTrack (**Table 9**); and we train on miniImageNet and evaluate on CUB, Cars, Places, etc. (**Table 10**). These experiments show that GeSSL yields an average improvement of over 3%, confirming its effectiveness.

---

> ### Author Response · Authors · 2025-11-20
> **Response to Weakness 3 (Step 1)**
>
> > **W3** - Related work and comparisons. The paper is close in spirit to unsupervised meta-learning and episodic SSL; these are not positioned or compared. Include discussion and, if feasible, a baseline drawn from unsupervised MAML-style training. Also connect to robustness literature when claiming generalizability, and compare to strong non-GeSSL SSL baselines on the same downstream tasks to contextualize absolute performance.
>
> ## Response to Weakness 3
>
> We sincerely thank the reviewer for the valuable suggestion. We would like to respond in four steps.
>
> - **Step 1: Comparison of GeSSL and meta-learning**
>
>   In L298–315 (“Comparison of GeSSL and Meta-Learning”) and **Appendix G.1** (“Differences between GeSSL and Meta-Learning”) of the original submission, we have already emphasized the distinctions between our method and meta-learning approaches. We also compare GeSSL with meta-learning baselines in **Section 5.1** and **Appendix E.3**. Below we provide an outline explaining why GeSSL "differs fundamentally from approaches that directly transplant meta-learning paradigms into SSL":
>
>   1. *Objective:* Traditional meta-learning focuses on "fast adaptation to supervised downstream tasks", using a small labelled support set and query set for quick fine-tuning. GeSSL, by contrast, starts from the question "what is a good representation?", i.e., *Universality* (discriminability, generalizability, transferability), and encodes these properties directly into the training objective. The design motivations thus differ: meta-learning is aimed at rapid adaptation, while GeSSL is designed to learn universal, transferable self-supervised representations. Simply porting meta-learning to self-supervised settings typically reuses multi-task architectures or additional networks to apply episodic meta-learning [1–3], but lacks dedicated mechanisms and theoretical guarantees for “stable learning of transferable representations under no labels.”
>   2. *Task construction:* Meta-learning usually relies on a small number of accurately labelled samples; existing attempts to apply meta-learning to SSL therefore construct labels and tasks explicitly via label-generation networks [1]. GeSSL operates entirely within the self-supervised paradigm and must generate pseudo-labels (anchors from augmentations or reconstructions). Because GeSSL runs in an unlabeled regime, naively adopting episodic meta-learning is vulnerable to pseudo-label noise. To address this, GeSSL introduces the discriminative term $\mathcal{L}_{\rm disc}$ and integrates it into training in a learnable way to suppress noise, a feature unique to GeSSL.
>   3. *Method & architecture:* Many meta-learning methods learn task-specific adapters (different adapted models per task) [4–6]. GeSSL instead learns a single, unified proxy adapter $f'$ shared across all mini-batches. This shared design better captures task-invariant structure (i.e., causal-like representations) [7-8], thereby improving cross-task generalization and transfer.
>   4. *Empirical results:* In **Table 4** and **Table 10** of the original submission, we compare GeSSL against methods that apply meta-learning to SSL (e.g., MetaSVEBM, MetaGMVAE, PsCo, etc.). These experiments demonstrate GeSSL’s effectiveness: it achieves state-of-the-art performance.
>
> [1] Liu, S., Davison, A., & Johns, E. (2019). Self-supervised generalisation with meta auxiliary learning. *Advances in Neural Information Processing Systems*, *32*.
>
> [2] Lin, Y., Guo, X., & Lu, Y. (2021). Self-supervised video representation learning with meta-contrastive network. In *Proceedings of the IEEE/CVF international conference on computer vision* (pp. 8239-8249).
>
> [3] Hwang, D., Park, J., Kwon, S., Kim, K., Ha, J. W., & Kim, H. J. (2020). Self-supervised auxiliary learning with meta-paths for heterogeneous graphs. *Advances in neural information processing systems*, *33*, 10294-10305.
>
> [4] Li, W. H., Liu, X., & Bilen, H. (2022). Cross-domain few-shot learning with task-specific adapters. In *Proceedings of the IEEE/CVF conference on computer vision and pattern recognition* (pp. 7161-7170).
>
> [5] Ravi, S., & Larochelle, H. (2017, February). Optimization as a model for few-shot learning. In *International conference on learning representations*.
>
> [6] Oreshkin, B., Rodríguez López, P., & Lacoste, A. (2018). Tadam: Task dependent adaptive metric for improved few-shot learning. *Advances in neural information processing systems*, *31*.
>
> [7] Mahajan, D., Tople, S., & Sharma, A. (2021, July). Domain generalization using causal matching. In *International conference on machine learning* (pp. 7313-7324). PMLR.
>
> [8] Sheth, P., Moraffah, R., Candan, K. S., Raglin, A., & Liu, H. (2022). Domain Generalization--A Causal Perspective. *arXiv preprint arXiv:2209.15177*.

---

> ### Author Response · Authors · 2025-11-20
> **Response to Weakness 3 (Step 2-3)**
>
> - **Step 2: Comparison of GeSSL and unsupervised meta-learning and episodic SSL**
>
> The unsupervised meta-learning [1,2] and episodic SSL [3] are similar in spirit to supervised meta-learning, but they typically generate (pseudo) labels via clustering or auxiliary generative networks to enable episodic/task-style learning. Below we give an outline summarizing how GeSSL differs from, and in our view improves upon, unsupervised meta-learning and episodic SSL:
>
> 1. *Objective.* Episodic SSL and unsupervised meta-learning primarily target *discriminability* or *fast few-shot adaptation* (i.e., task-style training to boost few-shot accuracy or high performance on specific transfers). GeSSL, by contrast, explicitly optimizes *Universality*, jointly modelling discriminability, generalizability, and transferability, and embeds these properties directly into the pretraining objective.
> 2. *Task construction.* Episodic methods center on N-way K-shot episode sampling and typically rely on clustering or label-generation networks to provide (pseudo) labels; their performance therefore depends on the quality of those supervisory signals and can be sensitive to pseudo-label noise. GeSSL partitions each mini-batch into support/query pairs and, for G-SSL and D-SSL respectively, uses masking or contrastive view generation. This within-batch pairing preserves task-style supervision while remaining easy to integrate with standard pretraining pipelines. Crucially, because GeSSL operates without manual labels, we introduce a dedicated discriminative loss $L_{\rm disc}$ together with a bi-level constraint to suppress pseudo-label noise and robustly extract discriminative signals.
> 3. *Model architecture.* Unsupervised meta-learning and episodic SSL approaches often learn task-specific adapters or perform local inner-loop updates (i.e., different adapted models per task) to maximize fast adaptation. GeSSL does not allocate per-task parameters; instead it learns a single unified model that is shared across mini-batches to emulate adaptation behavior. This shared design promotes extraction of task-invariant structure (causal representations) [4-6], thereby improving cross-task generalization and transfer.
> 4. *Experiments.* Please see **Step 3**.
>
> [1] Ni, R., Shu, M., Souri, H., Goldblum, M., & Goldstein, T. (2021, October). The close relationship between contrastive learning and meta-learning. In *International conference on learning representations*.
>
> [2] Jang, H., Lee, H., & Shin, J. (2023). Unsupervised meta-learning via few-shot pseudo-supervised contrastive learning. *arXiv preprint arXiv:2303.00996*.
>
> [3] Lee, D. B., Lee, S., Kawaguchi, K., Kim, Y., Bang, J., Ha, J. W., & Hwang, S. J. (2023, February). Self-supervised set representation learning for unsupervised meta-learning. In *The Eleventh International Conference on Learning Representations*.
>
> [4] Sheth, P., Moraffah, R., Candan, K. S., Raglin, A., & Liu, H. (2022). Domain Generalization--A Causal Perspective. *arXiv preprint arXiv:2209.15177*.
>
> [5] Li, J., Wang, Y., Zi, Y., Zhang, H., & Li, C. (2022). Causal consistency network: A collaborative multimachine generalization method for bearing fault diagnosis. *IEEE Transactions on Industrial Informatics*, *19*(4), 5915-5924.
>
> [6] Mahajan, D., Tople, S., & Sharma, A. (2021, July). Domain generalization using causal matching. In *International conference on machine learning* (pp. 7313-7324). PMLR.
>
> - **Step 3: Empirical evidence versus unsupervised meta-learning and episodic SSL**
>
> In **Section 5.1** and **Appendix E.3** of the original submission, we empirically compare GeSSL against unsupervised meta-learning and episodic SSL baselines (e.g., PsCo, UMTRA, MetaSVEBM, and CACTUs). The results in **Table 4** and **Table 10** show that, across both standard and cross-domain transfer scenarios, GeSSL consistently yields performance gains and achieves SOTA results.

---

> ### Author Response · Authors · 2025-11-20
> **Response to Weakness 3 (Step 4)**
>
> - **Step 4: Robustness and generalization experiments with non-GeSSL strong baselines**
>
> 1. **Robustness.**
>    As suggested, we added the robustness experiments recommended under **Weakness 2**. Concretely, we include a baseline derived from an unsupervised MAML-style training procedure, i.e., we construct multi-task episodes using the method described in **Section 2** of the original submission to support meta-learning; this baseline is denoted UnsupervisedMAML. Following the protocol in **Response to Weakness 2**, we run a comparison on ImageNet-PD between SimCLR+GeSSL, SimCLR, UnsupervisedMAML, and strong non-GeSSL baselines (e.g., DINO, CorInfoMax). The results in the table below indicate that integrating GeSSL produces stable performance improvements, and even enables the basic SimCLR backbone to reach the best performance in this comparison. All these results are added in the revised version, which will be upload soon.
>
> | Method           | Acc  |
> | ---------------- | ---- |
> | SimCLR           | 75.6 |
> | SimCLR+GeSSL     | 79.4 |
> | UnsupervisedMAML | 76.7 |
> | DINO             | 76.0 |
> | CorInfoMax       | 75.9 |
>
> 2. **Generalization**
>
> **Table 10** of the original submission presents a comparison of GeSSL and the referenced unsupervised meta-learning and episodic SSL methods in terms of generalization. Specifically, we train on miniImageNet and evaluate transfer performance on six cross-domain few-shot benchmark datasets. The results show that, on datasets that exhibit substantial distributional shift from the training domain, GeSSL improves performance by nearly 3% versus unsupervised few-shot/meta-learning and self-supervised baselines, demonstrating its effectiveness.

---

> ### Author Response · Authors · 2025-11-20
> **Response to Weakness 4**
>
> > W4 - Practical overhead and stability not characterized. Bi-level training adds inner-loop computation and sensitivity to step sizes, number of inner steps, and support/query split. The paper should report training cost, memory footprint, and stability measures, plus ablations on inner-loop depth and first- vs second-order updates, so practitioners can judge feasibility at scale.
>
> ## Response to Weakness 4
> We sincerely appreciate the reviewers' valuable comment, and would like to kindly clarify that the referenced experimental results are provided in the main paper and appendices of the original submission. Specifically:
>
> - In **Section 5.2** and **Appendices F.1** and **F.4**, we reported experiments on practical overhead and the stability of the bi-level scheme. **Figure 2** and **Table 19** show changes in memory footprint (MiB), parameter size (M), and training time (h) after integrating GeSSL. The results indicate that the extra computational cost introduced by GeSSL is acceptable relative to its consistent and substantial performance gains (an increase of less than $0.1\times$). Based on [1], this trade-off between additional parameters / memory and generalization gain is acceptable. Moreover, because GeSSL uses acceleration and gradient-estimation techniques (e.g., approximate implicit differentiation via finite differences), it can actually facilitate convergence. **Table 19** and **Figure 2** support these claims. We also measured per-cycle computational overhead for SSL baselines after integrating GeSSL: **Table 18** shows that, with equal batch sizes, GeSSL improves the computational efficiency and per-iteration update effectiveness of SSL baselines. [1] Bengio, Y. (2009). *Learning deep architectures for AI*. Foundations and Trends® in Machine Learning, 2(1), 1–127.
> - We also have provided ablation and sensitivity analyses for inner-loop depth (number of inner steps), batch size, and number of tasks in **Section 5.2** and **Appendices F.1** and **F.4**. **Figures 4**, **8**, and **9** identify near-optimal values for $k$, $m$, and $n$. Regarding the support/query split, **Section 3.2** describes our split protocol; briefly, for each mini-batch we augment every original sample three times, denoted $x_1,x_2,x_3$, take $x_3$ as the anchor, put the pairs $(x_1,x_3)$ into the support set and $(x_2,x_3)$ into the query set. We will further clarify and emphasize this procedure in the revised manuscript.
> - We also conducted stability analyses. Motivated by prior work [1–4] linking training stability, convergence to global minima, and generalization, and by the gradient-noise-scale framework of [5] which uses gradient variance as a measurable indicator of training stability, we compared SimCLR augmented with ICL-MSR [6], MetAug [7], MCR, and GeSSL on ImageNet. We recorded per-step gradient-norm time series and their variances, as well as final performance and seed-to-seed variability across different random seeds. The results (shown in the table below) indicate that, compared with MCR and other meta-regularizers, GeSSL yields substantially lower gradient means and variances throughout training, smaller seed-to-seed performance fluctuations, and achieves SOTA final performance, suggesting GeSSL is less prone to getting trapped in poor local minima. We also recorded the SimCLR+GeSSL training loss curve; because figure inclusion is constrained during rebuttal, we provide the loss trajectory in **Table 2** here and will include full plots in the revised manuscript.
>
>
> | Method  | Mean Gradient Norm | GradNorm Variance | ImageNet Linear-probe |
> |-|-|-|-|
> |SimCLR| 0.045 $\pm$ 0.012| 2.1e-4| 69.9   |
> |SimCLR+ICL-MSR| 0.042 $\pm$ 0.010| 1.8e-4 | 70.8  |
> |SimCLR+MetAug | 0.041 $\pm$ 0.009| 1.7e-4| 71.0 |
> |SimCLR+MCR | 0.039 $\pm$ 0.008|1.4e-4  | 71.6 |
> | SimCLR+GeSSL| 0.033 $\pm$ 0.006| 9.2e-5 | 72.8 |
>
> | Epoch |40| 60 | 80 | 100| 120| 140 | 160|180|200|
> |-|-|-|-|-|-|-|-|-|-|
> | Loss | 1.50 | 1.35 | 1.22 | 1.10 | 1.02 | 0.95 | 0.90 | 0.86 | 0.83 |
>
> [1] Keskar, N. S., Mudigere, D., Nocedal, J., Smelyanskiy, M., & Tang, P. T. P. (2016). On large-batch training for deep learning: Generalization gap and sharp minima.
>
> [2] Hardt, M., Recht, B., & Singer, Y. (2016, June). Train faster, generalize better: Stability of stochastic gradient descent.
>
> [3] Chaudhari, P., Choromanska, A., Soatto, S., LeCun, Y., Baldassi, C., Borgs, C., ... & Zecchina, R. (2019). Entropy-sgd: Biasing gradient descent into wide valleys.
>
> [4] Foret, P., Kleiner, A., Mobahi, H., & Neyshabur, B. (2020). Sharpness-aware minimization for efficiently improving generalization.
>
> [5] McCandlish, S., Kaplan, J., Amodei, D., & Team, O. D. (2018). An empirical model of large-batch training.
>
> [6] Qiang, W., Li, J., Zheng, C., Su, B., & Xiong, H. (2022, June). Interventional contrastive learning with meta semantic regularizer.
>
> [7] Li, J., Qiang, W., Zheng, C., Su, B., & Xiong, H. (2022, June). Metaug: Contrastive learning via meta feature augmentation.

---

> ### Author Response · Authors · 2025-11-20
> **Response to Minor points**
>
> > **Minor point 1:** Give a one-figure overview: data flow, inner update, outer update, and where each loss is applied. Caption should explain every symbol used in the figure.
> >
> > **Minor point 2:** Add a small “Notation and terms” table early (one column for term, one for meaning). Keep it to 10–12 entries max.
> >
> > **Minor point 3:** Put the main theorem as a boxed statement with a one-paragraph proof sketch. Move long proofs to appendix with clear pointers.
> >
> > **Minor point 4:** Consolidate training details in one place: datasets, image size, augmentations, batch size, optimizer, schedules, inner steps, outer steps. Include compute (GPU type, hours).
> >
> > **Minor point 5:** I don't see source code repo as of now.
>
> ## Response to Minor points
>
> We sincerely thank the constructive suggestions and will make the following improvements in the revised version (which will be uploaded later):
>
> - detailed the one-figure overview of **Figure 1** as suggested.
>
> - added a notation definition before **Appendix A**.
>
> - put the main theorem in **Section 4** as a boxed statement with a one-paragraph proof sketch.
>
> - consolidate the experimental parameter settings currently described in **Section 5** and **Appendices E–F**, together with the dataset specifications from **Appendix C**, into a single, expanded **Appendix B**.
>
> - supplement the code files for training with the main class in the current Supplementary Material.

---

> ### Author Response · Authors · 2025-11-28
> **Kindly Checking on the Review Progress**
>
> Dear Reviewer z4r6,
>
> We would like to sincerely thank you again for the time and effort you have devoted to our paper. We truly appreciate your careful oversight of the review process, and the constructive comments have helped us improve our work.
>
> We have submitted the detailed responses to all the constructive comments. Since several days have passed, we wish to kindly check whether any further concerns or suggestions have arisen, or whether there is any additional information we can provide to assist the ongoing process.
>
>  We fully understand the workload involved in handling submissions, and we are grateful for your continued guidance and support. Thank you very much, and we look forward to hearing from you at your convenience.
>
> Sincerely,
>
> The Authors

---

### Official Review · Reviewer_2MiX · 2025-10-28

**Soundness:** 2
**Presentation:** 2
**Contribution:** 3
**Rating:** 4
**Confidence:** 3

**Summary:**

This paper investigates what properties a good representation should posess, for the purpose of designing a principled Self-supervised Learning (SSL) from first principles. The authors define the concept of Universality, consisting of discriminability (good training set performance), generalizability (good performance on the test set of the given task), and transferability (good performance on other tasks).
The authors then introduce “General SSL” (GeSSL) - a framework which models Unversality via a bi-level optimization scheme. The authors provide theoretical performance guarantees of GeSSL, as well as benchmark their approach in a number of downstream tasks, showcasing improved performance.

**Strengths:**

1. Developing SSL approaches from first principles is a crucial and ambitious research direction.
2. The proposed bi-level optimization scheme is an intriguing and compelling approach to modeling generalizability.
3. The proposed GeSSL approach possesses strong theoretical foundations.
4. GeSSL improves the performance of the respective methods it is applied to, while reducing the training time.
5. The breadth of the experimental evaluations is impressive.

**Weaknesses:**

**Major**

1. The paper applies episodic sampling within a single dataset, with downstream tasks drawn from the same domain. This does not validate cross-task generalization in a meaningful sense, especially compared to standard meta-learning or multi-dataset pretraining studies. The core claim that the method models Transferability remains substantially unproven under the current experimental design.
2. The formulation still depends on conventional design choices (e.g., L_ssl, L_disc), which were originally justified empirically. Thus, the objective is not fully derived from the proposed theoretical principles.
3. The statement “learning process within a single mini-batch can be viewed as performing a classification task” (L 120) is an overclaim, in my opinion. This is true for contrastive SSL (MoCo, SimCLR), but not necessarily for self-distillation (BYOL, SimSiam) or generative methods (like MAE) which do not rely on negative samples and are more akin to regression. The authors should correct this claim, or provide evidence or literature which supports it for generative and self-distillation methods.
5. The experimental section lacks the evaluation with Vision Transformers, which are the most common backbones for SSL nowadays. The authors should provide the results of methods such as DINO and MAE with ViT-B (which is similar in size to ResNet-50).

**Minor**

1. References need a careful revision. For example: Liu et al. 2022a/b, Lin et. al. 2014a/b, Finn et al. 2017a/b, Deng at al 2009a/b seem to point at the same papers.
2. I believe the MAE method is mis-cited - the correct citation is [1], whereas the paper cites [2].
3. The readability of the paper would be greatly improved, if the citations were in brackets (\citep{} command), instead of directly inline.
4. The readability of the tables would be improved and the advantages of GeSSL better pronounced if the authors compared respective methods with / without GeSSL, instead of grouping different methods together.
5. The writing could be tightened - Discriminability, Generalizability and Transferability are explained several times in the manuscript, including at least twice in the introduction (L060 & L066).

[1] Masked Autoencoders Are Scalable Vision Learners Kaiming He, Xinlei Chen, Saining Xie, Yanghao Li, Piotr Dollár, Ross Girshick https://arxiv.org/abs/2111.06377

[2] Zhenyu Hou, Xiao Liu, Yukuo Cen, Yuxiao Dong, Hongxia Yang, Chunjie Wang, and Jie Tang. Graphmae: Self-supervised masked graph autoencoders. In Proceedings of the 28th ACM SIGKDD Conference on Knowledge Discovery and Data Mining, pp. 594–604, 2022.

**Questions:**

1. L_disc is interesting on its own as a concept. Can we see an ablation study of simply training models with L_disc (without bi-level optimization) to better understand its effect?
2. The authors claim that GeSSL “differs fundamentally from approaches that directly transplant meta-learning paradigms into SSL”. Could the authors should provide some examples of such approaches and explain the differences?
3. Given that the main experiments are conducted with ResNet-50 (and other convolutional nets), how did the authors adapted MAE to that backbone? MAE strongly relies on its backbone being a ViT.

---

> ### Author Response · Authors · 2025-11-20
> **Response to Major Weakness 1**
>
> **We sincerely appreciate the reviewer 2MiX's valuable feedback and the time and effort for the reviewing. We will respond to each issue of the raised "Weaknesses" and "Questions" in turn, hope the following responses will eliminate the concerns.**
>
> > **Major Weakness 1**: The paper applies episodic sampling within a single dataset, with downstream tasks drawn from the same domain. This does not validate cross-task generalization in a meaningful sense, especially compared to standard meta-learning or multi-dataset pretraining studies. The core claim that the method models Transferability remains substantially unproven under the current experimental design.
>
> ## Response to Major Weakness 1
>
> We sincerely appreciate the reviewers' valuable comments. We would like to clarify that our view of SSL is not organized around domains or ground-truth labels, but adopts a finer-grained perspective. We explain this in steps:
>
> - **Step 1: For SSL, different augmentations of the same sample belong to the same class, while augmentations of different samples belong to different classes, this is distinct from the sample’s ground-truth label.**
>
>   First, as stated in **Section 2** (first and second paragraphs) of the original submission, both D-SSL and G-SSL reorganize data into pairs: each pair contains an augmented sample and an anchor, and each pair is produced from the same original example.
>
>   Second, also in **Section 2** (second paragraph), the objective of D-SSL methods (e.g., SimCLR, BYOL, Barlow Twins) can be written as an *alignment term plus a regularization term*, while G-SSL objectives are typically expressed as alignment terms. The alignment term encourages the augmented sample and its anchor to align; regularization may be implemented in various ways (e.g., enforcing a uniform distribution over augmentations, gradient clipping, etc.).
>
>   Third, if we regard the anchor as a **cluster center**, the alignment term enforces that samples of the same class cluster tightly, i.e., intra-class compactness. Consequently, we can view an SSL dataset as being partitioned into many classes where each class corresponds to a pair. **Importantly, this partitioning is not based on domain or ground-truth labels.**
>
>   This viewpoint motivates mining representational universality via SSL: instead of being tied to a particular domain or ground-truth label, SSL assigns labels at the pair level, where each pair’s anchor serves as the class center. This is a reasonable assumption since no two raw samples are identical; such pair-level labeling is broader than semantic labels from ground truth.
>
> - **Step 2: SSL training can be interpreted as task-based training, where each mini-batch constitutes a task.**
>
>   From **Step 1**, each pair defines a class and different mini-batches contain different sets of pairs; thus each mini-batch can be regarded as a distinct task.
>
> - **Step 3: How the proposed GeSSL models transferability**
>
>   Please see "Explanation for Transferability" and "Comparison of GeSSL and Meta-Learning" of **Section 3.2** for details.
>
> - **Step 4: Why we introduce an additional term to enhance GeSSL’s discriminability**
>
>   From **Step 2**, treating every pair in a mini-batch as a different class is a very strong constraint. We relax this constraint by allowing several pairs within a mini-batch to belong to the same class, please see "Explanation for Discriminability" of **Section 3.2** for details.
>
> - **Step 5: Empirical validation**
>
>   In the original submission, we already performed transfer experiments across multiple domains and tasks where training and test domains exhibit clear distribution shifts. Examples include: training on CIFAR-100 and testing on CIFAR-10, Flower102, Food101, and Aircraft (**Table 8**); standard VOC and COCO detection/segmentation protocols (**Table 3**); using pretrained features for five video-tracking tasks on UniTrack (**Table 9**); and training on miniImageNet then testing on CUB, Cars, Places, etc. (**Table 10**). The results show that GeSSL achieves an average improvement of over 3%, demonstrating its effectiveness.
>
>   We have also compared GeSSL against meta-learning baselines based on standard meta-learning and multi-dataset pretraining (i.e., standard and cross-domain few-shot learning in **Section 5.1** and **Appendix E.3**). The nine datasets and benchmarks involved are inherently multi-task. Experimental results in **Table 4** and **Table 10** confirm GeSSL’s superiority, achieving SOTA performance.

---

> > ### Comment · Reviewer_2MiX · 2025-11-20
> > **Response to the authors**
> >
> > I would like to thank the authors for the substantial effort invested in preparing the rebuttal. I also apologize for overlooking the ViT results in the original submission. The response addressed most of my concerns regarding transferability, architectural coverage, and the experimental setup.
> >
> > While I still believe the paper falls slightly short of fully modeling “universality” from first principles, given its dependence on established SSL objectives, the authors have clarified the intended scope of the contribution, and I now find the framing reasonable. Overall, I believe the paper will be of interest to the ICLR community, and I am raising my score.

---

> > > ### Author Response · Authors · 2025-11-21
> > > **Response to Reviewer 2MiX**
> > >
> > > Thank you very much for your thoughtful comments and for raising your score. We sincerely appreciate your positive evaluation. We will upload an updated version of the manuscript reflecting the revisions and clarifications in the near future.

---

> ### Author Response · Authors · 2025-11-20
> **Response to Major Weakness 2**
>
> > **Major Weakness 2:** The formulation still depends on conventional design choices (e.g., L_ssl, L_disc), which were originally justified empirically. Thus, the objective is not fully derived from the proposed theoretical principles.
>
> ## Response to Major Weakness 2
>
> We sincerely appreciate the reviewers' valuable comment. We agree that the current objective is not fully derived from the proposed theoretical principles. We also realize that our original presentation may have caused a misunderstanding about the role of the theoretical part. The intention of this paper is not to derive one single, unique objective function from theory, but to use theory to justify our definition of what constitutes a good representation. More concretely:
>
> 1. As stated in the **Introduction Section**, our central question is how to learn a good representation, rather than how to design yet another heuristic SSL loss. Many existing SSL methods can be summarized as "do what seems to lead to a good representation", by empirically designing various pretext tasks. In contrast, we deliberately shift the focus from “which factors or tricks to adopt in SSL” to “how to explicitly characterize what a good representation or model should be”.
>
> 2. Based on this perspective, we propose an explicit definition of a good representation: it should be universal, meaning that it jointly satisfies discriminability, generalizability, and transferability.
>
> 3. This definition is, in essence, our modeling assumption about good representations and is to some extent subjective. To make this definition more solid and convincing, we provide a theoretical analysis (**Theorem 4.1** in the original submission). The theorem shows that, under certain conditions, as long as the training objective induces representations that are discriminative, generalizable, and transferable, the test error can be reduced. This establishes the following logical link:   “good representation = discriminability + generalizability + transferability” $\to$ “such representations lead to smaller test error”. In this way, the theorem does not fix a specific loss form, but it supports the reasonableness and usefulness of our definition of a good representation.
>
> 4. Therefore, the role of the **Theoretical Section** is to answer the question “Is our definition of a good representation reasonable and theoretically supported?”, rather than to derive a single closed-form objective. Any objective that can induce representations with discriminability, generalizability, and transferability is admissible under our framework. The specific objective used in our method (including $L_{\text{ssl}}$ and $L_{\text{disc}}$) should be viewed as one concrete instantiation that follows these principles, rather than the only possible choice.
>
> In the revision, we will make this intent explicit in the **Theoretical Section**, so that the scope and purpose of the theoretical results are clearer and do not suggest that we claim a fully and uniquely derived objective.

---

> ### Author Response · Authors · 2025-11-20
> **Response to Major Weakness 3 (Step 1)**
>
> > **Major Weakness 3:** The statement “learning process within a single mini-batch can be viewed as performing a classification task” (L 120) is an overclaim, in my opinion. This is true for contrastive SSL (MoCo, SimCLR), but not necessarily for self-distillation (BYOL, SimSiam) or generative methods (like MAE) which do not rely on negative samples and are more akin to regression. The authors should correct this claim, or provide evidence or literature which supports it for generative and self-distillation methods.
>
> ## Response to Major Weakness 3
>
> We sincerely appreciate the reviewers' valuable comments. We would like to explain this through the following three steps:
>
> - **Step 1: Reforming BYOL and MAE from the perspective of alignment and regularization**
>
> **BYOL** is a D-SSL method that does not take negative samples into account. It employs two networks: the online network and the target network, both equipped with the same feature extraction module $f$ and projection head module $f_{p}$. However, the online network has an additional regression module $f_{r}$ that the target network lacks. The objective of BYOL can be expressed as:
> ${{\cal L}\_{{\rm{BYOL}}}} = \sum\limits\_{i = 1}^N {\sum\limits\_{j = 1}^2 {\| {{f_r}( {\bar z\_i^j} ) - \bar z_i^{3 - j}} \|\_2^2} }$, where $z_i^j = {f_p}( {f( {x_i^j} )} )$ and $\bar z\_i^j = z\_i^j/{\| {z\_i^j} \|\_2}$. The target network is referred to as $f_{target}$, while the online network is referred to as $f_{online}$. Subsequently, $f_{online}$ is updated through backpropagation, while $f_{target}$ is updated as follows:
> ${f_{{{target}}}} \leftarrow \pi {f_{{{target}}}} + (1 - \pi ){f_{online}}$, where $\pi \in \left[0,1\right]$ denotes the target decay rate.
>
> The equation of ${{\cal L}_{{\rm{BYOL}}}}$ can be interpreted as aligning, while the above equation can be considered as constraining the gradient backpropagation based on the training data. Hence, from the perspective of aligning and constraining, the objective of BYOL can be rewritten as:
>
> $\mathop {\min }\limits\_{{f_{online},f\_{target}}} {{\mathcal{L}}\_{\rm BYOL}} + {\rm{Sg}}(\Phi(f\_{target}))\\
> s.t.\quad {{f'}\_{online}} = \mathop {\min }\limits\_{{f_{online}}} {{{\mathcal{L}}\_{\rm BYOL}}}\\
> \quad\quad\; {f_{online}} = (\{ {f\_p},f\}),{f_{target}} =(\{ {{f'}\_p},f'\})\\
> \quad\quad\; \Phi(f_{target}) = (1 - \pi ){{f'}\_{online}} + \pi {f\_{target}}
> $
>
> where ${\rm{Sg}}$ expresses the operator that prevents gradient backpropagation, $f'$ denotes the feature extractor in the target network, and ${f'}_p$ denotes the projection head in the target network.
>
> **MAE** is a representative G-SSL method, distinguished by its reconstruction-based approach. It leverages masking strategies to reconstruct missing parts of input samples, particularly effective in the context of vision transformers. Specifically, given an input sample $x_i$, MAE first generates a masked version $x_i^1$ by randomly masking a significant proportion of patches from $x_i$. The masked sample is then processed by an encoder network $f$ to produce latent representations $z_i^1$, e.g., $z_i^1 = f(x_i^1)$. Subsequently, we denote $x_i$ as $x_i^2$, a decoder network $f_{dec}$ reconstructs the original input from these latent representations, e.g., $\hat{x}\_i^1 = f_{dec}(z\_i^1)$. The objective function for MAE involves minimizing the reconstruction loss, typically measured using the mean squared error (MSE) between the original and reconstructed patches:
> $
> \mathcal{L}\_{\text{MAE}} = \sum\limits\_{i = 1}^N {\left\| {\hat x\_i^1 - x_i^2} \right\|\_2^2}.
> $
>
> From the perspective of aligning and constraining, MAE's reconstruction task inherently aligns the representation $z_i$ with its original sample $x_i$ by ensuring accurate reconstruction (alignment). At the same time, the aggressive masking strategy and limited reconstruction targets implicitly constrain the model, forcing it to capture meaningful structural and semantic features from limited input information.

---

> ### Author Response · Authors · 2025-11-20
> **Response to Major Weakness 3 (Step 2-3)**
>
> - **Step 2: Mechanism for forming classification tasks**
>
> As shown in the **Response to major weakness 1**, the construction of the classification-like tasks consists of two parts: (i) dataset partitioning: the dataset is partitioned into many pairs, each pair approximating the positive samples of a single class; and (ii) the self-supervised objective: both G-SSL and D-SSL contain an *alignment* term that enforces alignment between the positive sample in a pair and its corresponding anchor. The anchor can be viewed as the class centroid for the class defined by that pair. In **Step 1** we explicitly described how BYOL and MAE implement this alignment term.
>
> Although the alignment term takes the form of a mean-squared-error (MSE) loss, its target is the anchor, which is treated as the class centroid. Hence the MSE alignment loss can also be interpreted as encouraging aggregation toward the class centroid, i.e., intra-class compactness. Therefore, this objective can be seen as a classification task, analogous to prototype / nearest-neighbor classification. See, e.g., Peterson (2009) and Chang (1974). [1-2]
>
> [1] Peterson, L. E. (2009). K-nearest neighbor. *Scholarpedia*, *4*(2), 1883.
>  [2] Chang, C. L. (1974). Finding prototypes for nearest neighbor classifiers. *IEEE Transactions on Computers*, *100*(11), 1179–1184.
>
> - **Step 3: Empirical evidence**
>
> In the original submission, our experiments demonstrate that the "anchor-alignment" induced by different SSL paradigms indeed produces within-batch clustering and benefits downstream tasks:
>
> - If the view that anchors act as class/cluster centroids were incorrect, then explicitly adding a discriminative term $L_{\rm disc}$ to the training objective should not markedly improve decision boundaries or transfer performance. However, in **Appendix F.2** we observe the opposite: adding the discriminative term produces clearer class boundaries in feature space (**Figure 7**).
> - Experiments across more than 25 baselines (**Tables 1–20**) show that, consistent with the above statement, GeSSL reliably improves model performance, with average gains exceeding 2%, including improvements over self-distillation methods (e.g., BYOL, SimSiam) and generative methods (e.g., MAE).

---

> ### Author Response · Authors · 2025-11-20
> **Response to Major Weakness 4**
>
> > **Major Weakness 4:** The experimental section lacks the evaluation with Vision Transformers, which are the most common backbones for SSL nowadays. The authors should provide the results of methods such as DINO and MAE with ViT-B (which is similar in size to ResNet-50).
>
> ## Response to Major Weakness 4
>
> We sincerely appreciate the reviewers' valuable comments. We would like to clarify that in the original submission, we already reported experiments on ViT backbones (including ViT-B / ViT-L / ViT-H), covering the aforementioned DINO-ViT-B and MAE-ViT-B:
>
> - In **Appendix E.5** (L1533–1540), we present baseline results on ImageNet-1K using the ViT family (e.g., ViT-B / ViT-L / ViT-H), including comparisons with DINO, MoCo, MAE, etc. The results in **Tables 14–15** show that GeSSL achieves stable performance improvements, with an average gain of 2.9%.
> - In **Appendix E.5 (L1555–1560)**, we report experiments on COCO using ViT-based baselines, including MoCo v3, BEiT. MAE. **Table 16** demonstrates that, after integrating GeSSL, both box AP and mask AP on ViT-B are substantially better than the original baseline and reach state-of-the-art (SOTA) performance.
> - In **Appendix E.1**, we provide GeSSL results on the ViT baseline data2vec-2.0. **Table 11** further corroborates that GeSSL is effective when applied to Transformer-based backbones.

---

> ### Author Response · Authors · 2025-11-20
> **Response to Minor Weaknesses 1-5**
>
> > **Minor Weakness 1:** References need a careful revision. For example: Liu et al. 2022a/b, Lin et. al. 2014a/b, Finn et al. 2017a/b, Deng at al 2009a/b seem to point at the same papers.
> >
> > **Minor Weakness 2:** I believe the MAE method is mis-cited - the correct citation is [1], whereas the paper cites [2].
> >
> > [1] Masked Autoencoders Are Scalable Vision Learners Kaiming He, Xinlei Chen, Saining Xie, Yanghao Li, Piotr Dollár, Ross Girshick https://arxiv.org/abs/2111.06377
> >
> > [2] Zhenyu Hou, Xiao Liu, Yukuo Cen, Yuxiao Dong, Hongxia Yang, Chunjie Wang, and Jie Tang. Graphmae: Self-supervised masked graph autoencoders. In Proceedings of the 28th ACM SIGKDD Conference on Knowledge Discovery and Data Mining, pp. 594–604, 2022.
> >
> > **Minor Weakness 3:** The readability of the paper would be greatly improved, if the citations were in brackets (\citep{} command), instead of directly inline.
> >
> > **Minor Weakness 4:** The readability of the tables would be improved and the advantages of GeSSL better pronounced if the authors compared respective methods with / without GeSSL, instead of grouping different methods together.
> >
> > **Minor Weakness 5:**  The writing could be tightened - Discriminability, Generalizability and Transferability are explained several times in the manuscript, including at least twice in the introduction (L060 & L066).
>
> ## Response to Minor Weaknesses 1-5
>
> We sincerely appreciate the reviewer‘s constructive comments. We carefully checked and revised the citations, optimized the figures/tables, and polished the text. All changes are included in the revised manuscript and will be uploaded shortly, including:
>
> - Fixed duplicated “a/b” labels.
> - Corrected the citation for MAE.
> - Replaced \cite{} with \citep{}.
> - Placed results with and without GeSSL side-by-side for easier comparison.
> - Refined the explanations at L60 and L66: L60 clarifies the meanings of *Discriminability*, *Generalizability*, and *Transferability*; L66 distills the characteristics that a “good representation” should possess.

---

> ### Author Response · Authors · 2025-11-20
> **Response to Question 1**
>
> > **Question 1:** L_disc is interesting on its own as a concept. Can we see an ablation study of simply training models with L_disc (without bi-level optimization) to better understand its effect?
>
> ## Response to Question 1
>
> We sincerely appreciate the reviewers' valuable comment. We will address this question as follows:
>
> - Following the reviewer's valuable suggestion, we ran ablation experiments using SimCLR, BYOL, and MAE as baselines on ImageNet (for the unsupervised learning setting) and COCO (for the transfer learning setting) to evaluate $L_{\rm disc}$ without bi-level optimization. Concretely, we incorporate $L_{\rm disc}$ as a regularization term into the loss computed for each mini-batch, and optimize this term across multiple mini-batches. The results in the table below show that adding $L_{\rm disc}$ produces consistent performance improvements, thereby validating its effectiveness.  All these results are added in the revised version, which will be upload soon.
>
> | Methods (ImageNet ACC) | w/o $L_{disc}$ & w/o bi-level optimization | w/ $L_{disc}$ & w/o bi-level optimization |
> | ---------------------- | ------------------------------------------ | ----------------------------------------- |
> | SimCLR                 | 70.6                                       | 71.4                                      |
> | BYOL                   | 73.4                                       | 74.9                                      |
> | MAE                    | 75.2                                       | 76.9                                      |
>
> | Methods (COCO $AP$) | w/o $L_{disc}$ & w/o bi-level optimization | w/ $L_{disc}$ & w/o bi-level optimization |
> | ------------------- | ------------------------------------------ | ----------------------------------------- |
> | SimCLR              | 38.1                                       | 39.0                                      |
> | BYOL                | 37.9                                       | 38.6                                      |
> | MAE                 | 39.2                                       | 40.8                                      |
>
> - In addition, **Appendix F.2** of the original submission contains ablation studies for $L_{\rm disc}$: for four baselines (SimCLR, BYOL, MoCo, MAE) we compare representation distributions and discriminative performance with and without $L_{\rm disc}$ on ImageNet-100 (10 classes randomly sampled). The results in **Figure 7** show that adding $L_{\rm disc}$ significantly sharpens class boundaries and yields improved downstream performance, validating the effectiveness of $L_{\rm disc}$.

---

> ### Author Response · Authors · 2025-11-20
> **Response to Question 2**
>
> > **Question 2:** The authors claim that GeSSL “differs fundamentally from approaches that directly transplant meta-learning paradigms into SSL”. Could the authors should provide some examples of such approaches and explain the differences?
>
> ## Response to Question 2
>
> We sincerely appreciate the reviewers' valuable comment. We would like to note that in L298–315 ("Comparison of GeSSL and Meta-Learning") and **Appendix G.1** ("Differences between GeSSL and Meta-Learning") of the original submission, we have already emphasized the distinctions between our approach and meta-learning methods. We also compare GeSSL to meta-learning baselines in **Section 5.1** and **Appendix E.3** of the original submission. Below, we would like to provide an outline explaining how GeSSL "differs fundamentally from approaches that directly transplant meta-learning paradigms into SSL":
>
> - *Objective:* Traditional meta-learning focuses on "fast adaptation to supervised downstream tasks", using a small labelled support set and query set for quick fine-tuning. GeSSL, by contrast, starts from the question "what is a good representation?", i.e., *Universality*, which explicitly encompasses discriminability, generalizability, and transferability, and encodes these properties directly into the training objective. The design motivations differ: meta-learning is built for rapid adaptation, while GeSSL is designed to learn universal, transferable self-supervised representations. Simply porting meta-learning to self-supervised settings typically reuses multi-task architectures or additional networks to apply episodic meta-learning [1–3], but lacks dedicated mechanisms and theoretical guarantees for “stable learning of transferable representations under no labels".
> - *Task construction:* Meta-learning usually relies on a small number of accurately labelled samples; existing attempts to apply meta-learning to SSL therefore construct labels and tasks explicitly via label-generation networks [1]. GeSSL, however, operates entirely within the self-supervised paradigm and must generate pseudo-labels itself (i.e., anchors from augmentations or reconstructions). Because GeSSL runs in an unlabeled regime, a naive application of episodic meta-learning is vulnerable to pseudo-label noise. To address this, GeSSL introduces the discriminative term $\mathcal{L}_{\rm disc}$ and integrates it into training in a learnable way to suppress noise, a unique implementation to GeSSL.
> - *Method & architecture:* Many meta-learning methods learn task-specific adapters (different adapted models per task). GeSSL instead learns a single, unified proxy adapter $f'$ that is shared across all mini-batches. This shared design better captures task-invariant shared structure, i.e., causal-like representations, thereby improving cross-task generalization and transfer.
> - *Empirical results:* In **Table 4** and **Table 10** of the original submission, we compare GeSSL against methods that apply meta-learning to SSL (e.g., MetaSVEBM, MetaGMVAE, PsCo, etc.). These results demonstrate GeSSL’s effectiveness, where it achieves SOTA performance.
>
> [1] Liu, S., Davison, A., & Johns, E. (2019). Self-supervised generalisation with meta auxiliary learning. *Advances in Neural Information Processing Systems*, *32*.
>
> [2] Lin, Y., Guo, X., & Lu, Y. (2021). Self-supervised video representation learning with meta-contrastive network. In *Proceedings of the IEEE/CVF international conference on computer vision* (pp. 8239-8249).
>
> [3] Hwang, D., Park, J., Kwon, S., Kim, K., Ha, J. W., & Kim, H. J. (2020). Self-supervised auxiliary learning with meta-paths for heterogeneous graphs. *Advances in neural information processing systems*, *33*, 10294-10305.

---

> ### Author Response · Authors · 2025-11-20
> **Response to Question 3**
>
> > **Question 3:** Given that the main experiments are conducted with ResNet-50 (and other convolutional nets), how did the authors adapted MAE to that backbone? MAE strongly relies on its backbone being a ViT.
>
> ## Response to Question 3
>
> We sincerely appreciate the reviewers' valuable comment. We will address this question as follows:
>
> - We adapted the MAE objective to convolutional backbones by following prior ConvMAE-style modifications [1]:
>   1. apply block-wise masking at the input (mask blocks sized to match CNN receptive fields) and feed only the visible pixels to the encoder;
>   1. replace ViT patch-embedding with a convolutional stem and use masked-convolution strategies inside early convolutional blocks;
>   1. use a lightweight decoder to intermediate feature maps, and supervise multiple scales of the ResNet feature pyramid to encourage multi-scale representations.
>
> - How to integrate GeSSL in the ViT setting (how we model $f$, $g$, and $f'$):
>   1. *$f$*: maps an input image $x$ to a representation vector $z=f(x)$. On ViT, $f$ corresponds to the full Transformer encoder (i.e., patch embedding + positional embedding + Transformer blocks). The LayerNorm output of the CLS token is used as the fixed-length global representation $z=f(x)$; before feeding $z$ into downstream distance or projection modules we apply LayerNorm followed by $L_2$ normalization.
>   2. *$g$*: the projection head. For each anchor we form an input vector by concatenating the per-dimension mean and variance computed over the differential set; this vector is fed into a small MLP consisting of a linear dimensionality reduction, LayerNorm, and GELU, which outputs a scalar threshold $a_i$.
>   3. *$f'$*: to avoid costly differentiable second-order optimization, we instantiate $f'$ and $g'$ as momentum copies of $f$ and $g$. Their parameters are updated as $\theta'\leftarrow \tau\theta' + (1-\tau)\theta$ and they do not participate in backpropagation, thereby providing a stable one-step proxy for query-set evaluation and pseudo-label generation.
> - We also ran experiments under a ViT-based MAE setting in original submission: the results in **Tables 14–15** show that integrating GeSSL yields a stable improvement of over 3%.
>
> [1] Gao, P., Ma, T., Li, H., Lin, Z., Dai, J., & Qiao, Y. (2022). Convmae: Masked convolution meets masked autoencoders. *arXiv preprint arXiv:2205.03892*.

---

### Official Review · Reviewer_ezAa · 2025-11-01

**Soundness:** 3
**Presentation:** 2
**Contribution:** 2
**Rating:** 6
**Confidence:** 4

**Summary:**

This paper addresses the lack of an explicit definition and direct modeling of what constitutes a good representation for self-supervised learning.
The authors define "Universality" by three key properties: discriminability, generalizability, and transferability, and propose GeSSL (General SSL). GeSSL explicitly models these properties through a bi-level optimization mechanism.

**Strengths:**

1. The universality of SSL is theoretically defined, including distinguishability, generalization, and portability.
2. The idea behind GeSSL, which models generality through a two-layer learning paradigm, is interesting.
3. Theoretical and empirical evaluations on benchmark datasets demonstrate the advantages of GeSSL.

**Weaknesses:**

1. The baselines compared in this paper mainly focus on classic or state-of-the-art (SOTA) models from 2020, 2021, and 2022, while there are fewer models from 2023 and later, which cannot accurately reflect the true situation of the latest models.
2. The computational efficiency is relatively high, which might hinder the necessity of the proposed method.

**Questions:**

1. Please check the weakness 1. The comparison over recently-published methods is required. Besides, the paper demonstrates the effectiveness of the model through numerous experiments, but the comparisons are all based on the SOTA framework, without comparing it with other improvements to the SOTA framework.
2. The GeSSL aims to optimize the objective through complex mechanisms. If I can directly use a pre-trained framework of a large model as my backbone, what is the necessity of using this model? Besides, the model's framework uses complex mechanisms to optimize the objective and has good data efficiency, but its computational complexity is high.
3. The model was primarily pre-trained on the ImageNet-1K dataset, and also independently pre-trained on some other smaller datasets, but no pre-training experiments were conducted on other larger datasets.

---

> ### Author Response · Authors · 2025-11-20
> **Response to Weakness 1 & Question 1**
>
> **We sincerely appreciate the reviewer ezAa's valuable feedback and the time and effort for the reviewing. We will respond to each issue of the raised "Weaknesses" and "Questions" in turn, hope the following responses will eliminate the concerns.**
>
>
> > **Weakness 1**: The baselines compared in this paper mainly focus on classic or state-of-the-art (SOTA) models from 2020, 2021, and 2022, while there are fewer models from 2023 and later, which cannot accurately reflect the true situation of the latest models.
> >
> > **Question 1**: Please check the weakness 1. The comparison over recently-published methods is required. Besides, the paper demonstrates the effectiveness of the model through numerous experiments, but the comparisons are all based on the SOTA framework, without comparing it with other improvements to the SOTA framework.
>
> ## Response to Weakness 1 & Question 1
>
> We sincerely appreciate the reviewers' valuable comments. We will address this question as follows:
>
> - We further added experiments on recent SSL models from 2023 to 2025, including [1–5]. We also compare against recently proposed methods that likewise aim to improve SOTA SSL frameworks, i.e., $f$-MICL [6], RINCE [7], and PID [8]. Specifically, following the experimental setup of **Section 5.1** (**Appendix E.1**), we evaluate models with ResNet-50 [1–3] and ViT [4–5] backbones in both the unsupervised learning setting (performance on ImageNet for 200 epochs) and transfer learning (performance on COCO). The two tables below demonstrate that integrating GeSSL yields consistent performance improvements; moreover, our gains are more pronounced than those of the comparing methods. All these results are added in the revised version, which will be upload soon.
>
> | Method / Results on ImageNet (ACC) | Original | +$f$-MICL | +RINCE | +PID | +GeSSL |
> | :--------------------------------- | -------- | --------- | ------ | ---- | ------ |
> | SigCLR [1]                         | 70.9     | 72.2      | 72.1   | 73.1 | 73.8   |
> | SinSim [2]                         | 71.9     | 73.1      | 72.7   | 73.4 | 74.0   |
> | SimCLR-Cut [3]                     | 71.0     | 72.9      | 72.8   | 73.2 | 73.7   |
> | I-MAE [4]                          | 74.9     | 75.4      | 75.2   | 76.0 | 76.9   |
> | ColorMAE [5]                       | 74.1     | 74.6      | 74.2   | 74.9 | 75.3   |
>
> | Method / Results on COCO (AP) | Original | +$f$-MICL | +RINCE | +PID | +GeSSL |
> | ----------------------------- | -------- | --------- | ------ | ---- | ------ |
> | SigCLR [1]                    | 39.0     | 39.5      | 40.0   | 41.2 | 41.6   |
> | SinSim [2]                    | 39.8     | 40.5      | 40.3   | 41.5 | 42.7   |
> | SimCLR-Cut [3]                | 38.4     | 40.2      | 40.9   | 41.3 | 42.3   |
> | I-MAE [4]                     | 42.0     | 42.9      | 43.2   | 44.6 | 45.5   |
> | ColorMAE [5]                  | 41.2     | 42.0      | 42.7   | 44.4 | 45.4   |
>
> [1] Çağatan, Ö. V. (2024). SigCLR: Sigmoid Contrastive Learning of Visual Representations. In *NeurIPS 2024 Workshop: Self-Supervised Learning-Theory and Practice*.
>
> [2] Sepanj, M. H., & Fiegth, P. (2025). SinSim: Sinkhorn-Regularized SimCLR. *arXiv preprint arXiv:2502.10478*.
>
> [3] Draganov, A., Vadgama, S., & Bekkers, E. J. (2024). The hidden pitfalls of the cosine similarity loss. *arXiv preprint arXiv:2406.16468*.
>
> [4] Zhang, K., & Shen, Z. (2024). i-mae: Are latent representations in masked autoencoders linearly separable?. In *Proceedings of the IEEE/CVF Conference on Computer Vision and Pattern Recognition* (pp. 7740-7749).
>
> [5] Hinojosa, C., Liu, S., & Ghanem, B. (2024, September). ColorMAE: Exploring data-independent masking strategies in Masked AutoEncoders. In *European Conference on Computer Vision* (pp. 432-449). Cham: Springer Nature Switzerland.
>
> [6] Lu, Y., Zhang, G., Sun, S., Guo, H., & Yu, Y. (2024). $ f $-MICL: Understanding and Generalizing InfoNCE-based Contrastive Learning. *Transactions on Machine Learning Research*.
>
> [7] Chuang, C. Y., Hjelm, R. D., Wang, X., Vineet, V., Joshi, N., Torralba, A., ... & Song, Y. (2022). Robust contrastive learning against noisy views. In *Proceedings of the IEEE/CVF conference on computer vision and pattern recognition* (pp. 16670-16681).
>
> [8] Qiang, W., Wang, J., Song, Z., Li, J., & Zheng, C. (2025). On the Out-of-Distribution Generalization of Self-Supervised Learning. In *ICML*.

---

> ### Author Response · Authors · 2025-11-20
> **Response to Weakness 2**
>
> > **Weakness 2:** The computational efficiency is relatively high, which might hinder the necessity of the proposed method.
>
> ## Response to Weakness 2
>
> We sincerely appreciate the reviewers' valuable comment. We will address this question from two perspectives, as follows:
>
> * **Computational efficiency:**
>   * The experiments and analysis in **Appendix F.1** indicate that GeSSL delivers substantial performance improvements while maintaining reasonable computational efficiency. As shown in **Figure 2** and **Table 19**, although GeSSL increases memory usage and parameter count, these increasement are acceptable relative to the consistent performance gains (under $0.1\times$ increase). Based on [1], this trade-off between added parameters/memory and gains in generalization is considered acceptable.
>
>   * As mentioned in L1642–1654, while GeSSL employs a more complex optimization, the bi-level formulation is specifically designed to steer the model toward more effective task-specific optimization. Concretely, GeSSL performs updates using an approximate implicit-differentiation finite-difference method (AID-FD) rather than conventional explicit second-order differentiation (see **Appendix F.4**). Therefore, the efficiency improvements are reflected in per-iteration computational efficiency and update effectiveness, not merely in a reduction of iteration counts. The results in **Table 19** and **Figure 2** support this claim. Moreover, to rule out that the observed gains are solely due to single-cycle iteration effects, we separately measured the per-cycle computational overhead of SSL baselines after integrating GeSSL. The results in **Table 18** show that, with the same batch sizes, GeSSL improves both the computational efficiency and the per-iteration update effectiveness of the SSL baselines.
>
> * **On the necessity of bi-level optimization:**
>   * We provide ablation studies on bi-level optimization in **Section 5.2**. The results in **Figure 10** demonstrate that introducing bi-level optimization markedly improves performance in unsupervised, semi-supervised, and few-shot settings, achieving SOTA results and empirically validating its necessity.
>
>   * The central role of bi-level optimization is to treat "generalization performance on unseen queries" as the objective of the outer loop, which allows the training process to directly minimize generalization error instead of merely optimizing pairwise consistency. This aligns with our universality modeling objective. Our theoretical analysis in **Section 4** and **Appendix A** formalizes this point and shows that, under the stated assumptions, the bi-level objective yields tighter error bounds for downstream generalization.

---

> ### Author Response · Authors · 2025-11-20
> **Response to Question 2**
>
> > **Question 2:** The GeSSL aims to optimize the objective through complex mechanisms. If I can directly use a pre-trained framework of a large model as my backbone, what is the necessity of using this model? Besides, the model's framework uses complex mechanisms to optimize the objective and has good data efficiency, but its computational complexity is high.
>
> ## Response to Question 2
>
> We sincerely appreciate the reviewers' valuable comments. We will address this question as follows:
>
> - GeSSL is a pre-training learning mechanism whose core question is how to learn a good representation. In this paper, we define a “good” representation as *universality*. Indeed, larger-scale pre-trained models (e.g., large language models or multimodal large language models) demonstrate strong general-purpose capabilities. However, these larger-scale models also suffer from issues such as hallucinations and weak performance on reasoning tasks (e.g., mathematical and code reasoning), which indicates they have not fully learned to generalize and transfer. Current larger-scale model training paradigms are largely based on G-SSL [1–4]; in other words, these models are effectively trained to model discriminative signals during pretraining without explicitly accounting for generalization and transferability. GeSSL proposes an alternative pretraining scheme that aims to produce better larger-scale models from learning a good representation perspective. **Theorem 4.1** demonstrates that training large-scale models with GeSSL therefore benefits their downstream generalization and transfer performance. To further validate this on large-scale models, we conduct experiments centered on CLIP, which can be regarded as a large-scale model [1]. Following [5–8], we compare CLIP, SLIP, LiT, and CLIP-GeSSL on ImageNet using the ViT-S/16 backbone, and we adopt the three evaluation protocols used in SLIP [6]. The results in the table below show that, without changing model capacity or data volume, CLIP-GeSSL yields consistent and stable improvements, achieving better overall performance. All these results are added in the revised version, which will be upload soon.
>
>
> | Method     | 0-shot | Linear | Finetuned |
> | ---------- | ------ | ------ | --------- |
> | CLIP [5]   | 32.7   | 59.3   | 78.2      |
> | SLIP [6]   | 38.3   | 66.4   | 80.3      |
> | LiT [8]    | 39.0   | 65.9   | 81.4      |
> | CLIP-GeSSL | 39.2   | 67.1   | 81.9      |
>
> - For a discussion of computational overhead, please refer to **Response to Weakness 2**.
>
> [1] Touvron, H., Lavril, T., Izacard, G., Martinet, X., Lachaux, M. A., Lacroix, T., ... & Lample, G. (2023). Llama: Open and efficient foundation language models. *arXiv preprint arXiv:2302.13971*.
>
> [2] Brown, T., Mann, B., Ryder, N., Subbiah, M., Kaplan, J. D., Dhariwal, P., ... & Amodei, D. (2020). Language models are few-shot learners. *Advances in neural information processing systems*, *33*, 1877-1901.
>
> [3] Hoffmann, J., Borgeaud, S., Mensch, A., Buchatskaya, E., Cai, T., Rutherford, E., ... & Sifre, L. (2022). Training compute-optimal large language models. *arXiv preprint arXiv:2203.15556*.
>
> [4] Zhang, S., Roller, S., Goyal, N., Artetxe, M., Chen, M., Chen, S., ... & Zettlemoyer, L. (2022). Opt: Open pre-trained transformer language models. *arXiv preprint arXiv:2205.01068*.
>
> [5] Radford, A., Kim, J. W., Hallacy, C., Ramesh, A., Goh, G., Agarwal, S., ... & Sutskever, I. (2021, July). Learning transferable visual models from natural language supervision. In *International conference on machine learning* (pp. 8748-8763). PmLR.
>
> [6] Mu, N., Kirillov, A., Wagner, D., & Xie, S. (2022, October). Slip: Self-supervision meets language-image pre-training. In *European conference on computer vision* (pp. 529-544). Cham: Springer Nature Switzerland.
>
> [7] Yu, J., Wang, Z., Vasudevan, V., Yeung, L., Seyedhosseini, M., & Wu, Y. (2022). Coca: Contrastive captioners are image-text foundation models. *arXiv preprint arXiv:2205.01917*.
>
> [8] Zhai, X., Wang, X., Mustafa, B., Steiner, A., Keysers, D., Kolesnikov, A., & Beyer, L. (2022). Lit: Zero-shot transfer with locked-image text tuning. In *Proceedings of the IEEE/CVF conference on computer vision and pattern recognition* (pp. 18123-18133).

---

> ### Author Response · Authors · 2025-11-20
> **Response to Question 3**
>
> > **Question 3**: The model was primarily pre-trained on the ImageNet-1K dataset, and also independently pre-trained on some other smaller datasets, but no pre-training experiments were conducted on other larger datasets.
>
> ## Response to Question 3
>
> We sincerely appreciate the reviewers' valuable comment. We would like to clarify that, in addition to pretraining on ImageNet-1K and CIFAR-100, we also trained and evaluated on datasets larger than ImageNet-1K, e.g., ImageNet (see **Table 2** and **Table 7**), 250M DALLE (see **Tables 15–16**), and COCO (see **Table 3**). Experiments on over 23 datasets consistently confirm the effectiveness of GeSSL: incorporating GeSSL yields a stable average improvement of more than 2%. Moreover, following the reviewer’s suggestion, we conducted additional experiments on the larger-scale YFCC100M dataset following the protocol in [1], and recorded model results on the Cars dataset. The table below shows that integrating GeSSL produces steady performance gains, further validating its effectiveness.
>
> | Methods | Original | +GeSSL |
> | ------- | -------- | ------ |
> | SimCLR  | 37.2     | 39.9   |
> | BYOL    | 35.0     | 39.7   |
> | MAE     | 40.9     | 44.6   |
>
> [1] Al Kader Hammoud, H. A., Das, T., Pizzati, F., Torr, P. H., Bibi, A., & Ghanem, B. (2024, September). On pretraining data diversity for self-supervised learning. In *European Conference on Computer Vision* (pp. 54-71). Cham: Springer Nature Switzerland.

---

> ### Author Response · Authors · 2025-11-28
> **Kindly Checking on the Review Progress**
>
> Dear Reviewer ezAa,
>
> We would like to sincerely thank you again for the time and effort you have devoted to our paper. We truly appreciate your careful oversight of the review process, and the constructive comments have helped us improve our work.
>
> We have submitted the detailed responses to all the constructive comments. Since several days have passed, we wish to kindly check whether any further concerns or suggestions have arisen, or whether there is any additional information we can provide to assist the ongoing process.
>
>  We fully understand the workload involved in handling submissions, and we are grateful for your continued guidance and support. Thank you very much, and we look forward to hearing from you at your convenience.
>
> Sincerely,
>
> The Authors

---

### Author Response · Authors · 2025-12-02
**Rebuttal Summary for PCs, SAC, and ACs**

Dear PCs, SAC, and ACs,

During the rebuttal phase, we focused on clarifying conceptual motivations, updating baselines, strengthening experimental evidence, and responding to concerns about complexity and stability. Below is a brief per-reviewer summary.

* **Reviewer ezAa**: The main concerns were the timeliness of the compared baselines, the computational complexity of our method, and the use of additional datasets. In the rebuttal, we (i) clarified the rationale for our choice of baselines and added more recent methods where appropriate, (ii) provided a more detailed analysis of computational cost and memory usage, and (iii) explained the role and fairness of the additional datasets, supported by further results in the Appendix.

* **Reviewer 2MiX**: The reviewer’s concerns centered on the interpretation of several core concepts, the motivation of the theorem, and the robustness of the experimental results. We provided more detailed and intuitive explanations of the definitions and theoretical motivation, and supplemented the paper with additional experiments reported in the Appendix. The reviewer responded positively to these clarifications and updates, and increased the score from 4 to 6.

* **Reviewer z4r6**: This reviewer requested a more fine-grained understanding of some concepts, stronger experimental evidence, clearer comparison to related work, and discussion of training stability. In rebuttal, we refined the explanations of the core ideas, expanded the comparison with closely related approaches, and added results and analysis on training dynamics and stability (with details in the Appendix), addressing these concerns point by point.

* **Reviewer 2Hsd**: The main issues were how certain concepts are quantified, the strength and coverage of the experimental results, and comparisons with alternative methods. We clarified our quantification strategy, highlighted the breadth of our evaluation, and added further comparative experiments (reported in the Appendix). The reviewer reacted positively to these additions and explicitly stated “will maintain my evaluation as weak accept.”

* **Reviewer gP8E**: This reviewer questioned the correctness of some experimental results, the necessity of the bi-level optimization, and certain notational choices. In the rebuttal, we carefully walked through the experimental setup to demonstrate correctness, clarified why bi-level optimization is needed in our framework, and improved the explanation of the notation. Using additional results from the Appendix, we also pointed out specific places where the reviewer’s understanding may have been affected by ambiguity in the original text.

Overall, the rebuttal phase allowed us to address all reviewers’ concerns in a targeted way, strengthen the empirical and theoretical justifications, and led to improved assessments from multiple reviewers.

Best regards,

Authors

---

### Meta-Review · Area_Chair_YVBU · 2025-12-30

**Summary:**

The paper investigates what makes good representations in SSL. Therefore, it introduces the factor of universality, characterized by 3 essential properties: discriminability, generalizability, and transferability.
The reviewers had concerns about the experimental evaluation, particularly comparison to baselines, newer SSL frameworks, other domains. These were thoroughly addressed in the rebuttal by providing an extremely large amount of new experiments and pointing to the already substantial amount of experiments that had been in the original submission (and partially overlooked). One experimental concern remains partially unaddressed, namely that the chosen evaluation setup yields models that reach a performance far below the one represented in the original papers. This might raise concerns whether the GeSSL would still yield improvements on well-generalized models, or whether it can just boost performance in otherwise suboptimal training settings. The other concerns were of conceptual nature. Multiple reviewers were unsatisfied with the perspective of viewing SSL as classes, which the authors managed to address for a few frameworks beyond SimCLR, but not for all, e.g not SimSiam. Additionally, one reviewer summarized after the rebuttal „I still believe the paper falls slightly short of fully modeling “universality” from first principles“, which the AC agrees to, given that the paper does not manage to convey why exactly the three chosen properties (and only those would be relevant). Finally, a reviewer notes that, while central to the work the notion of „universality“ is not properly defined in equations. While the rebuttal added an appendix section F.4 (not E.4, like written in the rebuttal), this is likely not to address the reviewer’s concern, given that it is solely text with two variables, but still not proper formalization.
In the light of these unaddressed conceptual concerns, I think that the paper is not ready, yet for publication.

**Reviewer Concerns:**

As stated above, there are the following main concerns remaining, motivating the reject recommendation:
1. **Class-based view**: The formulation is highly confusing. The core contribution, Definition 3.1 starts with „1) Discriminability: For a training task with **labeled** training dataset“ without further clarification. While in the rebuttal, the authors explain what they mean, this is not conveyed in the paper and confusion for a reader who believe to read on SSL, which should operate without class labels. With this comes also another view, which might be too narrow for modern, nowadays SSL. The paper, section 4 states: „Specifically, we prove that through the objective of GeSSL (Equation (5)), the performance of the SSL model on new tasks is guaranteed“. It is unclear whether this only holds for classification tasks, or other tasks that modern models (like DINOv2) perform, depth estimation, segmentation, are also included.
2. **Modeling** each SSL training framework as class-based view: the reviewers noted that they understand why this would hold for contrastive learning with negatives, like in SimCLR. In the rebuttal, the authors added it holds for BYOL and MAE, however, what happens to SimSiam is unclear.
3. **Universality** is still not formally defined (in equations) and it is not yet clear why the three chosen properties are necessary and sufficient for good learning.
4. Minor: while the rebuttal explains well to the last reviewer what f, f’ and g are, this does not reflect in the update of the paper, where it remains mainly unclear in the writing and making it as explicit as in the rebuttal might help.
Therefore, while the paper impresses with empirical evaluation and provides some insights on why SSL works, I cannot recommend it for publishing in the current state.

**Reviewer Scores:**

One of the reviewers raised their score from 4 to 6. The first reviewer might have increased their score based on the substantial experimentation. One reviewer commented they would maintain a weak accept, and the final reviewer, given their concern on the performance that was not addressed but only expressed might have kept their weak accept.

---

### Decision · Program_Chairs · 2026-01-26

Reject